# Interplay between c-Src and the APC/C co-activator Cdh1 regulates mammary tumorigenesis

Tao Han[1,12], Shulong Jiang [1,2,3,12], Hong Zheng[4], Qing Yin[1], Mengyu Xie[4,5], Margaret R Little[1,6], Xiu Yin[1,2], Ming Chen [7,8], Su Jung Song[7,9], Amer A. Beg[4,10], Pier Paolo Pandolfi [7] & Lixin Wan [1,11]

The Anaphase Promoting Complex (APC) coactivator Cdh1 drives proper cell cycle progression and is implicated in the suppression of tumorigenesis. However, it remains elusive how Cdh1 restrains cancer progression and how tumor cells escape the inhibition of Cdh1. Here we report that Cdh1 suppresses the kinase activity of c-Src in an APC-independent manner. Depleting *Cdh1* accelerates breast cancer cell proliferation and cooperates with *PTEN* loss to promote breast tumor progression in mice. Hyperactive c-Src, on the other hand, reciprocally inhibits the ubiquitin E3 ligase activity of APC[Cdh1] through direct phosphorylation of Cdh1 at its *N*-terminus, which disrupts the interaction between Cdh1 and the APC core complex. Furthermore, pharmacological inhibition of c-Src restores APC[Cdh1] tumor suppressor function to repress a panel of APC[Cdh1] oncogenic substrates. Our findings reveal a reciprocal feedback circuit of Cdh1 and c-Src in the crosstalk between the cell cycle machinery and the c-Src signaling pathway.

[1] Department of Molecular Oncology, H. Lee Moffitt Cancer Center and Research Institute, Tampa, FL 33612, USA. [2] Department of Oncology, Affiliated Jining NO.1 People's Hospital of Jining Medical University, Jining, Shandong 272000, P.R. China. [3] Department of Oncology, Guang'anmen Hospital, China Academy of Chinese Medical Sciences, Beijing 100053, P.R. China. [4] Department of Immunology, H. Lee Moffitt Cancer Center and Research Institute, Tampa, FL 33612, USA. [5] Department of Cancer Biology PhD Program, University of South Florida, Tampa, FL 33620, USA. [6] Nova Southeastern University, Fort Lauderdale, FL 33314, USA. [7] Cancer Research Institute, Beth Israel Deaconess Cancer Center, Department of Medicine and Pathology, Beth Israel Deaconess Medical Center, Harvard Medical School, Boston, MA 02215, USA. [8] Department of Pathology, Duke University School of Medicine, Duke Cancer Institute, Duke University, Durham, NC 27710, USA. [9] Soonchunhyang Institute of Medi-bio Science, Soonchunhyang University, Cheonan-si, Chungcheongnam-do 31151, Republic of Korea. [10] Department of Thoracic Oncology, H. Lee Moffitt Cancer Center and Research Institute, Tampa, FL 33612, USA. [11] Department of Cutaneous Oncology, H. Lee Moffitt Cancer Center and Research Institute, Tampa, FL 33612, USA. [12] These authors contributed equally: Tao Han, Shulong Jiang. Correspondence and requests for materials should be addressed to L.W. (email: lixin.wan@moffitt.org)

The anaphase promoting complex/cyclosome (APC/C or APC) plays critical roles in regulating both the M and G1 phases[1] by forming two sub-complexes APC$^{Cdc20}$ and APC$^{Cdh1}$ with two substrate-binding coactivators, Cdc20 and Cdh1, respectively[2]. Unlike Cdc20, which has restricted function in M phase, its close homolog, Cdh1 (also named FZR1), associates with the APC core complex in late M phase/early G1 phase and controls G1 phase cell cycle fate decisions. During the remainder of the cell cycle, phosphorylation of Cdh1 at its N-terminus by CDK kinases[3–8], and the inhibitory interaction with Emi1[9] abolish the interaction between Cdh1 and the APC core complex, thereby inhibiting APC$^{Cdh1}$. In addition to its function as the APC coactivator, Cdh1 has been shown to activate the Smurf1 E3 ligase in osteoblasts[10] and to suppress BRAF kinase activity in melanoma cells[8] in an APC-independent manner. These findings indicate that Cdh1 may possess unique characteristics in cell cycle phases other than M and G1, and more importantly, in malignant settings where APC$^{Cdh1}$ is largely inhibited[11].

Conditional knockout of Cdh1 in mice led to the development of epithelial tumors, suggesting a tumor suppressor role for Cdh1[12], which has been partially attributed to its roles in maintaining genomic stability as well as promoting the ubiquitination and subsequent proteolysis of a number of oncogenic substrates including Plk1, Cdc6, Skp2, and cyclin A[11]. However, deletion and mutations of Cdh1 are not frequent events in most human cancers (cBioPortal.org), suggesting that post-transcriptional and post-translational mechanisms suppress the E3 ligase activity of APC$^{Cdh1}$. Indeed, N-terminal phosphorylation-mediated inactivation of APC$^{Cdh1}$ has been highlighted as a mechanism to inhibit APC$^{Cdh1}$ [5–8]. However, it remains poorly characterized for the population of APC-free Cdh1 in tumor cells.

The c-Src (Src thereafter) non-receptor tyrosine kinase (RTK) is one of the first proto-oncogenes identified[13]. Src and other Src family kinases directly interact with and mediate the signal transduction of multiple RTKs[14]. Src functions as a signaling hub to phosphorylate and regulate a broad spectrum of cytoplasmic targets[14–20]. By controlling numerous cellular signaling pathways, Src links extracellular signals to cell proliferation, migration and apoptosis[14]. In support of an oncogenic function of Src in breast carcinogenesis[21], ectopic expression of the constitutively active mutant form of v-Src was capable to transform MCF10A cells[22].

Our findings here demonstrate that in breast tumor cells, Cdh1 suppresses Src kinase activity independent of the E3 ligase function of APC. Cdh1 binds to Src in a head-to-toe manner to restrain Src in a closed, inactive conformation. Intriguingly, with N-terminus of Cdh1 binds to the kinase domain of Src, Src catalyzes tyrosine phosphorylation of Cdh1 at Y148. Analogous to the previously identified serine/threonine phosphorylation, this feed-forward signaling functions to displace Cdh1 from the APC, thus blocking APC$^{Cdh1}$ E3 ligase activity. Furthermore, pharmacologically inhibiting Src restores the tumor suppressor function of APC$^{Cdh1}$, and synergistically suppresses the growth of triple negative breast cancer cells with MEK inhibition.

## Results

**Cdh1 negatively regulates Src kinase activity**. We have recently reported that Cdh1 suppresses the BRAF/MEK/ERK signaling cascade in melanoma cells[8]. Similar to melanoma, dysregulation of the cell cycle machinery plays an important role in driving breast carcinogenesis[23]. To this end, we unexpectedly found that depletion of Cdh1 led to the upregulation of its known ubiquitin substrates, whereas the MEK/ERK signal was not affected (Fig. 1a–c and Supplementary Fig. 1a–c). Instead, increase of Src kinase activity was observed as evidenced by the Y419-Src

activating phosphorylation (Fig. 1a–c and Supplementary Fig. 1a–c). Src drives its own activation via the autophosphorylation at Y419 while is subjected to inhibition via the Y530 phosphorylation by Csk (C-terminal Src kinase)[24]. These results indicate that in breast cancer cells, Cdh1 might negatively regulate the kinase activity of Src. In support of this notion, p-Y357-YAP[16,25] and p-Y705-STAT3[26], known Src targets, were elevated in Cdh1-depleted breast cancer cells (Fig. 1a–c and Supplementary Fig. 1a–c). Moreover, CRISPR/Cas9-mediated deletion of Cdh1 (Supplementary Fig. 1d) escalated Src activity in breast cancer cells (Fig. 1d and Supplementary Fig. 1e, f). Furthermore, compared to its WT counterparts, Cdh1$^{-/-}$ MEFs displayed stronger Src activation in response to platelet-derived growth factor (PDGF) stimuli (Fig. 1e).

Cdh1 and Cdc20 are the two substrate interacting coactivators for the APC ubiquitin E3 ligase complex[27]. However, unlike Cdh1, depletion of Cdc20 failed to induce p-Y419-Src in breast cancer cells (Fig. 1f and Supplementary Fig. 1g). In contrast to the accumulation of its known ubiquitin substrates upon Cdh1 depletion, the protein abundance of Src was not affected (Fig. 1a–e and Supplementary Fig. 1a–f), suggesting a nonproteolytic regulation of Src function by Cdh1. Since the APC core complex is required for the ubiquitination and degradation of APC$^{Cdh1}$ substrates, this finding suggests that Cdh1 might govern Src activity in an APC-independent fashion. Indeed, knockdown of APC core subunits Cdc27 and APC10 failed to escalate p-Y419-Src in breast cancer cells (Fig. 1g, h and Supplementary Fig. 1h, i).

To further substantiate the role of Cdh1 in suppressing Src function, we found that re-introducing full-length Cdh1, but not its N-terminus, into Cdh1-depleted MCF7 and MDA-MB-231 cells led to the suppression of upregulated p-Y419-Src (Fig. 1i and Supplementary Fig. 1j, k). Moreover, ectopically expression of Cdh1 in Cdh1$^{-/-}$ MEFs led to an attenuated Src activation in response to PDGF stimulation (Supplementary Fig. 1l). Further depletion of Src in Cdh1-deficient T47D and MDA-MB-231 cells eliminated the increased downstream p-YAP and p-STAT3 signals (Fig. 1j and Supplementary Fig. 1m), suggesting an important role of the Cdh1-Src axis in regulating these oncogenic pathways. Cdh1 protein level oscillates across the cell cycle[28], and we found that p-Y419-Src level decreased when Cdh1 was accumulated in MDA-MB-231 and T47D cells in synchronization experiments (Fig. 1k and Supplementary Fig. 1n–p). Moreover, depleting Cdh1 resulted in a non-fluctuating pattern of p-Y419-Src across the cell cycle (Fig. 1k and Supplementary Fig. 1n–p).

**Depletion of Cdh1 promotes breast cancer tumorigenesis**. To determine if Cdh1 deficiency accelerates the growth of breast cancer cells. We generated stable cell lines expressing control (shScramble, shScr for short) or anti-Cdh1 shRNAs. Compared to control MDA-MB-231 and BT474 cells, depletion of Cdh1 promoted the proliferation of breast cancer cells (Fig. 2a–e and Supplementary Fig. 2a–e). In line with a pro-metastatic role of Src, we found that depletion of Cdh1 in MDA-MB-231 cells led to increased cell migration (Fig. 2f, g). To further assess the importance of Src in mediating Cdh1 deficiency-induced gain of proliferation, we found that further depletion of Src from shCdh1-MDA-MB-231 cells led to a significant reduction of cell proliferation (Supplementary Fig. 2f, g) and anchorage-independent growth in soft agar (Supplementary Fig. 2h, i). Further inoculation of control and Cdh1-depleted MDA-MB-231 and BT474 cells into immunodeficient mice revealed that compared to control cells, Cdh1-depleted breast cancer cells developed larger tumors in vivo (Fig. 2h, i and Supplementary Fig. 2j, k).

To better understand the tumor suppressor function of Cdh1 in vivo, we next crossed mice harboring only one Cdh1 allele

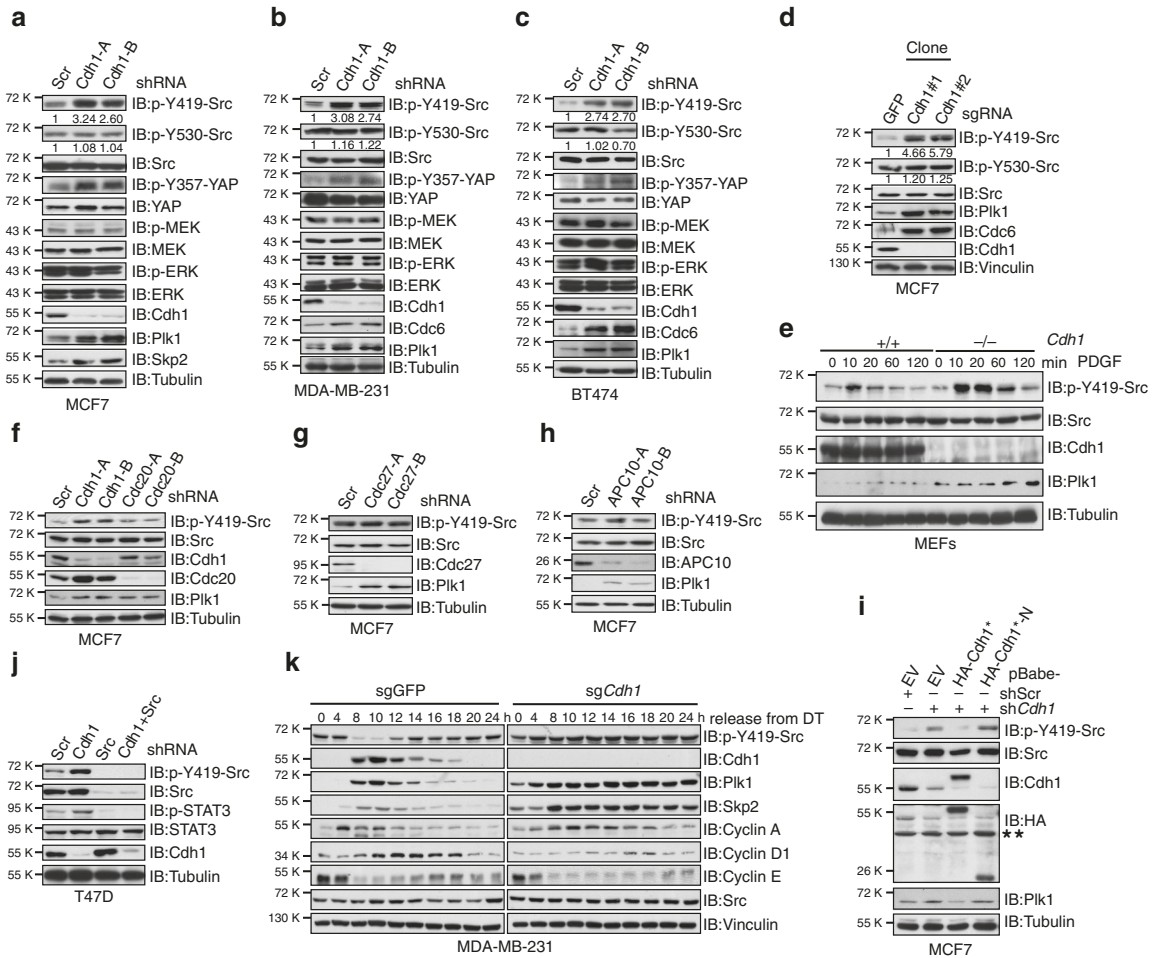

**Fig. 1** Cdh1 negatively regulates Src kinase activity in an APC-independent manner. **a–c** Immunoblot (IB) analysis of MCF7 (**a**), MDA-MB-231 (**b**), and BT474 (**c**) cells infected with control (shScramble, shScr for short) or the indicated shCdh1 lentiviral shRNA constructs. The infected cells were selected with 1 μg ml$^{-1}$ puromycin for 72 h before harvest. **d** CRISPR/Cas9-mediated deletion of Cdh1 activated Src. IB analysis of MCF7 cells infected with control (sgGFP) or sgCdh1 lentiviral construct. The infected cells were selected with 1 μg ml$^{-1}$ puromycin for 7 days before plating for single clone selection. **e** Src was activated in Cdh1$^{-/-}$ MEFs. IB analysis of WT and Cdh1$^{-/-}$ MEFs treated with 4 ng ml$^{-1}$ PDGF for the indicated periods of time after 16 h serum deprivation. **f** IB analysis of MCF7 cells infected with the indicated lentiviral shRNA constructs. The infected cells were selected with 1 μg ml$^{-1}$ puromycin for 72 h before harvest. **g**, **h** IB analysis of MCF7 cells infected with control (shScr) or the indicated shCdc27 (**g**) and shAPC10 (**h**) lentiviral shRNA constructs. The infected cells were selected with 1 μg ml$^{-1}$ puromycin for 72 h before harvest. **i** MCF7 cells stably expressing retroviral empty vector (EV), WT-, or N-Cdh1 were further infected with shScr or shCdh1 lentiviral constructs as indicated. The infected cells were selected with 1 μg ml$^{-1}$ puromycin for 72 h before harvest. *Cdh1 cDNA used in this experiment has been mutated to escape shCdh1-mediated gene silencing. ** indicates nonspecific bands. **j** IB analysis of T47D cells infected with the indicated lentiviral shRNA constructs. The infected cells were selected with 1 μg ml$^{-1}$ puromycin and 100 μg ml$^{-1}$ hygromycin for 72 h before harvest. **k** IB analysis of WCL derived from sgGFP- and sgCdh1-infected MDA-MB-231 cells that were synchronized at the G1–S boundary by double-thymidine block and then released back into the cell cycle for the indicated periods of time

(hereafter referred to as Cdh1$^{+/-}$)[12] with Pten$^{+/-}$ mice[29]. Intriguingly, Pten$^{+/-}$;Cdh1$^{+/-}$ compound mice displayed a marked increase in spontaneous mammary tumorigenesis over time compared with Pten$^{+/-}$ mice (Fig. 2j). Notably, ductal and lobular carcinomas have been observed in Pten$^{+/-}$;Cdh1$^{+/-}$ compound mice (at approximately 30% incidence) at 6–7 months old, whereas no tumor has been observed in Pten$^{+/-}$ or Cdh1$^{+/-}$ animals at the same age (Fig. 2j, k). Consistent with our observations in cell lines, a marked increase of p-Y419-Src was observed in breast tissues obtained from WT and Pten$^{+/-}$; Cdh1$^{+/-}$ mice (Fig. 2l).

**Cdh1 directly interacts with Src in a head-to-toe manner.** We next sought to determine the molecular mechanism by which Cdh1 suppresses Src kinase activity. Consistent with our results

that depletion of Cdh1, but not Cdc20, activates Src (Fig. 1f and Supplementary Fig. 1g), we found that Src specifically bound Cdh1 but not Cdc20 (Fig. 3a). On the other hand, Cdh1 specifically interacted with Src, but not two close homologs Fyn and Yes (Fig. 3b). Furthermore, we observed a direct interaction between Cdh1 and Src in vitro and at the endogenous level in cells (Fig. 3c, d and Supplementary Fig. 3a–d). Since Src mainly exerts its function in the cytoplasm[30], whereas most of the canonical APC$^{Cdh1}$ functions are carried out within the nucleus[31]. Proximity ligation assay (PLA) and immunofluorescence analyses revealed that the majority of Cdh1 and Src are localized to the cytoplasm in breast cancer cells (Fig. 3e and Supplementary Fig. 3e). This observation was supported by subcellular fractionation experiment that different from nontransformed human foreskin fibroblasts and MCF10A cells, Cdh1 is predominantly localized in the cytoplasm in breast cancer cells (Fig. 3f).

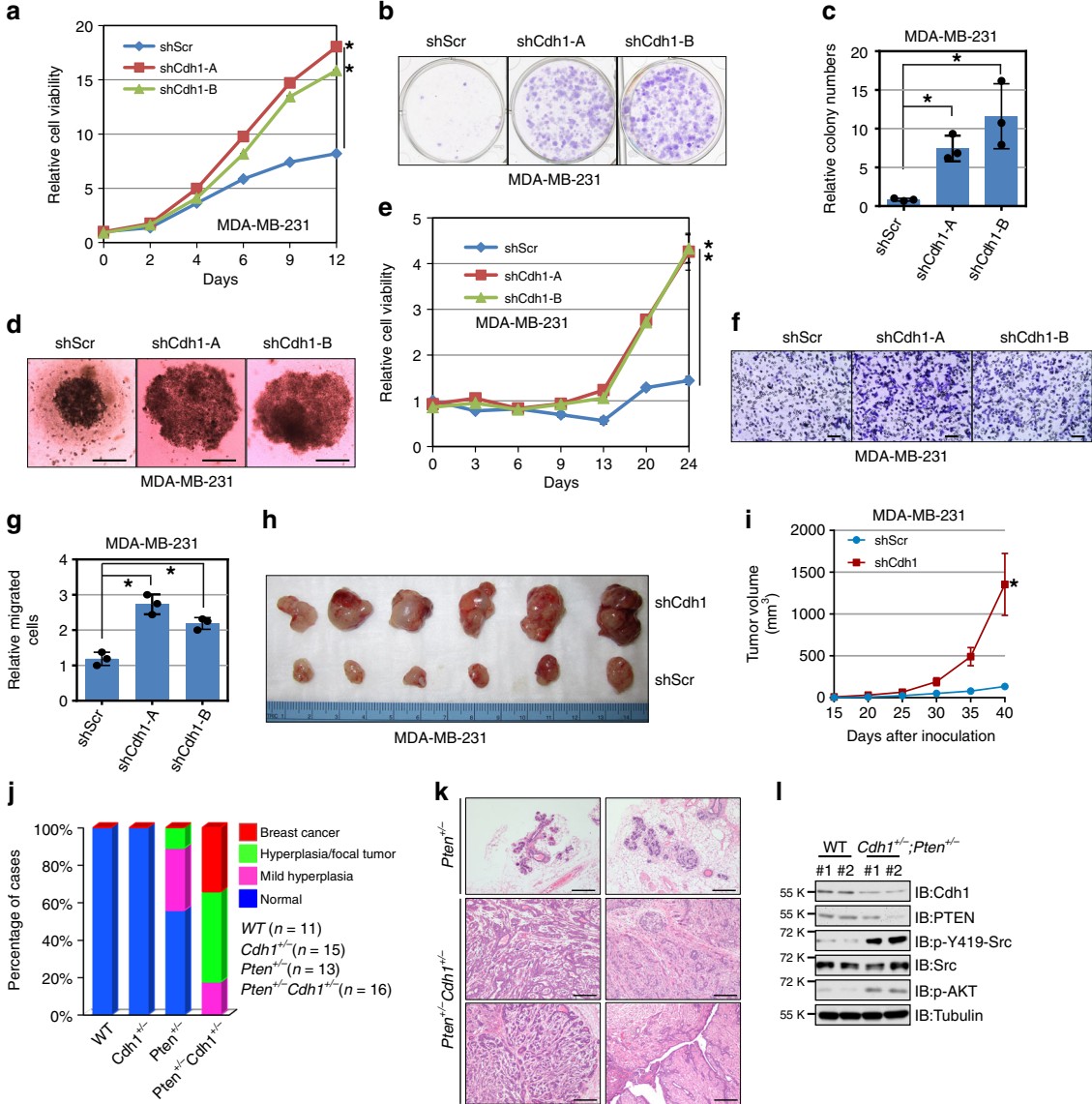

**Fig. 2** *Cdh1* deficiency facilitates breast tumorigenesis. **a** MDA-MB-231 cells infected with control shRNA (shScr) or sh*Cdh1* lentiviral constructs as described in Fig. 1b were subjected to cell proliferation assays in DMEM medium supplemented with 10% FBS for up to 12 days. Relative cell viability was determined at the indicated time points and was calculated as mean ± SD from three independent experiments. $*P < 0.05$; Student's *t* test. **b**, **c** MDA-MB-231 cells generated in (**a**) were subjected to clonogenic survival assays in DMEM medium supplemented with 10% FBS for 14 days. Crystal violet was used to stain the formed colonies (**b**) and the colony numbers were calculated as mean ± SD ($n = 3$), $*P < 0.05$; Student's *t* test (**c**). **d**, **e** MDA-MB-231 cells generated in (**a**) were subjected to 3D spheroid formation experiments in DMEM medium supplemented with 10% FBS for 24 days. The representative pictures are shown from three independent experiments (**d**), Scale bar, 100 μm. Anchorage-independent cell growth was measured using CellTiter-Glo 3D Cell Viability Assay Kit at the indicated time points (**e**). Relative cell viability was calculated as mean ± SD ($n = 3$). $*P < 0.05$; Student's *t* test (**e**). **f**, **g** MDA-MB-231 cells generated in (**a**) were subjected to Transwell migration assays. The representative pictures are shown from three independent experiments. Scale bar, 100 μm (**f**). The relatively migrated cells were calculated as mean ± SD ($n = 3$). $*P < 0.05$; Student's *t* test (**g**). **h**, **i** Tumor pictures (**h**) and the growth curves (**i**) for the xenograft experiments with the MDA-MB-231 cells generated in (**a**) were inoculated subcutaneously. In each flank of six nude mice, $3 \times 10^6$ cells were injected. The visible tumors were measured at the indicated days. Error bars represent ±SEM ($n = 6$) (**i**). **j** The incidence of mammary gland hyperplasia and tumor in *Pten*$^{+/-}$ and *Pten*$^{+/-}$;*Cdh1*$^{+/-}$ mice (6–7 months old) were quantified. **k** Hematoxylin and eosin (H&E)-stained sections of mammary tissues isolated from 6-month-old *Pten*$^{+/-}$ and *Pten*$^{+/-}$;*Cdh1*$^{+/-}$ littermates. Scale bar, 100 μm. **l** Immunoblot (IB) analysis of lysates from breast tissues of *Pten*$^{+/-}$ and *Pten*$^{+/-}$;*Cdh1*$^{+/-}$ mice

Moreover, G2 phase (9 h after released from double thymidine block) MDA-MB-231 cells exhibited a stronger Cdh1/Src binding compared to S (6 h) and G1 (18 h) phase cells (Supplementary Fig. 3f, g).

Cdh1 utilizes its WD40 repeats domain to recruit substrates to the APC core complex2 (Fig. 3g). Intriguingly, Src was able to bind both the *N*-terminal and the *C*-terminal domains of Cdh1 (Fig. 3h). The Src protein contains a *C*-terminal kinase domain and *N*-

terminal SH2 and SH3 regulatory domains (Fig. 3g)[14]. Further analysis revealed that the *C*-terminal WD40 repeats domain of Cdh1 interacted with the *N*-terminus of Src (Fig. 3i), while Cdh1 *N*-terminus specifically bound to Src *C*-terminal kinase domain (Fig. 3j). These results thus advocate a head-to-toe interacting pattern for the binding between Cdh1 and Src (Fig. 3k).

Most of the APC$^{Cdh1}$ substrates contain consensus motifs that interact with the WD40 repeats domain of Cdh1[32]. The best

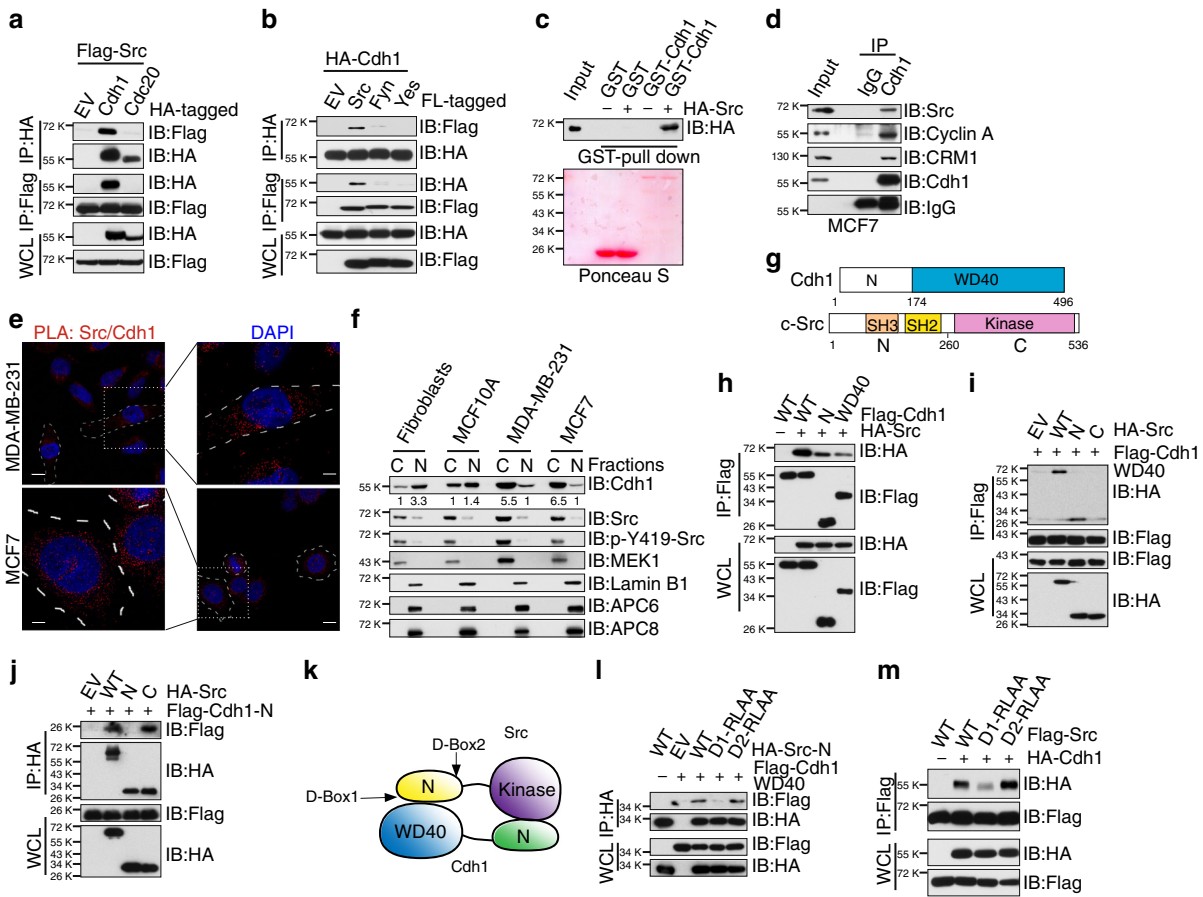

**Fig. 3** Cdh1 interacts with Src in a head-to-toe manner. **a** Immunoblot (IB) analysis of whole-cell lysates (WCL) and immunoprecipitates (IP) derived from 293T cells transfected with Flag-Src and HA-Cdh1 or HA-Cdc20. **b** IB analysis of WCL and IP derived from 293T cells transfected with HA-Cdh1 and the indicated constructs. **c** GST pull-down analysis to determine Src bound to purified recombinant GST-Cdh1, but not the GST recombinant protein. **d** Endogenous Src bound to endogenous Cdh1. IB analysis of WCL and anti-Cdh1 IP derived from MCF7 cells. **e** Proximity ligation assay (PLA) demonstrated a cytoplasmic interaction between Cdh1 and Src proteins in MCF7 and MDA-MB-231 cells. Scale bar, 50 μm. **f** IB analysis of cytoplasmic and nuclear fractions derived from the indicated cell lines. The relative band intensities of Cdh1 were quantified using ImageJ. **g** Schematic illustrations of the domain structures of Cdh1 and Src. **h** IB analysis of WCL and IP derived from 293T cells transfected with HA-Src and the indicated Flag-Cdh1 constructs. **i** IB analysis of WCL and IP derived from 293T cells transfected with Flag-Cdh1 WD40 domain and the indicated HA-Src constructs. **j** IB analysis of WCL and IP derived from 293T cells transfected with Flag-Cdh1 N terminus (aa1–174) and the indicated HA-Src constructs. **k** A schematic illustration of how Cdh1 interacts with Src in a head-to-toe manner. **l** IB analysis of WCL and IP derived from 293T cells transfected with Flag-Cdh1 WD40 domain and the indicated HA-Src constructs. **m** IB analysis of WCL and IP derived from 293T cells transfected with HA-Cdh1 and the indicated Flag-Src constructs

characterized motif is the destruction box (D-box for short, Supplementary Fig. 3h)[28]. Examination of the *N*-terminal sequence of Src revealed two candidate D-boxes (Supplementary Fig. 3h). Notably, D-box1 is unique to Src while D-box2 is found in all Src family kinases (Supplementary Fig. 3h). Compared to WT-Src, mutating D-box1 (D-box1-RLAA) abolished the binding between WD40 repeats domain of Cdh1 and Src *N*-terminal domain, while mutating D-box2 failed to influence the interaction (Fig. 3l), indicating that D-box1 is the primary site through which WD40 repeats domain of Cdh1 binds to Src (Fig. 3k). The D-box1 mutation also led to reduced binding between full-length Cdh1 and Src (Fig. 3m).

**Cdh1 inhibits Src by locking Src in a closed conformation.** Consistent with an inhibitory role of Cdh1 in regulating Src function, we found that purified recombinant GST-Cdh1 or Cdh1 suppressed the auto-phosphorylation of immuno-purified Src at Y419 but not Y530 in vitro (Fig. 4a and Supplementary Fig. 4a, b). Similarly, recombinant Cdh1 protein inhibited the kinase activity of Src in promoting CDK1[33] and p85 cortactin

(CTTN) phosphorylation[34] in vitro (Fig. 4b and Supplementary Fig. 4c–e).

In basal status, Src adopts a closed, inhibitory conformation shaped by the binding between the *N*-terminal SH2 domain and the phosphorylated Y530 at the *C*-terminus of the kinase domain (Fig. 4c)[35]. As shown in Fig. 4d, Cdh1 enhanced the binding between *N*- and *C*-terminal domains of Src, while failed to promote the interaction between *C*-terminal Src and the D-box1 mutated *N*-terminal Src (Fig. 4e), which exhibited reduced binding to Cdh1 (Fig. 3l, m). With such evidence, we thus hypothesized a model in which Cdh1 secures a closed, inhibitory conformation of Src by reinforcing the binding between *N*- and *C*-terminus of Src (Fig. 4f). In support of this notion, we found that only full-length Cdh1 but not the *N*-terminus of Cdh1 effectively suppressed Src (Supplementary Fig. 4f). Moreover, Cdh1 failed to inhibit the kinase activity of D-box1-mutated Src (D1-RLAA) (Fig. 4g and Supplementary Fig. 4g). Compared to WT- and D2-RLAA-Src, MDA-MB-231 cells transiently expressing the Cdh1-binding-deficient D1-RLAA Src exhibited higher Src activity (Fig. 4h).

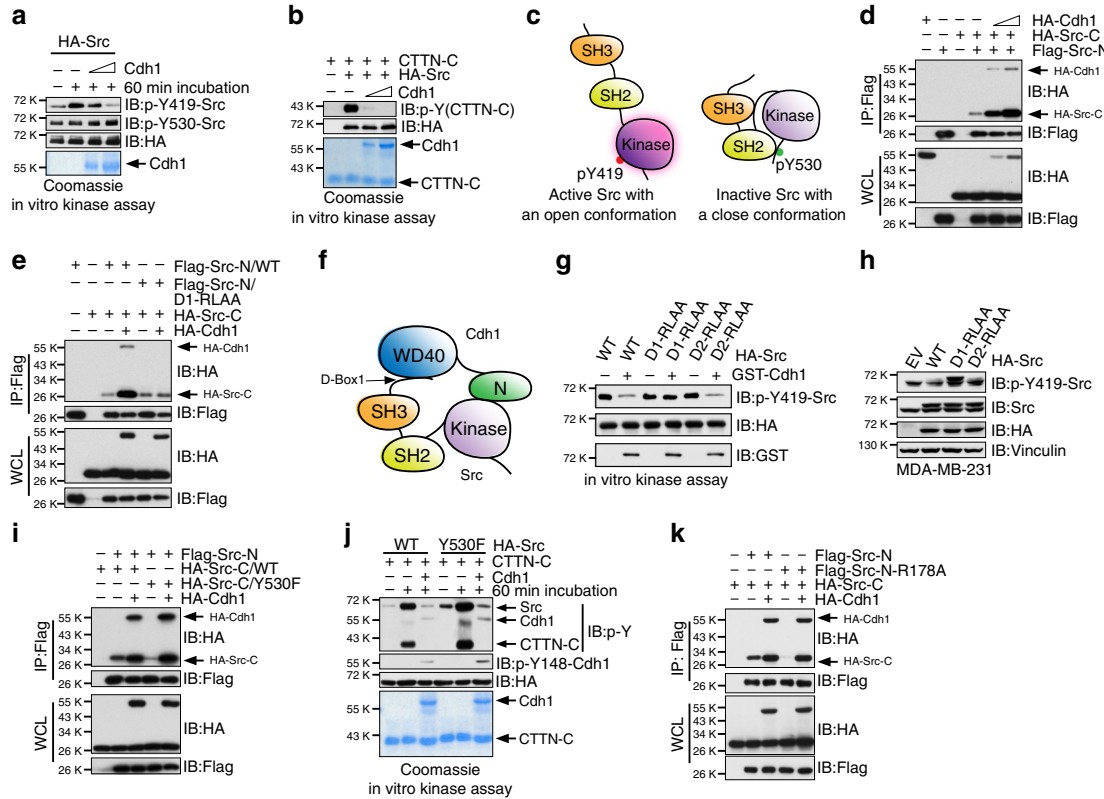

**Fig. 4** Cdh1 suppresses Src kinase activity by locking Src in a closed conformation. **a** Immuno-purified HA-Src from 293T cells were incubated with bacterially purified and cleaved recombinant Cdh1 as indicated. The reaction was performed at 30 °C for 60 min followed by SDS-PAGE and Immunoblot (IB) analyses. **b** Recombinant Cdh1 inhibited the kinase activity of Src to promote the C-terminal domain of p85 cortactin (CTTN-C) phosphorylation in vitro. **c** A schematic illustration of the open and closed conformations of the Src kinase. **d** IB analysis of whole-cell lysates (WCL) and immunoprecipitates (IP) derived from 293T cells transfected with HA-Cdh1 and the indicated Src constructs. **e** IB analysis of WCL and IP derived from 293T cells transfected with HA-Cdh1 and the indicated Src constructs. **f** A schematic illustration of how Cdh1 secures the closed, inhibitory conformation of Src by reinforcing the binding between N- and C- terminus of Src. **g** Immuno-purified HA-Src proteins from 293T cells were incubated with bacterially purified GST-Cdh1. The reaction was performed at 30 °C for 60 min and followed by SDS-PAGE and IB analyses. **h** IB analysis of WCL derived from MDA-MB-231 cells transfected with EV, WT, D-box1-mutated, and D-box2-mutated HA-Src as indicated. Cells were harvested 48 h after transfection. **i** IB analysis of WCL and IP derived from 293T cells transfected with HA-Cdh1 and the indicated Src constructs. **j** Immuno-purified WT and Y530F-HA-Src from 293T cells were incubated with bacterially purified Cdh1 and CTTN-C. The reaction was performed at 30 °C for 60 min and followed by SDS-PAGE and IB analyses. **k** IB analysis of WCL and IP derived from 293T cells transfected with HA-Cdh1 and the indicated Src constructs

Y530 phosphorylation of Src promotes intracellular interaction between the N-terminal SH2 domain and the C-tail pY530 site, a mechanism that inactivates Src[20]. The Y530F mutant of Src, which fails to adopt the closed conformation, represents a constitutively active form of Src[24]. Compared to the C-terminal kinase domain of WT-Src, we observed a weak interaction between the Y530F-Src-C and Src-N (Fig. 4i). Notably, introducing Cdh1 promoted the interaction between Y530F-Src-C and Src-N (Fig. 4i), suggesting that Cdh1 could suppress the aberrantly active Y530F-Src. Indeed, we found that although Y530F-Src displayed a higher p-Y419, it was still sensitive to Cdh1-mediated inhibition (Fig. 4j and Supplementary Fig. 4h). In line with this finding, compared to WT-Src, an increased interaction between Cdh1 and Y530F-Src was observed (Supplementary Fig. 4i). On the other hand, the Src [531]QPG-[531]EEI mutation (Supplementary Fig. 4j)[36] displayed reduced binding with Cdh1 (Supplementary Fig. 4i). Furthermore, although a SH2-deficient, R178A-Src-N mutant[37] failed to bind the C-terminal kinase domain of WT- or Y530F-C-Src, the addition of Cdh1 promoted the interaction between both parts of the Src protein (Fig. 4k and Supplementary Fig. 4k), suggesting that Cdh1-Src interaction is not controlled by Src Y530 phosphorylation. Together these results unravel the molecular mechanism

that Cdh1 suppresses Src kinase activity via blocking the Src into a closed, inactive conformation (Fig. 4f).

**Cdh1-binding deficient Src promotes breast tumorigenesis.** Based on the results that Cdh1 suppresses breast tumorigenesis partly through inhibiting Src, we next intended to examine if generating breast cancer cell lines harboring the Cdh1-interacting deficient mutant accelerates breast cancer cell growth. Consistent of a higher kinase activity of transiently expressed D1-RLAA-Src in MDA-MB-231 cells (Fig. 4h), compared to WT- and D2-RLAA-Src, MDA-MB-231, and BT474 cells expressing the D1-RLAA mutant form of Src displayed higher Y419-Src phosphorylation as well as elevated downstream p-YAP and p-STAT3 signals (Fig. 5a and Supplementary Fig. 5a). Further cell proliferation (Fig. 5b and Supplementary Fig. 5b) and clonogenic survival assays (Fig. 5c, d and Supplementary Fig. 5c, d) demonstrated that D1-RLAA-Src-expressing MDA-MB-231 and BT474 cells exhibited faster growth compared to cells harboring WT- or D2-RLAA-Src.

Moreover, MDA-MB-231 and BT474 cells bearing the D1-RLAA Src mutant formed an increased number of colonies in the soft agar (Fig. 5e, f and Supplementary Fig. 5e, f), and displayed

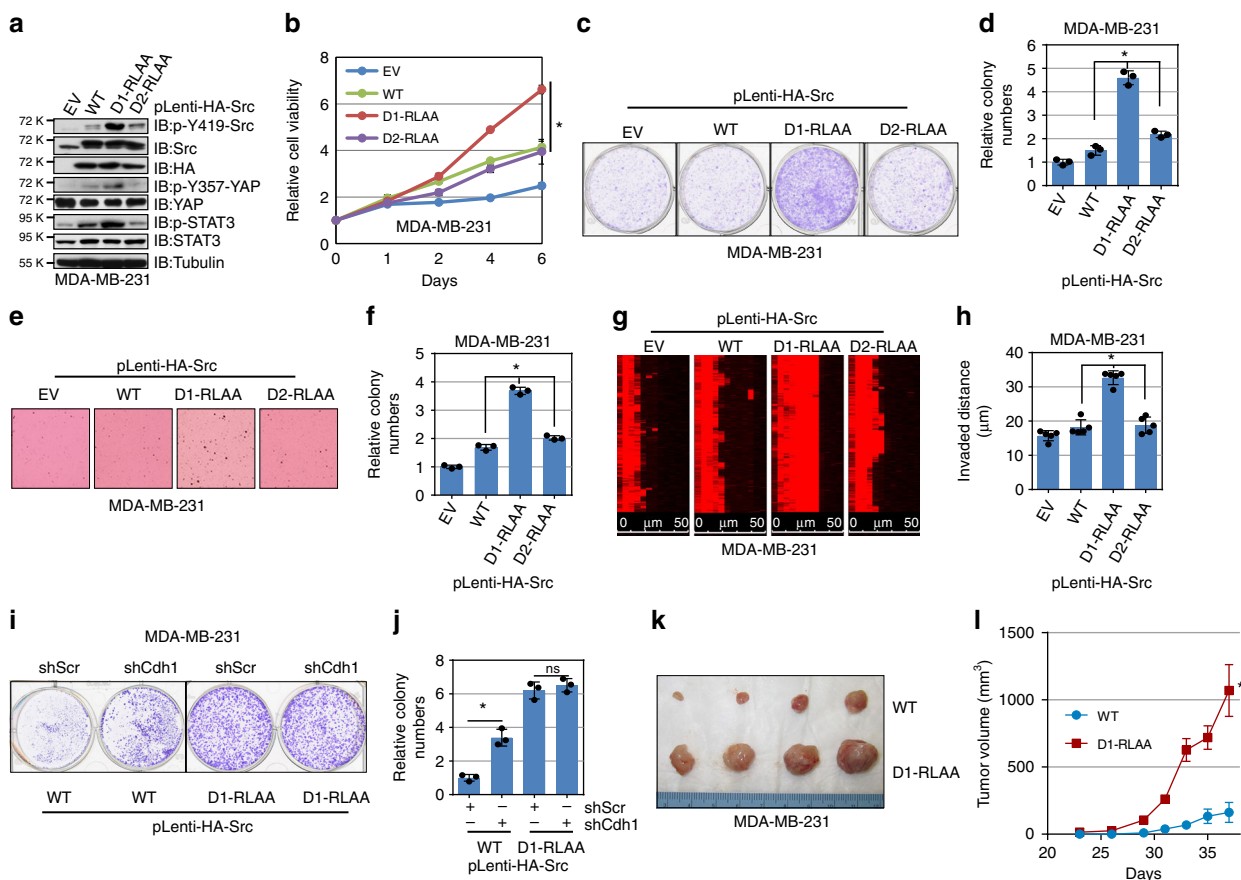

**Fig. 5** Cdh1-binding deficient Src mutant promotes breast cancer cell growth. **a** Immunoblot (IB) analysis of whole-cell lysates (WCL) derived from MDA-MB-231 cells stably expressing empty vector (EV), WT-Src, D-box-1-mutated, or D-box-2-mutated Src. **b** MDA-MB-231 cells generated in (**c**) were subjected to cell proliferation assays in DMEM medium supplemented with 10% FBS for 6 days. Cell viability was determined at the indicated time points. The relative cell viability was calculated as mean ± SD ($n = 3$). *$P < 0.05$; Student's $t$ test. **c, d** MDA-MB-231 cells generated in (**a**) were seeded for clonogenic survival assays (1000 cells per well). Fourteen days after plating, crystal violet was used to stain the formed colonies (**c**) and the relative colony numbers were counted as mean ± SD ($n = 3$), *$P < 0.05$; Student's $t$ test (**d**). **e, f** MDA-MB-231 cells generated in (**a**) were subjected to soft agar colony formation assays for 21 days. Formed colonies were stained with iodonitrotetrazolium chloride (INT) (**e**). The relative colony numbers were calculated as mean ± SD ($n = 3$). *$P < 0.05$; Student's $t$ test (**f**). **g, h** MDA-MB-231 cells generated in (**a**) were subjected to Matrigel invasion assays for 24 h. Invaded cells were stained with phalloidin (**g**). The invaded distance was calculated as mean ± SD ($n = 5$). *$P < 0.05$; Student's $t$ test (**h**). **i, j** MDA-MB-231 cells generated in Supplementary Fig. 5g were seeded for clonogenic survival assays (1000 cells per well). Fourteen days after plating, crystal violet was used to stain the formed colonies (**i**) and the relative colony numbers were counted as mean ± SD ($n = 3$), *$P < 0.05$; Student's $t$ test (**j**). **k, l** Tumor pictures (**k**) and the growth curves (**l**) for the xenograft experiments with the MDA-MB-231 cells generated in (**a**) were inoculated subcutaneously. In each flank of the nude mice, $1 \times 10^6$ cells were injected. The visible tumors were measured at the indicated days. Error bars represent ±SEM ($n = 4$). *$P < 0.05$; Student's $t$ test

increased capability to invade the matrigel (Fig. 5g, h). Notably, depletion of *Cdh1* in D1-RLAA-Src-expressing MDA-MB-231 cells (Supplementary Fig. 5g) failed to further induce cell proliferation in the plate or in soft agar (Fig. 5i, j and Supplementary Fig. 5h, i), suggesting that the observed increase of cell growth of D1-RLAA-Src is primarily through escaping the negative regulation of Cdh1. Furthermore, compared to WT-Src, D1-RLAA-Src-expressing MDA-MB-231 cells formed larger tumors in immunodeficient mice (Fig. 5k, l). These results together demonstrate increase tumorigenicity of D-box1 mutated Src in propelling breast tumor growth both in vitro and in vivo.

**Src phosphorylates Cdh1 at Y148 to inhibit APC^Cdh1 activity.** The binding between *N*-Cdh1 and the kinase domain of Src suggests that *N*-Cdh1 could be phosphorylated by Src. In vitro kinase assay results demonstrated that Src, but not the enzymatically inactive K298R mutant, phosphorylated *N*-Cdh1 (Fig. 6a–b). Moreover, WT-, but not K298R-Src, promoted the

tyrosine phosphorylation of Cdh1 both in cells (Fig. 6c) and in vitro (Supplementary Fig. 6a). It is intriguing that compared to full-length Cdh1, *N*-terminal Cdh1 was phosphorylated by Src at a substantially higher level (Fig. 6d). This could be explained by the inhibitory action toward Src by full-length Cdh1 but not its *N*-terminal truncation, which is not capable to restrain Src in a closed conformation (Fig. 4f). Indeed, as shown in Fig. 6d, p-Y419-Src was largely abolished by full length but not *N*-terminal Cdh1.

Notably, unlike Src, other Src family tyrosine kinases including Fyn, Yes, and Syk, failed to promote Cdh1 phosphorylation (Fig. 6e). Analysis of Cdh1 *N*-terminal sequence revealed four candidate tyrosine sites (Fig. 6f and Supplementary Fig. 6b). Among these tyrosines, only mutating Y148 to phenylalanine (Y148F) abolished Src-mediated Cdh1 phosphorylation (Fig. 6g, h). A p-Y148-Cdh1 specific antibody was generated and the kinase assays performed in vitro and in cells validated the specificity of this antibody in recognizing the Y148-Cdh1 phosphorylation (Fig. 6h and Supplementary Fig. 6c, d).

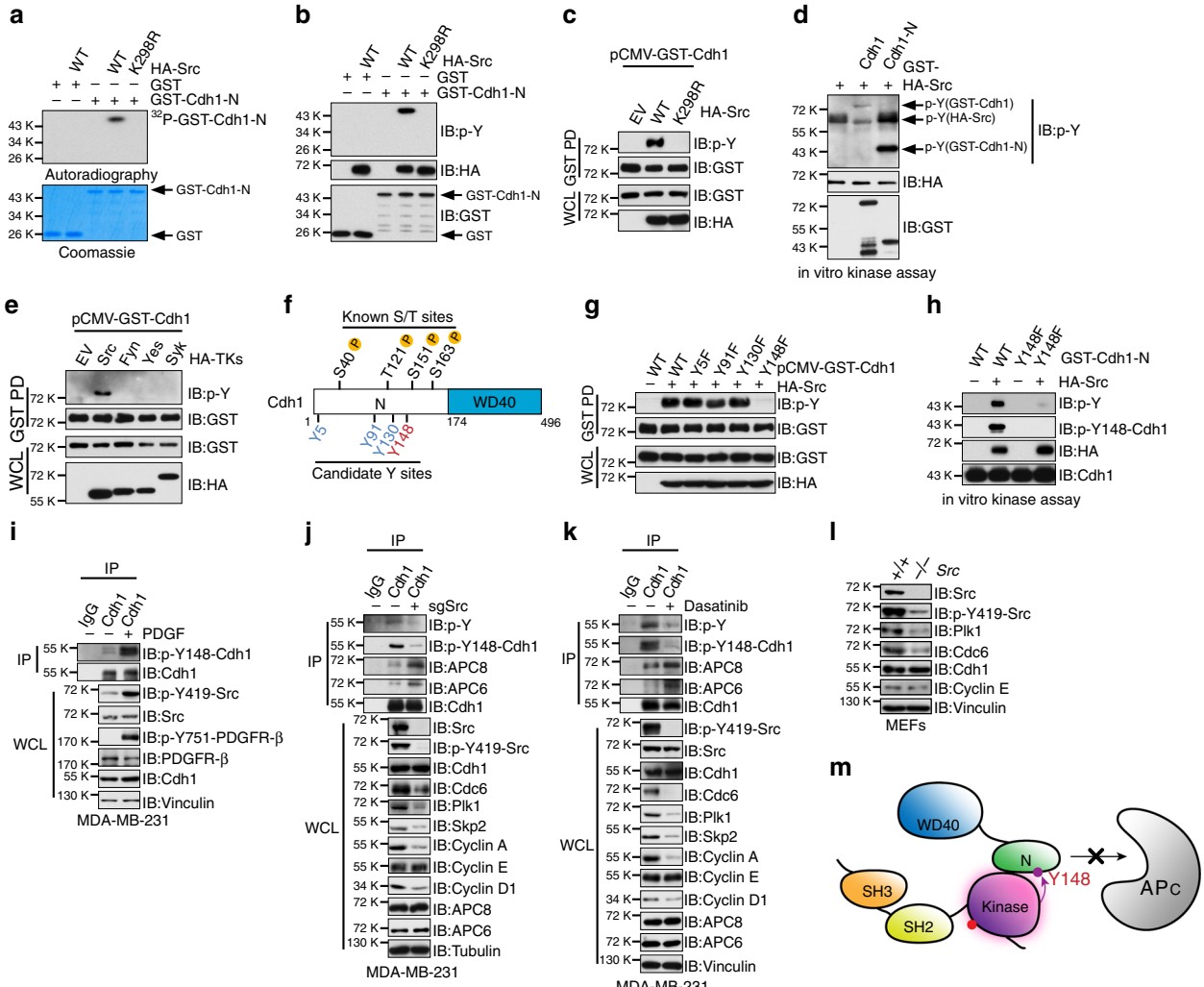

**Fig. 6** Src phosphorylates Cdh1 at Y148 to inhibit APC^Cdh1 E3 ligase activity. **a, b** In vitro kinase assays showing that bacterially purified GST-Cdh1 N-terminus could be phosphorylated by immuno-purified wild type (WT) Src, but not the enzymatic deficient K298R-HA-Src mutant, by autoradiography (**a**) or immunoblot (IB) analysis using the phospho-tyrosine (p-Y) antibody (**b**). **c** IB analysis of whole-cell lysates (WCL) and GST-pull down precipitates (GST PD) derived from 293T cells transfected with pCMV-GST-Cdh1 and the indicated HA-Src constructs. **d** In vitro kinase assays showing that immuno-purified Src kinase phosphorylated both bacterially purified GST-Cdh1 and its N terminus. **e** IB analysis of WCL and GST PD derived from 293T cells transfected with pCMV-GST-Cdh1 and the indicated HA-tyrosine kinases (TKs) constructs. **f** A schematic illustration of the previously identified phospho-serine/threonine sites as well as the candidate phospho-tyrosine sites of Cdh1 that could be phosphorylated by the Src kinase. **g** IB analysis of WCL and GST PD derived from 293T cells transfected with HA-Src and the indicated pCMV-GST-Cdh1 constructs. **h** In vitro kinase assays showing that immuno-purified WT-HA-Src failed to promote the phosphorylation of bacterially purified Y148F mutated-GST-Cdh1 N-terminus. **i** IB analysis of WCL derived from MDA-MB-231 cells that were treated with 4 ng ml$^{-1}$ PDGF for 30 min as indicated. Cells were serum-starved overnight before treatment. **j** IB analysis of WCL and anti-Cdh1 IP derived from control or sg*Src*-infected MDA-MB-231 cells. Cells were pretreated with 4 ng ml$^{-1}$ PDGF for 30 min before harvest. **k** IB analysis of WCL and anti-Cdh1 IP derived from DMSO or dasatinib-treated MDA-MB-231 cells. Cells were pretreated with 4 ng ml$^{-1}$ PDGF for 30 min before harvest. **l** IB analysis of WCL derived from WT and *Src*$^{-/-}$ MEFs. **m** A schematic illustration of phosphorylation of Cdh1 at Y148 by Src disrupts the binding between Cdh1 N terminus and the APC core complex

Moreover, treatment of MDA-MB-231 cells with PDGF stimulated p-Y148-Cdh1 (Fig. 6i).

Src has been shown binding to its substrate via a conserved motif located within the activation segment of its kinase domain[14] (Supplementary Fig. 6e). We found that deletion of the Src substrate binding motif abolished the binding between Src and *N*-Cdh1 (Supplementary Fig. 6f, g). Furthermore, the *C*-terminal domain of the well-characterized Src substrate, p85 cortactin (CTTN), efficiently displaced Cdh1 *N*-terminus from Src in vitro (Supplementary Fig. 6h), again suggesting Cdh1 *N*-terminus as a Src substrate.

*N*-terminal Cdh1 binds to the APC core complex to fulfill the E3 ligase activity of the APC^Cdh1 holoenzyme (Supplementary Fig. 6i)[38]. Serine and threonine phosphorylation of Cdh1 at multiple sites disrupts the binding between Cdh1 *N*-terminus and the APC core complex[5–8] (Fig. 6f). Consistent with the role of Src as a Cdh1 upstream tyrosine kinase, deleting Src or pharmacologically inhibiting Src using dasatinib[21] led to a reduction of Cdh1 tyrosine phosphorylation (Fig. 6j, k and Supplementary Fig. 6j, k), as well as an increased binding between Cdh1 and APC core subunits APC6 and APC8 (Fig. 6j, k and Supplementary Fig. 6j, k). In support of this notion, Y148F- and 4A-Cdh1

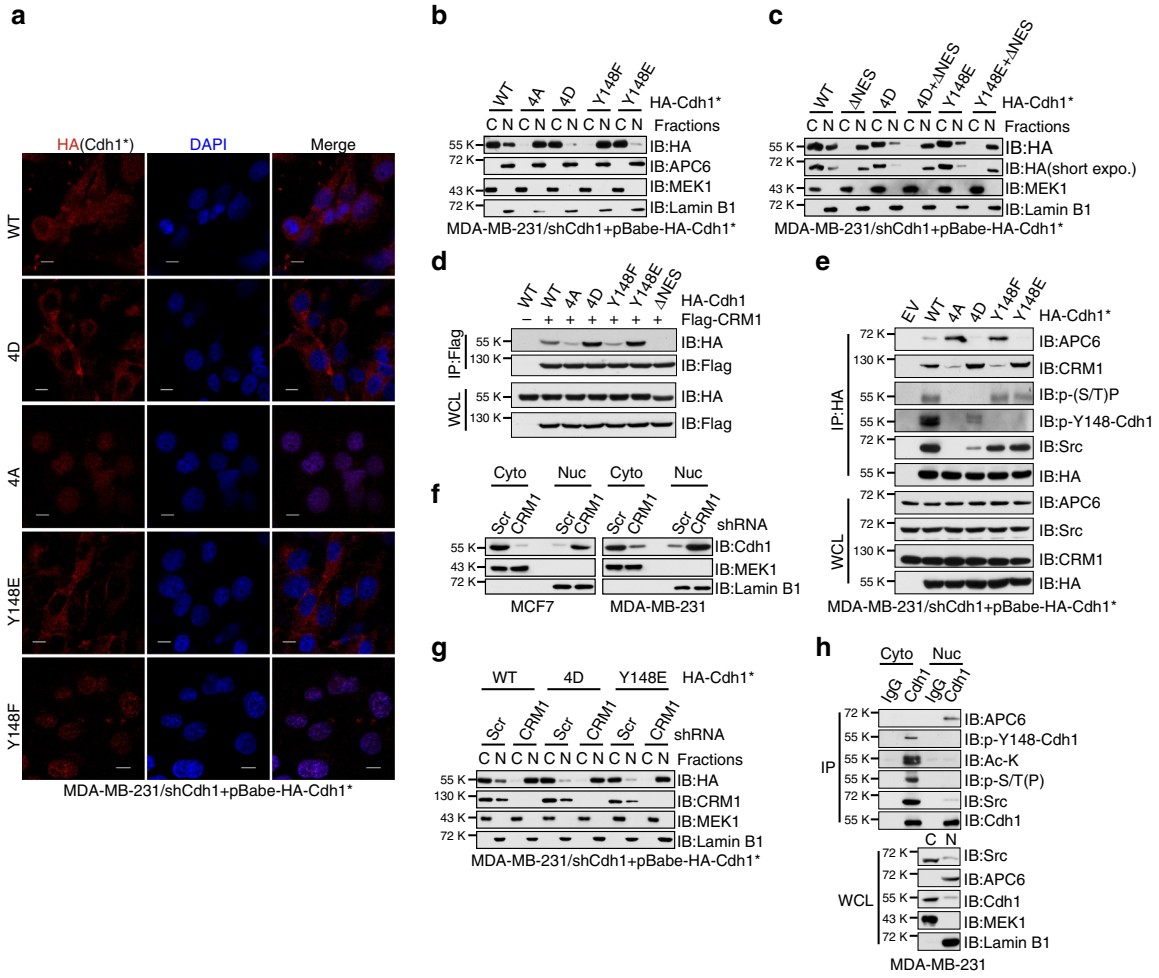

**Fig. 7** *N*-terminal phosphorylated Cdh1 is translocated to the cytoplasm by CRM1. **a** Immunofluorescent staining of sh*Cdh1*-MDA-MB-231 cells stably expressing WT-, 4D-, 4A-, Y148E-, or Y148F-Cdh1 using anti-HA(Cdh1) antibody, DAPI was used for DNA staining. *Cdh1 cDNA used in this experiment has been mutated to escape sh*Cdh1*-mediated gene silencing. Scale bar, 50 μm. **b** MDA-MB-231 cells generated in (**a**) were subjected to cytoplasm/ nucleus fractionation followed by immunoblot (IB) analysis. **c** sh*Cdh1*-MDA-MB-231 cells stably expressing the indicated Cdh1 retroviral constructs were subjected to cytoplasm/nucleus fractionation followed by IB analysis. **d** IB analysis of whole-cell lysates (WCL) and anti-Flag immunoprecipitates (IP) derived from 293T cells transfected with Flag-CRM1 and the indicated HA-Cdh1 constructs. **e** IB analysis of WCL and anti-HA IP derived from MDA-MB-231 cells generated in (**a**). **f** MCF7 and MDA-MB-231 cells were infected with shScr or sh*CRM1* lentiviral shRNA constructs. The infected cells were selected with 1 μg ml$^{-1}$ puromycin for 72 h before harvest for cytoplasm/nucleus fractionation and IB analysis. **g** sh*Cdh1*-MDA-MB-231 cells stably expressing WT-, 4D-, Y148E-Cdh1 were infected with shScr or sh*CRM1* lentiviral shRNA constructs. The infected cells were selected with 1 μg ml$^{-1}$ puromycin for 72 h before harvest for cytoplasm/nucleus fractionation and IB analysis. **h** MDA-MB-231 cells were subjected to cytoplasm/nucleus fractionation followed by anti-Cdh1 IP and IB analyses

(Supplementary Fig. 6l) exhibited higher activity to promote poly-ubiquitination of *N*-Cdc20, a bona fide substrate for APC$^{Cdh1}$ (Supplementary Fig. 6m). In contrast, Y148E- and 4D-Cdh1 (Supplementary Fig. 6l) failed to facilitate *N*-Cdc20 ubiquitination (Supplementary Fig. 6m). Furthermore, Src deficiency in both MEFs and breast cancer cells resulted in a decrease of various known APC$^{Cdh1}$ substrates (Fig. 6j–l and Supplementary Fig. 6j, k) without a significant impact on the cell cycle progression (Supplementary Fig. 6n, o). These results thus illustrate a negative regulation of APC$^{Cdh1}$ function by Src via direct phosphorylating Cdh1 at Y148 (Fig. 6m).

**CRM1 facilitates cytoplasmic translocation of Cdh1**. A previous report demonstrated that the *N*-terminal serine/threonine phosphorylation mimetic 4D-Cdh1 was predominantly localized to the cytoplasm, whereas the phosphorylation-deficient 4A-Cdh1

was found mainly in the nucleus[39], which is consistent with our observation in MDA-MB-231 cells (Fig. 7a–b). Intriguingly, Y148E-Cdh1 resembled the localization of 4D-Cdh1, while Y148F-Cdh1 mimicked 4A-Cdh1 (Fig. 7a, b). Notably, although a small portion of WT-Cdh1 was found in the nucleus, the majority of WT-Cdh1 appeared as cytoplasmic (Fig. 7a, b), similar to what we found for endogenous Cdh1 in both MCF7 and MDA-MB-231 cells (Fig. 3e, f). The close proximity of S151 and S163 to the nuclear localization signal (NLS) sequence found in the Cdh1 *N*-terminus (Supplementary Fig. 7a) has been suggested as the cause for Cdh1 *N*-terminal phosphorylation-mediated cytoplasmic retention[39], which might also be the major reason accounting for the p-Y148-induced cytoplasmic localization of Cdh1 (Fig. 7a, b).

Cdh1 recruits substrates to the APC core complex for ubiquitination. Since the APC complex is primarily localized in the nucleus[40] (Fig. 3f), it is thus mysterious how Cdh1 shuttles between the cytoplasm and the nucleus. Protein export from the

nuclear typically relies on the specific binding to exportin proteins via certain nuclear export signals (NES)[41]. Using the NetNES 1.1 Server (http://www.cbs.dtu.dk/services/NetNES/)[42], we found that Cdh1 contains a leucine-rich NES at L184-L192 (Supplementary Fig. 7a, b), right next to its NLS signal (K156-K177)[39]. Importantly, deletion of the NES sequence led to a predominant nuclear localization of WT-, 4D-, and Y148E-Cdh1 in MDA-MB-231 cells (Fig. 7c). Leucine-rich NES motifs mediate the binding between cargo proteins and exportins, particularly the export receptor CRM1/XPO1[43]. Co-immunoprecipitation experiments demonstrated that endogenous Cdh1 bound to endogenous CRM1 (Fig. 3d and Supplementary Fig. 3c). Moreover, Cdh1/CRM1 interaction was diminished when the NES sequence was deleted (Fig. 7d). Notably, 4D- and Y148E-Cdh1 exhibited a stronger binding to CRM1 compared to 4A- and Y148F-Cdh1 (Fig. 7d), indicating that hyper-phosphorylated Cdh1 might be a better cargo for CRM1 presumably due to its compromised interaction with the APC core complex. In support of this notion, we found that in contrast to 4A- and Y148F-Cdh1, 4D- and Y148E-Cdh1 displayed enhanced binding to CRM1 but not APC6 (Fig. 7e). The competitive interaction with Cdh1 between APC core complex and CRM1 thus advocate for a model in which *N*-terminal hyper-phosphorylated Cdh1, which binds poorly to the APC core complex, could be exported by CRM1 to the cytoplasm (Supplementary Fig. 7c).

Although CRM1 knockdown or inhibiting CRM1 using Leptomycin B did not affect Cdh1 expression (Supplementary Fig. 7d, e), a significant enrichment of Cdh1 in the nuclear fraction was observed in CRM1-deficient breast cancer cells (Fig. 7f and Supplementary Fig. 7f). Moreover, depletion of CRM1 in 4D- and Y148E-Cdh1-expressing MDA-MB-231 cells led to predominant nuclear retention of 4D- and Y148E-Cdh1 (Fig. 7g). Further analysis of Cdh1 *N*-terminal phosphorylation and its interacting proteins by fractionation unveiled that serine/threonine phosphorylated, Y148 tyrosine phosphorylated or acetylated[44] species of Cdh1 were all localized in the cytoplasm, while its binding with APC6 could only be found in the nucleus (Fig. 7h).

**Synergistic suppression of TNBC by Src and MEK inhibitions**. Having identified Src as an upstream kinase to displace Cdh1 from the APC core complex (Fig. 6m), we next sought to explore if inhibition of Src could restore the E3 ligase activity of APC^Cdh1 in breast cancer cells. Notably, by using multiple breast cancer cell lines, we found that pharmacologically inhibiting Src using dasatinib led to the reduction of known APC^Cdh1 substrate Plk1 in a similar fashion as using MEK inhibitor PD0325901[45], which has been shown previously to inhibit Cdh1 *N*-terminal phosphorylation and to restore APC^Cdh1 function in melanoma cells[8] (Fig. 8a, b and Supplementary Fig. 8a, b). Treatment of triple negative breast cancer cells with dasatinib led to reduced p-Y148-Cdh1 while PD0325901 suppressed serine/threonine phosphorylation of Cdh1 (Fig. 8a, b). Accompanied by decreased phosphorylation, increased interaction with APC6 was observed from Cdh1 immunoprecipitates (Fig. 8a, b). More importantly, the combinational treatment of dasatinib and PD0325901 resulted in a more dramatic increase of Cdh1/APC6 binding and a further reduction of Plk1 levels (Fig. 8a, b). These results together support a role for Src and MEK inhibitors to restore the ubiquitin E3 ligase activity of APC^Cdh1 by inhibiting Cdh1 *N*-terminal phosphorylations.

Given a tumor suppressor role for Cdh1 in breast cancer cells, our findings indicate that Cdh1 *N*-terminal phosphorylation could be a therapeutic target for breast cancers. Of note, dasatinib and PD0325901 exhibited IC50 doses around 1–4 μM in both

MDA-MB-231 and SUM159PT TNBC cells as a single-agent (Supplementary Fig. 8c–f). Intriguingly, combinational treatment of MDA-MB-231 and SUM159PT cells with dasatinib and PD0325901 exhibited a cooperative effect in suppressing cell survival (Fig. 8c, d and Supplementary Fig. 8g, h). Notably, dasatinib and PD0325901 displayed a synergistic effect in suppressing cell viability (Fig. 8e and Supplementary Fig. 8i), as evidenced by combination indices (CI) less than 0.8 for the majority of the combinations (Supplementary Fig. 8j, k).

To directly examine if dasatinib and PD0325901-mediated suppression of TNBC cell survival is at least partly due to the restoration of APC^Cdh1 function, shCdh1-MDA-MB-231 cells were stably expressed with WT-Cdh1, 4D-Cdh1 or Y148E-Cdh1 at an expression level analogous to endogenous Cdh1 (Fig. 8f and Supplementary Fig. 8l). As shown in Fig. 8f, compared to parental and WT-Cdh1-expressing cells, known APC^Cdh1 substrates were minimally affected by Src and MEK inhibitions in 4D- or Y148E-Cdh1-expressing cells. Furthermore, 4D- or Y148E-Cdh1-expressing cells were refractory to both single-agent and combinational treatment (Fig. 8g–j). Similarly, further depletion of Src in 4D- or Y148E-Cdh1-expressing MDA-MB-231 cells failed to suppress cell proliferation compared with WT-Cdh1 bearing cells (Supplementary Fig. 8m, n). In contrast to 4D- and Y148E-Cdh1, MDA-MB-231 cells expressing 4A- or Y148F-Cdh1 showed reduced survival (Supplementary Fig. 8o).

In addition to dasatinib and PD0325901, we found that saracatinib and trametinib, which target Src and MEK, respectively, exhibited similar efficacy in restoring APC^Cdh1 E3 ligase activity (Supplementary Fig. 8p, q). Furthermore, combinational treatment of saracatinib and trametinib in MDA-MB-231 cells suppressed cell survival (Supplementary Fig. 8r, s). Similar to dasatinib, saracatinib suppressed Cdh1 Y148 phosphorylation and promoted the binding between Cdh1 and APC core subunits APC6 and APC8 (Supplementary Fig. 8t). Combinational treatment of SUM159PT TNBC xenografts with dasatinib and trametinib exhibited a suppression of tumor growth in the mouse (Fig. 8k and Supplementary Fig. 8u). Analyses of tumor samples revealed a reduction of APC^Cdh1 substrates after combinational treatment (Supplementary Fig. 8v). Our results hence suggest that in addition to direct inhibition of Src and MEK kinases in TNBC cells, Src and MEK inhibitors cooperate to restore the E3 ligase activity of APC^Cdh1. And as a result, a cohort of APC^Cdh1 substrates, including Plk1, Cdc6, Skp2, cyclin A, are eliminated, which eventually helps to amplify and sustain the anti-cancer function of these inhibitors (Supplementary Fig. 9h).

## Discussion

Cdh1 is implicated in harnessing malignances through maintaining genomic stability as well as restraining oncogenic signals such as Polo like kinases, mitotic cyclins, Skp2, and BRAF[11,46]. In line with previous reports that loss of *Cdh1* augments genomic instability[12], we observed an increase of aneuploid cells in *Cdh1*-deleted MDA-MB-231 cells (Supplementary Fig. 2l, m). However, different from our previous observation that *Cdh1* knockdown led to premature senescence in primary fibroblasts[47], in breast cancer cells (Fig. 1 and Fig. 2), although depletion of *Cdh1* led a transient growth arrest, cells without *Cdh1* eventually gained proliferative advantage over the control cells (Fig. 2a, b and Supplementary Fig. 2a, b). We postulate that tumor cells might utilize oncogenic potential to overcome Cdh1-deficiency caused genomic stress while taking advantage of the elevated Cdh1 downstream targets to facilitate their growth and metastasis.

Our results reveal that the oncogenic Src signaling is one of the targets for Cdh1 in constraining breast carcinogenesis. Instead of promoting Src ubiquitination and degradation, Cdh1 binds Src to

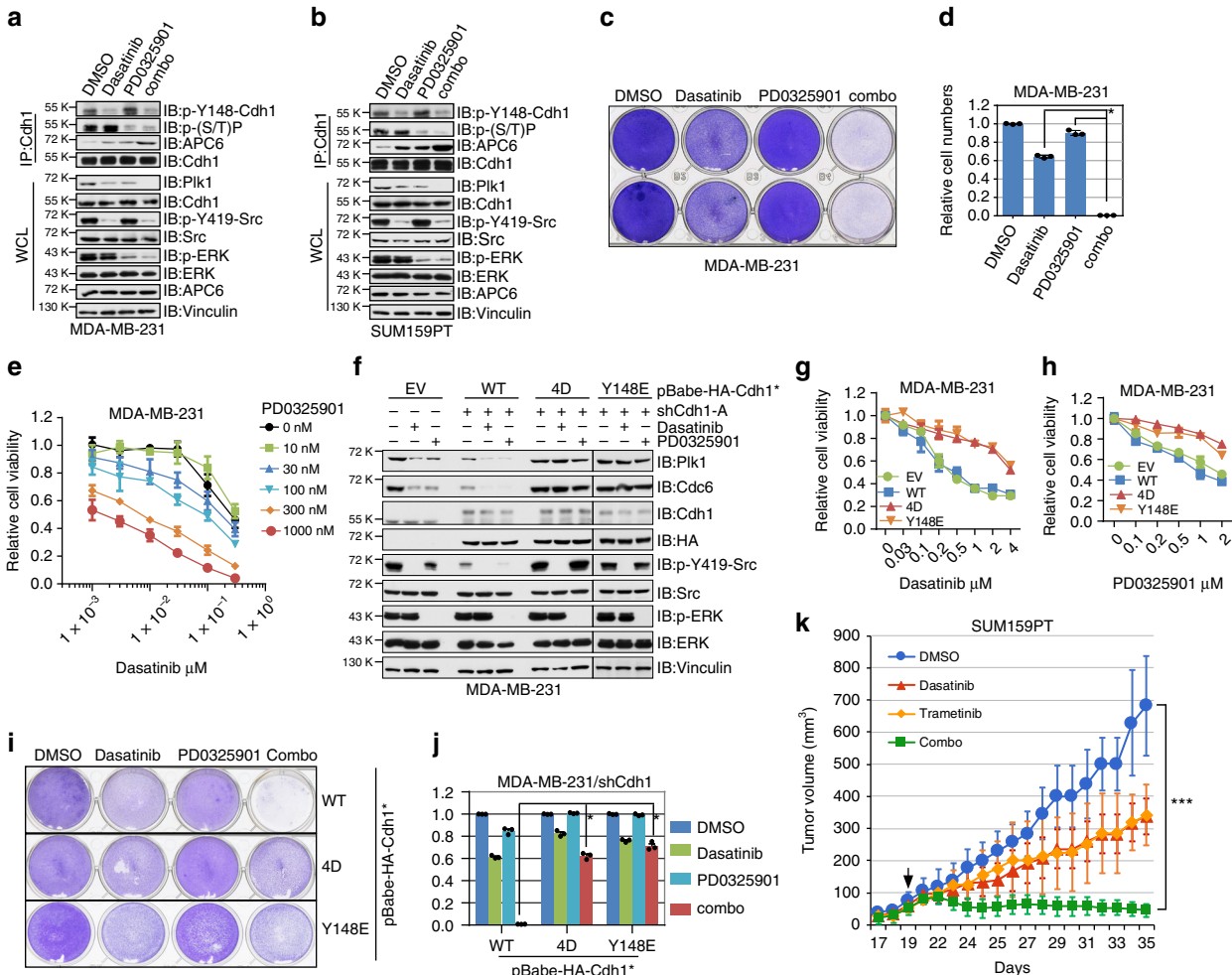

**Fig. 8** Src and MEK inhibitors synergistically suppress breast cancer cell survival. **a**, **b** N-terminal phosphorylation and protein levels of APC$^{Cdh1}$ ubiquitin substrates were reduced upon Src and/or MEK inhibition in MDA-MB-231 (**a**) and SUM159PT (**b**) cells. Immunoblot (IB) analysis of MDA-MB-231 (**a**) and SUM159PT (**b**) cells treated with either 100 nM Src inhibitor dasatinib, 100 nM MEK inhibitor PD0325901, a combination of these two inhibitors (combo), or DMSO as a negative control for 24 h before harvest. **c**, **d** Cells were treated with 300 nM dasatinib, 300 nM PD0325901, or the combination of dasatinib with PD0325901 for 3 days before being fixed and stained with crystal violet (**c**). Relative colony numbers were calculated as mean ± SD (n = 3). *P < 0.05; Student's t test (**d**). **e** Dose–response curves of MDA-MB-231 cells treated with dasatinib, PD0325901 or the combination. Relative cell viability was determined after 72 h treatment. Combination indexes (CI) were calculated using the CompuSyn software as shown in Supplementary Fig. 8j. **f** IB analysis of EV-, WT-Cdh1-, 4D-Cdh1-, and Y148E-Cdh1-expressing MDA-MB-231 cells treated with either 300 nM dasatinib, 300 nM PD0325901, or DMSO as a negative control for 24 h before harvest. *Cdh1 cDNA used in this experiment has been mutated to escape sh*Cdh1*-mediated gene silencing. **g**, **h** Dose–response curves of EV-, WT-Cdh1-, 4D-Cdh1-, or Y148E-Cdh1-expressing MDA-MB-231 cells treated with dasatinib (**g**) and PD0325901 (**h**). Relative cell viability was determined after 72 h treatment. **i**, **j** WT-, 4D-, and Y148E-Cdh1-expressing MDA-MB-231 cells were treated with 300 nM dasatinib, 300 nM PD0325901, or the combination of dasatinib with PD0325901 for 3 days before being fixed and stained with crystal violet (**i**). Relative colony numbers were calculated as mean ± SD (n = 3). *P < 0.05; Student's t test (**j**). **k** Growth curves of the SUM159PT xenograft tumors with the vehicle, dasatinib, trametinib or the combinational treatment started as the arrow indicates. In each flank of six nude mice, 3 × 10⁶ cells were injected. The tumor volumes were measured at the indicated days. Error bars represent ± SEM (n = 8). ***P < 0.001; Student's t test

reinforce the inhibitory conformation of the Src kinase in an APC-independent fashion (Fig. 4f). The Cdh1-binding motif identified in Src is close to its N-terminus, where the myristoylation occurs[48] (Supplementary Fig. 3h). Myristoylation of Src facilitates its plasma membrane enrichment and activation[48]. We found that a myristoylation-deficient G2A-Src mutant[49] bound Cdh1 to the same extent as the WT-Src (Supplementary Fig. 3i). Moreover, deletion of *Cdh1* in MCF7 cells did not significantly alter the subcellular localization of Src (Supplementary Fig. 3j), and the Cdh1-binding deficient D1-RLAA-Src showed a similar subcellular distribution as WT-Src in MDA-MB-231 cells (Supplementary Fig. 5j).

In addition to the N-terminal myristoylation, PKA has been shown to phosphorylate Src at S17, which promotes Src activation[50]. S17 is located in the D-box1 motif of Src (Supplementary Fig. 3h), thus it is important to determine if S17 phosphorylation modulates Cdh1/Src interaction. As shown in Supplementary Fig. 3k, compared to WT-Src, neither the phosphorylation mimetic S17D mutation nor the phosphorylation-deficient S17A mutation affected Cdh1/Src interaction. Furthermore, when breast cancer cells were treated with PKA agonizts Forskolin and IBMX (3-isobutyl-1-methylxanthine), although a strong increase of p-S133-CREB was observed, p-S17-Src and p-Y419-Src remained unchanged (Supplementary Fig. 3l–n). On the

contrary, p-S17-Src could be induced by PKA agonists in HCT116 colon cancer and A375 melanoma cell lines (Supplementary Fig. 3o, p). These results suggest that PKA-dependent S17 Src phosphorylation might not be a major event in breast cancer cells.

Our results underpin the existence of an APC-free, cytoplasmic population of Cdh1 in breast cancer cells whose function has not been fully explored. Quantification of Cdh1 and Src proteins in 293T cells revealed an approximately 4:1 ratio between Cdh1 and Src (Supplementary Fig. 4l). This ratio also applies to breast cancer cells based on the similar expression levels of Cdh1 and Src in multiple cell lines (Supplementary Fig. 4m). The stoichiometry of Cdh1 and Src proteins substantiates the potential of Cdh1 in suppressing Src activity via a direct interaction (Fig. 4f). The accumulation of Cdh1 in the cytoplasm might account for the observed higher Cdh1 protein levels in breast cancer cells compared to nontransformed cells (Supplementary Fig. 3q). Notably, although in breast cancer cells Cdh1 exhibited an elevated protein abundance, APC$^{Cdh1}$ substrates were also stabilized (Supplementary Fig. 3q), indicating an inactive Cdh1 population in breast cancer cells.

It is noteworthy that although 100 nM MEK inhibitor PD0325901 effectively eradicated ERK activity in breast cancer cells (Supplementary Fig. 9a, b), such dose or higher exhibited marginal efficacy in restraining MDA-MB-231 and SUM159PT cell proliferation (Supplementary Fig. 8d, f). Intriguingly, higher doses of PD0325901 led to a moderate suppression of Y419-Src phosphorylation in MDA-MB-231 and SUM159PT cells (Supplementary Fig. 9a, b), whereas high doses of dasatinib displayed no inhibitory effect on p-ERK levels (Supplementary Fig. 9c, d). We found that PD0325901 failed to directly suppress the kinase activity of Src (Supplementary Fig. 9e), suggesting that ERK might interfere with the binding between Cdh1 N-terminus and the Src kinase domain, thereby attenuating Cdh1-mediated Src inhibition. Indeed, the phosphorylation mimetic 4D-Cdh1 mutation displayed reduced binding to Src (Supplementary Fig. 9f), while compared to WT-Cdh1, 4D-Cdh1 failed to suppress the kinase function of Src in vitro (Supplementary Fig. 9g). These results indicate potential crosstalk between Cdh1 N-terminus serine/threonine and tyrosine phosphorylations.

Src has long been highlighted as a potent target for breast cancer therapy[21]. Previous pre-clinical studies found that inhibiting Src using dasatinib suppressed breast tumor growth both in vitro and in vivo[21]. However, the efficacy of dasatinib monotherapy in breast cancer clinical trials was disappointing[21]. Having identified Src, ERK, and CDK2/4 as upstream kinases that inhibit the tumor suppressor function of APC$^{Cdh1}$, here we explore the combinational treatment of TNBC cells using Src inhibitor dasatinib and MEK inhibitor PD0325901, both of which only displayed moderate efficacy in suppressing TNBC cells survival (IC50 at about 1–4 μM). We observed a synergy between dasatinib and PD0325901 at relatively low doses (Fig. 8e and Supplementary Fig. 8i–k). For instance, treatment of MDA-MB-231 cells with 100 nM dasatinib and 300 nM PD0325901 exhibited a 75% response rate and the combination index was 0.67; treatment of SUM159PT cells with 100 nM dasatinib and 100 nM PD0325901 exhibited an 88% response rate and the combination index was 0.10 (Supplementary Fig. 8j, k). This cooperative suppression of cell growth by Src and MEK inhibitors was also observed in a TNBC xenograft model (Fig. 8k and Supplementary Fig. 8u, v). Taken together, our studies here demonstrate a multilayered interplay between Cdh1 and Src, which might represent as an example of the highly cross-linked, dynamic cancer signaling networks (Supplementary Fig. 9h).

## Methods

**Plasmids.** pLenti-PGK-Hygro-DEST(w530-1) (#19066), lentiCRISPRv2 (#52961), pDONR223-Src (#23934), pDONR223-Yes1 (#23938), pRK5-Fyn (#16032), pDONR223-Syk (#23907), Flag-hCRM1 (#17647) were obtained from Addgene. pcDNA3-myc-Cdh1, pCS2 + HA-Cdh1, pCS2 + HA-Cdc20, pGEX-4T-1-Cdh1, pBabe-Hygro-HA-Cdh1 and pBabe-Hygro-HA-Cdh1-N(1–174) were obtained from Dr. Wenyi Wei[8]. p85 Cortactin (CTTN) plasmid was a kind gift from Drs. Edward Seto and Xiaohong Zhang. Full-length Src and its N-terminal and C-terminal domains were subcloned into pcDNA3-C-terminal HA, pFlag-CMV2, pGEX-4T-1 and pLenti-PGK-Hygro-HA vectors. Full-length Cdh1 and its N-terminal and C-terminal domains were subcloned into pcDNA3-HA, pFlag-CMV2, and pGEX-6P-1 vectors. Yes, Fyn and Syk were subcloned into pcDNA3-HA and pFlag-CMV2 vectors. Lentiviral shRNA constructs were generated by inserting shRNA targeting sequences into pLKO.1 vector. The shRNA sequence targeting human Src is GACAGACCTGTCCTTCAAGAA, targeting human CRM1 is GCTCAAGAAGTACTGACACAT. shCdh1, shCdc20, shAPC10 and shCdc27 constructs were obtained from Dr. Wenyi Wei[8]. Lentiviral sgRNA constructs were generated by inserting guide RNA sequences into lentiCRISPRv2 vector. The guide RNA sequence targeting human Src is TAACCGCTCTGACTCCCGTC. The guide RNA sequence targeting human Cdh1 is GCAGTACACGGAGCACCTGG.

**Antibodies.** Anti-Src (2123, 1:2000), anti-p-Y419-Src (6943, 1:2000), anti-p-Y530-Src (2105, 1:2000), anti-p-S17-Src (5473, 1:2000), anti-p-Y1000 (8954, 1:1000), anti-p-MAPK/CDK Substrates (PXS*P or S*PXR/K) (2325, 1:1000), anti-p-T202/pY204-ERK1/2 (4370, 1:3000), anti-ERK1/2 (4695, 1:3000), anti-p-S217/p-S221-MEK1/2 (9154, 1:3000), anti-MEK1/2 (9122, 1:3000), anti-p-Y705-Stat3 (9145, 1:1000), anti-YAP (14074, 1:1000), anti-CREB (9197, 1:2000), anti-p-S133-CREB (9198, 1:2000), anti-Exportin-1/CRM1 (46249, 1:1000), anti-PDGFR-β (3169, 1:1000), anti-p-Y751-PDGFR-β (4549, 1:1000), anti-Lamin B1 (13435, 1:1000) and anti-GST (2622, 1:2000) antibodies were purchased from Cell Signaling Technology. Anti-cyclin A (H-432, 1:1000), anti-cyclin E (HE-12, 1:1000), anti-cyclin D1 (C-20, 1:1000), anti-STAT3 (F-2, 1:1000), anti-Plk1 (F-8, 1:1000), anti-APC10 (B-1, 1:1000), anti-Cdc6 (180.2, 1:1000), anti-Cdc27 (AF3.1, 1:1000), anti-Cdh1 (DCS-266, 1:1000), anti-Cdc20 (E-7, 1:2000), anti-Cdc2 (17, 1:1000), anti-Vinculin (H-10, 1:1000). anti-c-Myc (9E10, 1:2000) and polyclonal anti-HA (Y-11, 1:2000) antibodies were purchased from Santa Cruz. Anti-Tubulin (T-5168, 1:2000) and, anti-APC6 (A301-165A, 1:1000), and anti-APC8 (A301-181A, 1:1000) antibodies were purchased from Bethyl Labs. Polyclonal anti-Flag antibody (F-2425, 1:2000), monoclonal anti-Flag (F-3165, 1:2000) antibody, anti-Flag agarose beads (A-2220), anti-Flag agarose beads (A-2220) anti-HA agarose beads (A-2095) as well as peroxidase-conjugated anti-mouse secondary antibody (A-4416, 1:2000) and peroxidase-conjugated anti-rabbit secondary antibody (A-4914, 1:2000) were purchased from Sigma. Monoclonal anti-SKP2 antibody (32–3300, 1:2000) was purchased from Thermo Fisher Scientific. Anti-p-Y357-YAP (ab62751, 1:500) antibody was purchased from Abcam. Monoclonal anti-HA antibody (MMS-101P, 1:2000) was purchased from Covance. Human anti-centromere antibody (ACA) derived from human CREST patient serum was purchased from Antibodies, Inc. (#15–235–0001, 1:1000). Polyclonal anti-p-Y148-Cdh1(1:200) antibody was generated by Genescript.

**Cell culture, transfection, and infection.** 293T, HEK293, MCF7, T47D, HCT116, BT474, MCF10A, SH-SY5Y, Hs587T, and ZR75–1 were obtained from ATCC. MDA-MB-231 and SUM159PT are kind gift from Dr. Andriy Marusyk. Src$^{−/−}$ MEFs are kind gift from Drs. Philippe Soriano and Akira Imamoto. Cdh1$^{−/−}$ MEFs are kind gift from Dr. Marcos Malumbres[12]. Immortalized human ovarian epithelial cells are kind gift from Drs. Hidetaka Katabuchi and Tohru Kiyono. Immortalized human foreskin fibroblasts are kind gift from Dr. Wenyi Wei. 293 T, HEK293, MCF7, T47D, HCT116, Hs587T, ZR75-1, MEFs, and immortalized human foreskin fibroblasts were grown in Dulbecco's Modified Eagle Medium (DMEM) supplemented with 10% fetal bovine serum (FBS), 2 mM Glutamine, 100 units ml$^{-1}$ penicillin and 100 μg ml$^{-1}$ streptomycin. BT474 was grown in RPMI1640 medium supplemented with 10% FBS, 2 mM Glutamine, 100 units ml$^{-1}$ penicillin and 100 μg·ml$^{-1}$ streptomycin. SH-SY5Y, SUM159PT, and immortalized human ovarian epithelial cells were grown in DMEM/F-12 medium supplemented with 10% FBS, 2 mM Glutamine, 100 units·ml$^{-1}$ penicillin and 100 μg ml$^{-1}$ streptomycin. MCF10A were grown in DMEM/F-12 medium supplemented with 5% FBS, 0.005 mg ml$^{-1}$ bovine insulin, 1 μg ml$^{-1}$ hydrocortisone, 10 mM HEPES, 2 mM Glutamine, 100 units ml$^{-1}$ penicillin and 100 μg ml$^{-1}$ streptomycin. All cell lines were routinely tested to be negative for mycoplasma contamination.

**Site-directed mutagenesis.** Site-directed mutagenesis to generate Src, Cdh1 mutants was performed using the QuikChange XL Site-Directed Mutagenesis Kit (Agilent) according to the manufacturer's instructions.

**Lentiviral and retroviral packaging and infection.** Lentiviral constructs were co-transfected with the pCMV-dR8.91 (Delta 8.9) plasmid containing gag, pol, and rev genes and the VSV-G envelope-expressing plasmid into 293T cells. For packaging retrovirus, retroviral constructs were co-transfected with VSV-G, JK3, and pCMV-

tat into 293T cells. Virus-containing media were collected, filtered before being used for infection[51].

**Immunoblots and immunoprecipitation**. Cells were lysed in EBC buffer (50 mM Tris pH 7.5, 120 mM NaCl, 0.5% NP-40) supplemented with protease inhibitors (Thermo Scientific) and phosphatase inhibitors (Thermo Scientific). To prepare the whole cell lysates, 3× sodium dodecyl sulfate (SDS) sample buffer was directly added to the cell lysates and sonicated before being resolved on SDS-polyacrylamide gel electrophoresis (PAGE) and subsequent immunoblotted with primary antibodies. The protein concentrations of the lysates were measured using the Bio-Rad protein assay reagent on a Bio-Rad Model 680 Microplate Reader. Nuclear/cytoplasmic fractionation was performed as previously described[52]. For immunoprecipitation, 1 mg lysates were incubated with the appropriate agarose-conjugated primary antibody for 3–4 h at 4 °C or with unconjugated antibody (1–2 μg) overnight at 4 °C followed by 1 h incubation with Protein G Sepharose beads (GE Healthcare). Immuno-complexes were washed four times with NETN buffer (20 mM Tris, pH 8.0, 100 mM NaCl, 1 mM EDTA and 0.5% NP-40) before being resolved by SDS-PAGE and immunoblotted with indicated antibodies. The relative band intensities were quantified using ImageJ.

**In vitro binding assays**. The pGEX-4T-1-Cdh1 plasmid was transformed into BL21(DE3) competent cells. The recombinant GST-Cdh1 proteins were expressed by Isopropyl β-D-1-thiogalactopyranoside (IPTG) induction for 18 h at 16 °C. The proteins were purified using Glutathione Sepharose 4B (GE Healthcare) according to the manufacturer's instructions. Agarose-bound GST and GST-Cdh1 proteins were further incubated with cell lysates from 293T cells expressing HA-Src proteins. GST-pull down experiments were also performed by incubating GST-fusion proteins with in vitro transcribed and translated indicated plasmid using TNT Quick Coupled Transcription/Translation System from Promega. The incubation was performed at 4 °C for 3–4 h followed by washing with NETN buffer as described in the "Immunoblots and immunoprecipitation" section above. Samples were resolved by SDS-PAGE and subjected to immunoblot analysis.

**In vitro kinase assays**. Src was immuno-purified from 293T cells transfected with HA-Src constructs. Totally, 0.5 M NaCl was used in the washing buffer to remove Src-associated proteins, the purity of HA-Src proteins was confirmed by SDS-PAGE and Coomassie blue staining (Supplementary Fig. 4b). GST-Cdh1, GST-Cdh1-N (1–174), and GST-CTTN-C (1–323) were expressed in BL21 (DE3) *Escherichia coli* and purified using Glutathione Sepharose 4B (GE Healthcare). His-CDK1 was expressed in BL21 (DE3) *E. coli* and purified using Ni-NTA agarose (Qiagen). Cdh1 and CTTN-C were further cleaved using PreScission Protease (GE Healthcare) following the manufacturer's instructions. Src kinase was incubated with indicated amount of GST-Cdh1, cleaved Cdh1, 0.1 μg His-CDK1, or 0.1 μg CTTN-C in kinase assay buffer (10 mM HEPES pH 7.5, 10 mM MgCl₂, 1 mM dithiothreitol, 0.1 mM ATP). The reaction was initiated by the addition of Src kinase in a volume of 30 μl for 60 min at 30 °C followed by the addition of SDS-PAGE sample buffer to stop the reaction before being resolved by SDS-PAGE.

**Cell synchronization**. MDA-MB-231 and T47D cells were arrested at M phase using 300 nM nocodazole or at the G1/S boundary by treating the cells with 10 mM thymidine treatment twice (double thymidine block). After release back to the cell cycle, cells were collected at the indicated time points for preparing cell lysate and fixation for FACS analysis.

**Flow cytometry**. Cells were trypsinized and re-suspended in 200 μl cold phosphate-buffered saline (PBS), 5 ml of cold 90% ethanol was added for fixation overnight. Prior to the assay, cells were centrifuged for 5 min at 200 × g and resuspended in 0.5 ml PBS with propidium iodide (PI, 50 μg ml⁻¹, Sigma) and RNase A (250 μg ml⁻¹, Roche). After incubating 30 min at 37 °C, cells were transferred into FACS tubes and analyzed using a BD LSR II flow cytometer. The results were analyzed using the FlowJo software.

**Clonogenic survival and soft agar assays**. The clonogenic survival and soft agar assays for MDA-MB-231, SUM159PT, and BT474 cells were performed by culturing the cells in 10% FBS containing DMEM or RPMI1640 before plating into 6-well plate at 1000 cells per well. Two weeks later, cells were stained with crystal violet and the colony numbers were counted.

For soft agar assays, cells (10,000 per well) were seeded in 0.5% low-melting point agarose in DMEM or RPMI1640 with 10% FBS, layered onto 0.8% agarose in DMEM or RPMI1640 with 10% FBS. The plates were kept in the cell culture incubator for 21 days after which colonies >50 μm were counted under a light microscope.

**3D spheroid formation assay**. Breast cancer cells were plated into 96-well Ultra Low Attachment Spheroid Microplate (Corning) (1000 cells per well). Spheroid growth was monitor daily using microscope. The anchorage-independent growth of the cell spheroids were determined using the CellTiter-Glo 3D Cell Viability Assay Kit (Promega).

**Transwell migration assay**. For breast cancer cell migration assays, $1 \times 10^5$ cells were plated in an 8.0 μm 24-well plate chamber insert (Corning) with serum-free medium on the top of the insert and 3T3 conditioned medium containing 10% FBS was added at the bottom of the insert. Analysis was performed on each sample in triplicate. Cells were incubated for 15 h and fixed with 4% paraformaldehyde for 15 min. After washing with PBS, cells on the top of the insert were scraped with a cotton swab. Cells adherent to the bottom were stained with 0.5% crystal violet blue for 15 min and then washed with double-distilled water (ddH₂O). The positive-staining cells were examined under the microscope.

**Matrigel invasion assay**. Breast cancer cell Matrigel invasion assays were performed by seeding the cells onto Transwell inserts coated with Matrigel (BD Biosciences) and allowed to invade for 24 h. Cells were then fixed and stained with phalloidin-AF594 and noninvasive cells removed before fluorescence imaging with an inverted Nikon Eclipse TS100 microscope. To quantify levels of invasion, fixed and stained cells were imaged with a Zeiss confocal microscope (20×) at 0 μm with 0.5-μm image slices taken throughout the distance of invasion.

**Immunofluorescence**. Breast cancer cells were seeded onto coverslips in 6-well plates. Prior to the experiments, cells were fixed with ice-cold methanol for 20 min, followed by incubation with 0.1% Triton X-100 in PBS for 10 min. Cells were pre-blocked with 2% bovine serum albumin/PBS for 45 min, then incubated with primary antibodies overnight at 4 °C and followed by secondary antibodies conjugated with Alexa-Fluor-488 or Alexa-Fluor-594. DAPI was used to stain the nuclei.

**Proximity ligation assay**. PLA was performed using the Duolink PLA kit purchased from Sigma (#DUO92101) following manufacturer's instructions.

**Xenograft tumor growth**. For assaying tumor growth in the xenograft model, 6-week-old female nude mice (NCRNU from Taconic) housed in specific pathogen-free environments were injected subcutaneously with $3.0 \times 10^6$ MDA-MB-231 and BT474 derivatives ($n = 6$ for each group) mixed with serum-free DMEM or RPMI1640 medium. Tumor size was measured every other day with a caliper, and the tumor volume was determined by the formula: $L \times W^2 \times 0.5$, where $L$ is the longest diameter and $W$ is the shortest diameter. Mice were then euthanized and xenograft solid tumors were dissected. All experimental procedures strictly complied with the IACUC guidelines and were approved by the IACUC of the University of South Florida.

**In vivo experimental therapy in SUM159PT xenograft mouse model**. SUM159T tumors were established by subcutaneously injecting $3.0 \times 10^6$ cells in 100 μl serum-free DMEM into 6-week-old female nude mice (NCRNU from Taconic). 21 days after tumor cells were inoculated; animals were pooled and randomly divided into four groups with comparable average tumor size. Laboratory members who measured the mice were blinded to the treatment groups. Mice were grouped into vehicle, dasatinib (5 mg kg⁻¹), trametinib (1 mg kg⁻¹) and the combo treatment (dasatinib 5 mg kg⁻¹ with trametinib 1 mg kg⁻¹). All treatments were administrated by oral gavage once a day until the endpoint. Tumor size was measured every other day with a caliper, and the tumor volume was determined by the formula: $L \times W^2 \times 0.5$, where $L$ is the longest diameter and $W$ is the shortest diameter. Mice were then euthanized and xenograft solid tumors were dissected. All experimental procedures strictly complied with the IACUC guidelines of the University of South Florida.

**Mouse models**. All animal experiments were approved by the Beth Israel Deaconess Medical Center IACUC Committee on Animal Research. $Pten^{+/-}$; $Cdh1^{+/-}$ mice cohorts were generated by crossing $Cdh1^{+/-}$ [12] with $Pten^{+/-}$ [29] mice. Autopsy and histological analysis were performed on cohorts of female mice from 6 to 7 months of age. Mouse tissues were fixed in 4% paraformaldehyde. Normal and tumor tissues were embedded in paraffin, sectioned, and hematoxylin and eosin (H&E) stained for pathological evaluation.

**Drug synergy analysis**. The synergy of combinational drug treatment was determined by CI using the CompuSyn program[53]. CI < 0.8 was considered a significant synergistic effect of double agent treatment.

**Statistical analysis**. All quantitative data were presented as the mean ± SEM (standard error of the mean) or the mean ± SD (standard deviation) as indicated of at least three independent experiments by Student's $t$ test for between-group differences. The $P < 0.05$ was considered statistically significant.

**Reporting summary**. Further information on research design is available in the Nature Research Reporting Summary linked to this article.

## Data availability

Full scans of the gels and blots are available in Supplementary Fig. 10. All relevant data are available from the corresponding author upon reasonable request. All the other data supporting the findings of this study are available within the article and its supplementary information files and from the corresponding author upon reasonable request. A reporting summary for this article is available as a Supplementary Information file.

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

## Acknowledgements

We thank Drs. Philippe Soriano, Akira Imamoto, Andriy Marusyk, Edward Seto, and Xiaohong Zhang for providing reagents, Chao Zhang for technical assistance on Matrigel invasion assays, Florian A. Karreth, Eric K. Lau, Peter A. Forsyth, Heather Han, Brian Czerniecki, Andriy Marusyk, Keiran S. M. Smalley, and Elsa R. Flores for useful discussions, and members of the Wan, Chen, and Pandolfi labs for comments and suggestions. This work was supported in part by the NIH grants (L.W., R00CA183914; P.P.P., R35CA197529), the Florida Breast Cancer Foundation (L.W.), the National Natural Science Foundation of China (S.J., 81873249) and the International Postdoctoral Exchange Fellowship Program (2015) from China Postdoctoral Council (S.J.). This work has been supported in part by the Flow Cytometry and the Analytic Microscopy Core Facilities at the Moffitt Cancer Center, an NCI designated Comprehensive Cancer Center (P30-CA076292).

## Author contributions

T.H. and S.J. performed most of the experiments with assistance from H.Z., Q.Y., M.X., M.L., X.Y., L.W., M.C. and S-J.S. L.W., M.C., S-J.S., A.A.B. and P.P.P. designed the experiments and supervised the study. L.W. wrote the paper with help from T.H., S.J., M.L. and M.C. All authors commented on the paper.

## Additional information

**Competing interests:** The authors declare no competing interests.

