## [Transparent Peer Review File · Nature Communications]

Reviewers' comments:

Reviewer #1, Expertise: kinase signalling and biochemistry (Remarks to the Author):

The authors have recently shown that Cdh1, a substrate adaptor for the APC/C E3 ubiquitin ligase, has a second role as a free protein in inhibiting BRAF-CRAF dimerization, thus reducing RAF kinase activity. Here they report a second non-APC/C target for Cdh1, namely the c-Src tyrosine kinase, showing that Cdh1 binding inhibits c-Src kinase activity, potentially endowing Cdh1 with a tumor suppressor function. They started by showing that pY419 c-Src levels were increased in MCF7, MDA-MB-231 and BT474 human breast cancer cell lines upon shRNA depletion of Cdh1, and that the levels of pY705 STAT3, a c-Src substrate, were increased concomitantly. These effects were reversed by re-expression of full length Cdh1 or the N-terminal domain of Cdh1. However, they did not observe any increase in pMEK/pERK signals, which they had observed in melanoma cell lines, although elevated pY419 c-Src was observed in Cdh1-depleted melanoma cells as well. sgRNA-mediated knockout of Cdh1 in MCF7 cells increased pY419 c-Src, and the level of PDGF-induced pY419 c-Src was also increased in Cdh1^{-/-} MEFs. When synchronized T47D cells were depleted of Cdh1, pY419 c-Src levels were increased throughout the cell cycle. Next, they showed that stably expressed shRNA Cdh1 depletion in MDA-MB-231 cells increased cell viability, and both anchorage independent colony formation and spheroid growth, as well as cell migration and xenograft tumor formation. They also found that Cdh1^{+/-};Pten^{+/-} compound heterozygous mice exhibited increased spontaneous mammary tumor formation. They went on to investigate whether Cdh1 exerts the inhibitory effect on c-Src via a direct interaction. They found that c-Src could associate with both the N-terminal domain and the C-terminal WD40 domain of Cdh1, based on transient expression of protein fragments and in vitro pull down assays with GST-fusion proteins. They identified a putative D-box motif of the type recognized by Cdh1-APC/C near the N-terminus of c-Src, 15RRSL18, and showed that a D-R15A/L18A mutant c-Src did not interact with co-expressed Cdh1. They went on to show that recombinant Cdh1 inhibited WT but not RLAA c-Src autophosphorylation in vitro, and that expression of Cdh1 increased co-precipitation of a WT Src-N fragment with a WT Src-C fragment, and to a greater extent with a Y530F Src-C fragment. Using a c-Src 531EEI mutant, which causes tighter interaction between pY530 and the SH2 domain, they found reduced Cdh1 binding, implying that Cdh1 binds to the open c-Src conformation. Next, they showed that expression of the D1-RLAA mutant c-Src in MDA-MB-231 cells increased colony formation and xenograft tumor growth, consistent with Cdh1 restraining c-Src activity in these cells. In addition, they found that c-Src could phosphorylate Y148 in Cdh1 in vitro, and that c-Src expression stimulated Cdh1 Tyr phosphorylation in vivo. The level of pTyr in Cdh1 in MDA-MB-231 cells was decreased by sgRNA-mediated knockout of c-Src, and by treatment of cells with dasatinib, a Src family inhibitor, which both led to decreased Cdh1/ APC/C association and reduced levels of APC/C targets, such as Cdc6 and Plk1, suggesting that c-Src-mediated Y148 phosphorylation decreases the level of Cdh1/APC/C complexes and thereby decreases ubiquitylation and degradation of its targets, thus acting as a negative feed back mechanism. Finally, they showed that combined use of dasatinib and the PD0325901 MEK inhibitor synergized to decrease MDA-MB-231 cell viability and colony formation, concomitant with decreased levels of APC/C substrates. Expression of a 4D N-terminal phosphomimetic mutant of Cdh1, which exhibited reduced c-Src binding, prevented the dasatinib/PD0325901 treatment induced decrease in Plk1/Cdc6 levels, viability and colony formation, suggesting that ERK-mediated phosphorylation of the Cdh1 N-terminal region reduces c-Src binding and c-Src phosphorylation of Y148.

The possibility that the Cdh1, an APC/C E3 ligase substrate specificity subunit, binds to and inhibits c-Src kinase activity, and that this interaction might play a role in negative control of proliferative signals downstream of c-Src, is certainly interesting, and could explain the proposed tumor suppressor activity attributed to Cdh1. The main concern with the authors conclusion is whether there is enough free Cdh1 protein, not associated with APC/C, in the cytoplasm of the cell available to interact with c-Src to exert the proposed inhibitory effect – indeed, it is unclear from the data presented where Cdh1 and c-Src are associated in the cell. Cdh1 has a strong independent NLS (Zhou et al. JBC 278: 12530, 2003), meaning that most of the endogenous Cdh1 protein

population in the cell will be nuclear, regardless of whether it is associated with APC/C in the nucleus. Although there have been reports of nuclear c-Src, most of the evidence is consistent with the majority of c-Src being associated with cytoplasmic membranes, including the plasma membrane. For this reason, the authors need to determine the relative absolute protein levels of Cdh1 and c-Src, the fraction of Cdh1 that is free, the localizations of Cdh1 and c-Src in the cells they analyzed, and, most importantly what fraction of endogenous c-Src is associated with Cdh1 (and whether this changes during the cell cycle). PLA analysis could be used to demonstrate close association of Cdh1 and c-Src in the cell, and the localization of c-Src/Cdh1 complexes. What fraction of the total c-Src population is occupied with bound Cdh1 was not assessed, and yet this is essential to know since the inhibitory effect is proposed to be directly due to Cdh1 binding, and therefore only c-Src molecules associated with Cdh1 will be inhibited. As it stands, it remains possible that the increased activity of endogenous c-Src upon Cdh1 knockdown/knockout is in part a result of an indirect effect due to reduced APC/C-Cdh1 ligase activity

Points: 1. The data demonstrating Cdh1 associates with c-Src via dual Cdh1-WD40/c-Src SH4 and N-Cdh1/c-Src catalytic domain interactions are reasonably convincing, but rely on overexpression studies of protein fragments, and there is only one experiment showing association between endogenous Cdh1 and c-Src (and this was only done one way round).

2. The D1-RAAL motif in the c-Src SH4 domain contains the S17 PKA phosphorylation site, and phosphorylation of this residue seems likely to negatively affect c-Src interaction with Cdh1; this could be tested by stimulating PKA activity. In this regard, it is unclear whether the D1-RLAA mutant form of c-Src is properly localized to cytoplasmic membranes,

3. It is unclear whether Cdh1 binding affects the stoichiometry of c-Src pY530, which is a key negative regulatory process - the author show blots for pY530, but these are not quantified. If Cdh1 binding to c-Src exposes pY530 this might lead to increased pY530 dephosphorylation.

4. There is no analysis of how the Cdh1 N-terminal region interacts with the c-Src catalytic domain or how this interaction would inhibit c-Src substrate phosphorylation. Is this via the front side or backside?

5. Figure 1A/B: Although the authors confirmed the effects of Cdh1 depletion on c-Src pY419 using Cas9/sgRNA knockout, they should really show that re-expression of an shRNA-resistant form of Cdh1 reduces pY419 levels.

6. Figure 1D/E: The authors generated a pool of Cdh1 knockout MCF7 cells, but it is clear that there are some cells that retain Cdh1. Can the authors generate a clone of Cdh1^{-/-} MCF7 cells? Cdh1 needs to be re-expressed in the Cdh1^{-/-} MEFs (panel E) to show that the effects on PDGF-induced pY419 c-Src are reversed. Is the level of PDGFR pTyr increased in the Cdh1^{-/-} MEFs?

7. Figure 2J-L: The authors conclude that the increased spontaneous mammary tumor formation observed in Cdh1^{+/-}:Pten^{+/-} compound heterozygous mice was due to activation of c-Src, because levels of pY419 c-Src were elevated, but provide no direct evidence for this.

8. Figure 3/S3: The c-Src 15RRSL18 D-box motif is very close to the N-terminal myristoyl group, which anchors c-Src to membranes, and this orientation may impose some topological constraints for Cdh1 binding. Conversely Cdh1 binding might cause c-Src dissociation from the membrane, since the basic residues in this reading are thought to serve as anchors via interaction with phospholipid head groups. The GST-Src protein used for pull downs will lack this N-terminal lipid modification and therefore may not recapitulate what happens with native c-Src.

9. Figure 4A: The authors have used an in vitro autophosphorylation assay to demonstrate an inhibitory effect of GST-Cdh1 on c-Src kinase activity, which is not really satisfactory. Since we are

not shown the level of pY419 in the starting HA-Src sample, it is unclear how much this signal was increased during the assay. Moreover GST fusion proteins are dimers, making it impossible to determine the true efficiency of inhibition due to avidity effects, and, in any case, the relative levels of HA-Src and GST-Cdh1 proteins used in the assay are not provided. An authentic c-Src substrate kinase assay needs to be carried out using monomeric Cdh1 - in Figure S4A the authors used recombinant CDK1 as a substrate, but CDK1 is not generally regarded as a bona fide c-Src substrate. Also, it is unclear why there are so many smaller GST antibody-positive bands in lanes 3 and 4. Finally, did the authors check if the N-terminal domain of Cdh1 alone inhibits c-Src in vitro.

10. Figure 4D: While it is true that co-expression of HA-Cdh1 increased the level of HA-Src-C that was brought down by Flag-Src-N (presumably this fragment lacks the N-terminal myristoyl group), all this demonstrates is that Cdh1 can bind both halves of Src and not that the association between the two halves of Src was increased by Cdh1 as the authors claim. The increased binding of the Y530F mutant Src-C might be due to the open configuration of the Y530F mutant, but a significant fraction of the N/C domain interaction energy comes from the SH2 domain-pY430 interaction, which would be missing in the case of Y530F mutant. Moreover, they did not use an SH2 domain (or an SH3 domain) mutant to determine whether loss of pY530/SH2 interaction is important.

11. Figure 4J: The same criticisms apply to this experiment as to Figure 4A.

12. Figure 5: The authors attribute the increased tumorigenicity of the c-Src D1-RLAA expressing MDA-MB-231 cells to the fact that the mutant c-Src is no longer subject to inhibition by Cdh1, but they did not test the consequences of expressing c-Src D1-RLAA MDA-MB-231 in cells depleted of Cdh1, which is a key control.

13. Figure 6: Based on the known structures of Cdh1, is Y148 actually accessible for phosphorylation. It should be noted that Y148 does not lie in a Src kinase consensus sequence, i.e. it lacks any preferred acidic residues on either side. Moreover, while c-Src expression was shown to increase Cdh1 pTyr levels upon co-expression in cells, the authors did not demonstrate that this increase occurred at Y148, and more importantly they did not determine what fraction of the Cdh1 population was phosphorylated at Y148, or whether or not pY148 was associated with APC/C. In addition, no measurements of APC/C E3 ligase activity with Y148F Cdh1 versus pY148 Cdh1 bound were carried out to establish that Y148 phosphorylation reduces its E3 ligase activity.

14. Figure 6: Dasatinib is a dirty kinase inhibitor, inhibiting Tyr kinases other than c-Src, such as Abl /Arg, other SFKs, as well as several Ser/Thr kinases. In consequence, one cannot interpret the results of experiments using dasatinib as being due to c-Src inhibition. It would be better to use sarcatinib/AZD0530, which is more selective but by no means c-Src selective.

15. Figure 7: The authors provide no evidence that the expression of the 4D phosphomimetic mutant Cdh1 decreased Y148 phosphorylation. It is unclear what fraction of Cdh1 is phosphorylated at Y148, and whether this represents a significant fraction of Cdh1, or whether the pY148 Cdh1 molecules are selectively released from APC/C. The authors need to generate pY148 specific antibodies to characterize Y148 phosphorylation.

16. Figures 6 and 7: In general, many additional experiments on ERK and c-Src-mediated phosphorylation of Cdh1 are needed to provide stronger evidence for the authors hypothesis that c-Src phosphorylation of Y148 in Cdh1 is an important regulatory event that inhibits Cdh1/APC/C activity as a negative feedback mechanism.

Reviewer #2, Expertise: cancer biology
(Remarks to the Author):

The manuscript by Han et al. reports a new interaction between the APC/C coactivator Cdh1 and the kinase Src in breast cancer cells. Silencing of Cdh1 results in Src hyperphosphorylation (Y419) and over-activation in an APC/C-independent manner, resulting in increased proliferation and tumorigenic properties of breast cancer cells. Biochemical studies show that Cdh1 and Src can bind and inhibit each other using different mechanisms. On one hand, Cdh1 locks Src in a closed conformation inhibiting its ability to be activated. On the other, Src is able to phosphorylate Cdh1, thus preventing its binding to APC/C. The authors finally proposed that inhibiting Src kinase activity may have therapeutic activity in triple negative breast cancer cells, at least partially by re-activating the tumor suppressor properties of Cdh1.

In general, the information provided in the manuscript is of high technical quality and the conclusions are solid, novel and supported by the data. The current version of the manuscript contains a significant amount of information, well presented and discussed, and it could be published after revision of a couple of questions that are not completely clear or have not been analyzed in detail.

Major points

1. Data in Figure 2 shows that ablation of Cdh1 results in higher oncogenic properties (proliferation, migration, etc.) in the presence of higher phosphorylation of Src. Similar correlation is found after crossing Cdh1 and Pten het mice. From these studies the authors conclude that “these findings demonstrate that Cdh1 functions as a tumor suppressor in vivo in part by suppressing the activation of the Src oncogenic pathway” (lines 159-160). This conclusion requires very easy assays in which (at least some of) the oncogenic properties of Cdh1 are prevented upon silencing (or inhibition or expression of kinase-dead mutants, etc) or Src (at least in vitro).
2. In Figure 1k, the kinetics of Plk1 is similar to Cdh1, but it should be the opposite as Plk1 is a Cdh1 target for proteasome-dependent degradation. This is unusual as they should follow a pattern similar to P-Y419? As this experiment describes the levels of these proteins during checkpoint recovery (see minor points below), a description of the exact stage of the cell cycle (either by FACS or expression of E, A, B cyclins) would help to understand this discrepancy.
3. lines 287-290. Src deficiency results in decrease in the levels of various APCC-Cdh1 substrates. However, this may be the effect of cell cycle arrest upon Src inhibition. Cell cycle profiles or protein levels of cell cycle regulators that are not APC/C-targets should be analyzed here. This also applies to the effect of Src inhibitors (lines 296-297, 328-330).

Minor points

Line 126. Cdh1 peaks in G2/M. The expression of Cdh1 shown in this panel does not correspond to normal levels during a cell cycle but levels after checkpoint recovery (samples recover from thymidine block). The authors should modify the text accordingly.

Lines 334-336. Epithelial tumors were described in Cdh1 +/- mice, not in Sox2-Cre conditional mice.

Reviewer #3, Expertise: Breast cancer, signalling (Remarks to the Author):

Cdh1 functions as a coactivator of the Anaphase Promoting Complex (APC/C) during late mitosis and early G1 phase negatively regulating the stability of several oncogenic substrates as PIK1, Cdc6, Skp2 or Cyclin A. However, Cdh1 protein has also been reported to have APC/C-independent functions. In this work, Han et al. report Cdh1 as a Src-interacting protein that negatively regulates its oncogenic activity in breast cancer cells independently of APC/C. Cdh1 reinforces Src inactive closed conformation perturbing its kinase activity and its depletion

promotes tumor growth in diverse breast cancer cell lines and Cdh1 +/- Pten +/- mouse models. Moreover, the authors also found that Src kinase phosphorylates Cdh1 and disrupts its interaction with the APC/C complex, thus impairing the tumor suppressor role of APC/Cdh1. Finally, pharmacological inhibition of Src and MEK, which also phosphorylates the N-terminus part of Cdh1, synergistically potentiates the suppression of cell viability in TNBC cells suggesting this drug combination as a more efficient treatment of breast cancer than monotherapy.

In this manuscript, Han and colleagues describe a novel interplay among Cdh1, Src and APC/C and its contribution in breast tumor growth. The work is, in general, experimentally well supported. However, some questions must be addressed prior to publication.

1) In the Figure 1 D. Authors show a Western-Blot performed in Cdh1 CRISPR-Cas9-mediated KO cells. However, a slight band is still present when Cdh1 is analyzed. Could be explained because the authors are not working with pure KO pools of cells?

2) In the figure 1k it is not clear why PIK1, which is targeted by APC/Cdh1, is upregulated when Cdh1 is high along cell cycle.

3) In this manuscript authors demonstrate a complex interplay among Cdh1, Src and APC/C. Cdh1 downregulation activates downstream oncogenic targets of Src (e.g. p-YAP). In fact, according to the pro-metastatic role of Src in cancer, Cdh1 depleted cells tend to increase its migration abilities (Figure 2g). On the other hand, APC/Cdh1 substrates are also negatively regulated by Cdh1 (e.g. PIK1, Skp2, Cdc6). However, it is unclear to what extent Cdh1 action is mediated by Src or alternatively by the APC/Cdh1 complex in breast cancer. Results from Figure 7 suggest that Cdh1 action is ultimately mediated through APC/C complex since Src/MEK inhibitors have not a deep effect when a phosphomimetic 4D-Cdh1 mutant is used. To further complete the study, additional experiments can be performed:

- Since Cdh1 deletion delays mitotic exit and increases genomic instability and chromosomal aberrations (Garcia-Higuera et al. 2008) due to defective APC functionality, analysis of genomic defects or study of abnormal mitosis in Cdh1 breast cancer cells is relevant.
- Downregulation of Src in shCdh1 cells (Figure 2) would help to discern whether the mechanism of action of Cdh1 is mainly mediated by Src downstream pathways or the APC/Cdh1 complex.
- Alternatively, overexpression/downregulation of Cdh1 in D1-RLAA Src mutant (abolishing the interaction between Cdh1 and Src) also is a sound strategy to test the relevance of Cdh1/Src axis in breast tumorigenesis.

4) Although it seems that combinatorial treatment might be potentially promising in BCa treatment, *in vitro* studies presented in the figure 7 could be significantly improved by treating mice harboring some of the breast cancer cell lines used in the work with monotherapy or combined therapy (Src/MEK inhibitors).

Minor points:

1) Size (kDa) of the bands should be shown in the WB figures to facilitate the understanding.

2) In the line 138 mention: "having demonstrated that Cdh1 suppresses Src function in melanoma cells..." should be replaced by "breast cancer cells".

3) In the lines 298/299 is mentioned that MEK has been shown "to phosphorylate the N-terminus of Cdh1 and to activate APC/Cdh1 in melanoma cells". However, according to the bibliography (Wan et al. 2017) MEK/ERK inhibits APC/Cdh1 activity.

Point-by-Point Responses to the Reviewers' Critiques (NCOMMS-18-18370)

Reviewer #1, Expertise: kinase signalling and biochemistry (Remarks to the Author):

The authors have recently shown that Cdh1, a substrate adaptor for the APC/C E3 ubiquitin ligase, has a second role as a free protein in inhibiting BRAF-CRAF dimerization, thus reducing RAF kinase activity. Here they report a second non-APC/C target for Cdh1, namely the c-Src tyrosine kinase, showing that Cdh1 binding inhibits c-Src kinase activity, potentially endowing Cdh1 with a tumor suppressor function.

They started by showing that pY419 c-Src levels were increased in MCF7, MDA-MB-231 and BT474 human breast cancer cell lines upon shRNA depletion of Cdh1, and that the levels of pY705 STAT3, a c-Src substrate, were increased concomitantly. These effects were reversed by re-expression of full length Cdh1 or the N-terminal domain of Cdh1. However, they did not observe any increase in pMEK/pERK signals, which they had observed in melanoma cell lines, although elevated pY419 c-Src was observed in Cdh1-depleted melanoma cells as well. sgRNA-mediated knockout of Cdh1 in MCF7 cells increased pY419 c-Src, and the level of PDGF-induced pY419 c-Src was also increased in Cdh1-/- MEFs. When synchronized T47D cells were depleted of Cdh1, pY419 c-Src levels were increased throughout the cell cycle. Next, they showed that stably expressed shRNA Cdh1 depletion in MDA-MB-231 cells increased cell viability, and both anchorage independent colony formation and spheroid growth, as well as cell migration and xenograft tumor formation. They also found that Cdh1+/-:Pten+/- compound heterozygous mice exhibited increased spontaneous mammary tumor formation.

They went on to investigate whether Cdh1 exerts the inhibitory effect on c-Src via a direct interaction. They found that c-Src could associate with both the N-terminal domain and the C-terminal WD40 domain of Cdh1, based on transient expression of protein fragments and in vitro pull down assays with GST-fusion proteins. They identified a putative D-box motif of the type recognized by Cdh1-APC/C near the N-terminus of c-Src, 15RRSL18, and showed that a D-R15A/L18A mutant c-Src did not interact with co-expressed Cdh1. They went on to show that recombinant Cdh1 inhibited WT but not RLAA c-Src autophosphorylation in vitro, and that expression of Cdh1 increased co-precipitation of a WT Src-N fragment with a WT Src-C fragment, and to a greater extent with a Y530F Src-C fragment. Using a c-Src 531EEI mutant, which causes tighter interaction between pY530 and the SH2 domain, they found reduced Cdh1 binding, implying that Cdh1 binds to the open c-Src conformation. Next, they showed that expression of the DI-RLAA mutant c-Src in MDA-MB-231 cells increased colony formation and xenograft tumor growth, consistent with Cdh1 restraining c-Src activity in these cells.

In addition, they found that c-Src could phosphorylate Y148 in Cdh1 in vitro, and that c-Src expression stimulated Cdh1 Tyr phosphorylation in vivo. The level of pTyr in Cdh1 in MDA-MB-231 cells was decreased by sgRNA-mediated knockout of c-Src, and by treatment of cells with dasatinib, a Src family inhibitor, which both led to decreased Cdh1/ APC/C association and reduced levels of APC/C targets, such as Cdc6 and Plk1, suggesting that c-Src-mediated Y148 phosphorylation decreases the level of Cdh1/APC/C complexes and thereby decreases ubiquitylation and degradation of its targets, thus acting as a negative feed back mechanism. Finally, they showed that combined use of dasatinib and the PD0325901 MEK inhibitor synergized to decrease MDA-MB-231 cell viability and colony formation, concomitant with decreased levels of APC/C substrates. Expression of a 4D N-terminal phosphomimetic mutant of Cdh1, which exhibited reduced c-Src binding, prevented the dasatinib/PD0325901 treatment induced decrease in Plk1/Cdc6 levels, viability and colony formation, suggesting that ERK-mediated phosphorylation of the Cdh1 N-terminal region reduces c-Src binding and c-Src phosphorylation of Y148.

The possibility that the Cdh1, an APC/C E3 ligase substrate specificity subunit, binds to and inhibits c-Src kinase activity, and that this interaction might play a role in negative control of proliferative signals downstream of c-Src, is certainly interesting and could explain the proposed tumor suppressor activity attributed to Cdh1. The main concern with the authors conclusion is whether there is enough free Cdh1 protein, not associated with APC/C, in the cytoplasm of the cell available to interact with c-Src to exert the proposed inhibitory effect “ indeed, it is unclear from the data presented where Cdh1 and c-Src are associated in the cell. Cdh1 has a strong independent NLS (Zhou et al. JBC 278:12530, 2003), meaning that most of the endogenous Cdh1 protein population in the cell will be nuclear, regardless of whether it is associated with APC/C in the nucleus.

Response: We thank the reviewer for recognizing the novelty and the potential impact of this study, as well as for the careful examination of our manuscript. We fully agree with the reviewer that it is very important to provide further evidence to demonstrate that a significant amount of APC-free Cdh1 exists in the cytoplasm where it could interact with c-Src and suppress c-Src function in our experimental settings. Enlightened by the constructive comments from the reviewer, we have obtained the following results to support our major conclusion:

- 1) Using both immunofluorescence (IF) and cytoplasmic/nuclear fractionation approaches, we found that in breast cancer cells, Cdh1 is mainly localized in the cytoplasm rather than in the nucleus (**Fig. 3e and Supplementary Fig. 3e**). In contrast, in non-transformed cells, such as human fibroblasts and normal mammary epithelial cell line MCF10A, more Cdh1 was found in the nuclear fraction (**Fig. 3e**) as the reviewer pointed out. Cdh1 and c-Src were also found co-localized in the cytoplasm using IF and PLA (Proximity Ligation Assay) approaches (**Fig. 3f and Supplementary Fig. 3e**).
- 2) Although Cdh1 was mainly found in the cytoplasm in breast cancer cells, core APC components APC6 and APC8 were exclusively observed in the nucleus (**Fig. 3e**), supporting the notion that in tumor cells, a significant amount of Cdh1 is APC-free and resides in the cytoplasm.
- 3) Examination of Cdh1 protein abundance across a panel of normal cells and breast tumor cells revealed that compared to normal cell lines, Cdh1 protein abundance is higher in breast tumor cells (**Supplementary Fig. 3q**). Interestingly, although a higher level of Cdh1 was found in tumor cells, known APC^{Cdh1} ubiquitin substrates including Plk1 and Cdc6 were also highly expressed (**Supplementary Fig. 3q**). These findings indicate that the excess amount of Cdh1 found in breast tumor cells is likely inactive due to inefficient APC binding and cytoplasmic retention.
- 4) Previous reports including ours found that serine and threonine phosphorylation of Cdh1 at multiple sites disrupts the binding between Cdh1 N-terminus and the APC core complex, which led to a compromised APC^{Cdh1} ubiquitin E3 ligase activity (Keck JM et al. (2007). *Journal of Cell Biology*. 178, 371–385; Fukushima H et al. (2013). *Cell Rep*. 4, 803–816; Lau AW et al. (2013). *Cell Res*. 23, 947–961; Wan L et al. (2017). *Cancer Discovery*. 7, 424–441). In this manuscript, we added the Y148 tyrosine phosphorylation to this inhibitory phosphorylation list (**Fig. 6**). It is also worth noting that Cdh1 N-terminus has been shown acetylated at K69 and K159. Similar to Cdh1 N-terminal phosphorylation, acetylation of Cdh1 disrupts Cdh1-APC interaction thus inhibiting APC^{Cdh1} activity (Kim HS et al. (2011). *Cancer Cell*. 20, 487–499). All these findings coherently demonstrate that post-translational modifications of Cdh1 at its N-terminus are important mechanisms through which the cells modulate APC^{Cdh1} activity in different cellular contexts. Therefore it is tempting to postulate that Cdh1 N-terminal phosphorylation might also govern Cdh1 subcellular localization.
- 5) In support of this notion, we found that in contrast to WT-, 4D(S40D/T121D/S151D/S163D)-, or Y148E-Cdh1, 4A(S40A/T121A/S151A/S163A)- and Y148F-Cdh1, which exhibited a stronger

- interaction with the APC core complex (**Fig. 7e**), were predominantly localized to the nucleus (**Fig. 7a-b**). This observation also echoes the previous report from the Jin group using 4A- and 4D-Cdh1-expressed HeLa cells (Zhou Y et al. (2003). *Journal of Biological Chemistry*. 278, 12530–12536).
- 6) We agree with the reviewer that Cdh1 contains a strong independent NLS sequence which directs Cdh1 to the nucleus to exert its function with the APC core complex (Zhou Y et al. (2003). *Journal of Biological Chemistry*. 278, 12530–12536; Zhou Y et al. (2003). *The Biochemical Journal*. 374, 349–358). Our results illustrated above, however, painted a rather different picture of the subcellular localization of Cdh1 in breast tumor cells, and in a number of other tumor cell lines we examined. To solve this discrepancy, we sought out to unravel the molecular mechanism that leads to Cdh1 cytoplasmic retention. Using the NetNES 1.1 Server (<http://www.cbs.dtu.dk/services/NetNES/>), we found that Cdh1 contains a leucine-rich nuclear export signal at L184-L192, right next to its NLS signal (K156-K177: Zhou Y et al. (2003). *Journal of Biological Chemistry*. 278, 12530–12536) (**Supplementary Fig. 7a-b**). The Cdh1 NES sequence (¹⁸⁴LQDDFYLN¹⁹², **Supplementary Fig. 7a-b**) resembles the canonical NES consensus quite well (ϕ 1-X(2-3)- ϕ 2-X(2-3)- ϕ 3-X- ϕ 4, ϕ denotes a hydrophobic residue including L, V, I, F and M, while X can be any amino acid but preferentially is charged or polar, Cautain B et al. (2015). *FEBS J*. 282, 445–462).
 - 7) Deletion of the NES signal resulted in a significant enrichment of nuclear Cdh1 in breast cancer cells (**Fig. 7c**). More importantly, deletion of the NES signal in 4D- and Y148E-Cdh1 mutants also led to dramatic nuclear retention of Cdh1 (**Fig. 7c**). These results suggest that the NES signal identified in Cdh1 serves to mediate the export of Cdh1 to the cytoplasm.
 - 8) Exportin proteins facilitate the translocation of nuclear proteins to the cytoplasm. Among all mammalian exportins, CRM1/XPO1 is the major exportin that recognizes leucine-rich NES motifs (Kutay U & Güttinger S (2005). *Trends Cell Biol*. 15, 121–124). Notably, we found that Cdh1 interacted with CRM1 at the endogenous level (**Fig. 3d and Supplementary Fig. 3c**). Intriguingly, compared to WT-Cdh1, deletion of the NES signaling abolished the binding between Cdh1 and CRM1, showing the NES we identified as the sequence to mediate Cdh1-CRM1 binding (**Fig. 7d**).
 - 9) Moreover, depletion of *CRM1* with shRNA or using a CRM1 inhibitor Leptomycin B (Kudo N et al. (1998). *Exp. Cell Res*. 242, 540–547) significantly enhanced Cdh1 nuclear localization in breast cancer cells (**Fig. 7f and Supplementary Fig. 7f**).
 - 10) Comparing the binding between CRM1 and a panel of Cdh1 mutants revealed that 4D- and Y148E-Cdh1 displayed a stronger binding with CRM1 whereas 4A- and Y148F-Cdh1 showed reduced interaction (**Fig. 7d**).
 - 11) Furthermore, depletion of *CRM1* in 4D- and Y148E-Cdh1-expressing cells led to increased nuclear localization of Cdh1 (**Fig. 7g**). These finding thus indicate that phosphorylation of Cdh1 at its N terminus might facilitate Cdh1/CRM1 interaction.
 - 12) Since these phosphorylation sites are not found within the NES motif, we reasoned that the increased binding between hyper-phosphorylated Cdh1 and CRM1 may be due to decreased binding between Cdh1 and the APC core complex. In concert with this notion, compared to 4A- and Y148F-Cdh1, 4D- and Y148E-Cdh1 exhibited reduced binding with APC core subunit APC6 while showed enhanced binding with CRM1 (**Fig. 7e**).
 - 13) Endogenous co-IP experiments revealed that Cdh1 serine/threonine phosphorylation, Y148 phosphorylation, acetylation, as well as Cdh1/Src interaction were only observed in the cytoplasmic Cdh1 population, whereas Cdh1/APC6 interaction was only found in the nucleus (**Fig. 7h**).

14) In support of a role of CRM1 in silencing APC^{Cdh1}, a large volume of studies have demonstrated that CRM1 is responsible for transporting and thus inhibiting a number of tumor suppressors including p53, APC (Adenomatous polyposis coli), p21^{CIP1} and p27^{KIP1} (Turner JG & Sullivan DM (2008). *Curr. Med. Chem.* 15, 2648–2655). Further, a recent study found that CRM1 is overexpressed in invasive breast carcinoma and predicts poor prognosis (Yue L et al. (2018). *Oncol Lett.* 15, 7515–7522) and a recent report unveiled an anti-TNBC effect of a CRM1 inhibitor Selinexor/KPY-330 (Arango NP et al. (2017). *Breast Cancer Res.* 19, 93).

These newly obtained results together demonstrate that Cdh1 is mainly localized in the cytoplasm in breast cancer cells due to the increased N-terminal phosphorylation by a number of oncogenic kinases including CDK2, CDK4, ERK, and c-Src. Phosphorylation of Cdh1 N-terminus not only blocks its homing to the APC core complex but also facilitates its binding with CRM1 and subsequent nuclear export (**Supplementary Fig. 7c**). We thank the reviewer for this insightful critique which guided us to this important discovery. We also acknowledge that there will be further questions raised regarding the cross-talks among Cdh1 N-terminal modifications, the binding between Cdh1 and the APC complex, and the Cdh1 subcellular localization, which in our opinion, will be more suitable for a separate future manuscript. We hope the reviewer would concur with us that we have obtained a substantial amount of results to support our major conclusion and to directly address the concerns raised by the reviewer.

Although there have been reports of nuclear c-Src, most of the evidence is consistent with the majority of c-Src being associated with cytoplasmic membranes, including the plasma membrane. For this reason, the authors need to determine the relative absolute protein levels of Cdh1 and c-Src, the fraction of Cdh1 that is free, the localizations of Cdh1 and c-Src in the cells they analyzed, and, most importantly what fraction of endogenous c-Src is associated with Cdh1 (and whether this changes during the cell cycle). PLA analysis could be used to demonstrate close association of Cdh1 and c-Src in the cell, and the localization of c-Src/Cdh1 complexes. What fraction of the total c-Src population is occupied with bound Cdh1 was not assessed, and yet this is essential to know since the inhibitory effect is proposed to be directly due to Cdh1 binding, and therefore only c-Src molecules associated with Cdh1 will be inhibited. As it stands, it remains possible that the increased activity of endogenous c-Src upon Cdh1 knockdown/knockout is in part a result of an indirect effect due to reduced APC/C-Cdh1 ligase activity.

Response: We agree with the reviewer that c-Src is predominantly localized in the cytoplasm, which is supported by our IF and fractionation experiments (**Fig. 3e** and **Supplementary Fig. 3e**). Our newly obtained results as described above demonstrate that the majority of Cdh1 (~80%) is also localized in the cytoplasm in breast cancer cells (**Fig. 3e**). Importantly, in contrast to the cytoplasmic localization of Cdh1, core APC components including APC6 and APC8 exhibited an exclusive nuclear localization (**Fig. 3e**), indicating that the cytoplasmic population of Cdh1 is largely APC-free.

As kindly instructed by the reviewer, we compared the relative protein abundance of Cdh1 and c-Src in cells. We utilized ectopically expressed GST-Cdh1 and GST-c-Src in 293T cells as the reference to compare the ratio between endogenous Cdh1 and endogenous c-Src. As demonstrated in **Supplementary Fig. 4l**, we found that Cdh1:c-Src ratio in 293T cells is about 4:1. Since both Cdh1 and c-Src display a relatively similar expression among 293T and a panel of breast cancer cell lines (**Supplementary Fig. 4m**), we conclude that in breast cancer cells used in our experiments the ratios of Cdh1:c-Src are approximately ranging from 3:1 to 25:1 (**Supplementary Fig. 4m**). Given that about 80% Cdh1 is localized in the cytoplasm and presumably APC-free, it is plausible to postulate that in breast cancer cells, there is abundant Cdh1 resides in the cytoplasm that could potentially interact with, and inhibit c-Src.

Following the reviewer's kind instruction, we compared the binding of endogenous Cdh1 and c-Src using synchronized cells followed by fractionation derived from different cell cycle phases. The cell lysates from 6h, 9h, and 18h after double thymidine block and release represent S, G2 and G1 populations, respectively (**Supplementary Fig. 3g**). Notably, Src displayed a stronger interaction with Cdh1 in the cytosol fractions compared to the nuclear fractions (**Supplementary Fig. 3f**). Cdh1/Src interaction peaked in the G2 phase, which is consistent with the results that p-Y419-Src is lower in G2/M phases (**Fig. 1k and Supplementary Fig. 1n**). On the other hand, Cdh1 exhibited a strong binding with APC6 in the G1 phase nuclear fraction (**Supplementary Fig. 3f**), supporting an active APC^{Cdh1} during G1 cell cycle phase.

Moreover, as suggested by the reviewer, we performed PLA analysis using both Cdh1 and c-Src antibodies in MCF7 and MDA-MB-231 cells. As shown in **Fig. 3f**, consistent with our fractionation and IF results (**Fig. 3e and Supplementary Fig. 3e**), Cdh1 mainly exhibited close proximity to c-Src in the cytoplasm. Moreover, we observed a strong interaction between endogenous Cdh1 and endogenous c-Src in MCF7 and MDA-MB-231 cells using both anti-Cdh1 and anti-c-Src immunoprecipitates (**Fig. 3d and Supplementary Fig. 3b-d**).

We also agree with the reviewer that it is possible that in a certain context, Cdh1 might affect c-Src activity via an indirect mechanism through the APC^{Cdh1} E3 ligase activity. However, since depletion of core APC subunits APC10 and Cdc27 in MDA-MB-231 cells failed to alter c-Src activity (**Fig. 1g-h and Supplementary Fig. 1h-i**), the suppression of c-Src function by Cdh1 is likely through a direct inhibitory mechanism at least in the breast cancer cell settings.

Points:

1. The data demonstrating Cdh1 associates with c-Src via dual Cdh1-WD40/c-Src SH4 and N-Cdh1/c-Src catalytic domain interactions are reasonably convincing, but rely on overexpression studies of protein fragments, and there is only one experiment showing association between endogenous Cdh1 and c-Src (and this was only done one way round).

Response: As kindly instructed by the reviewer, in the revised manuscript, we performed anti-Cdh1 and anti-c-Src co-IP using both MCF7 and MDA-MB-231 lysates. As shown in **Fig. 3d and Supplementary Fig. 3b-d**, co-IP results from both directions in two breast cancer cell lines clearly demonstrated the binding between Cdh1 and c-Src. Moreover, we also performed PLA analysis using MCF7 and MDA-MB-231 cells, as shown in **Fig. 3f**, we observed a strong PLA signal predominantly in the cytoplasm, which is consistent with our observations from IF experiments (**Supplementary Fig. 3e**).

2. The D1-RAAL motif in the c-Src SH4 domain contains the S17 PKA phosphorylation site, and phosphorylation of this residue seems likely to negatively affect c-Src interaction with Cdh1; this could be tested by stimulating PKA activity. In this regard, it is unclear whether the D1-RLAA mutant form of c-Src is properly localized to cytoplasmic membrane.

Response: We thank the reviewer for raising this important question regarding the S17 PKA site within the D1 motif which mediates c-Src-Cdh1 interaction. Compared to WT-c-Src, we found that neither the phosphorylation mimetic S17D mutation nor the phosphorylation-deficient S17A mutation affected Cdh1-c-Src interaction (**Supplementary Fig. 3k**), indicating that S17 phosphorylation by PKA (*Schmitt JM & Stork PJS (2002). Mol. Cell. 9, 85–94*) might not be able to disrupt Cdh1-c-Src binding. Furthermore, when breast cancer cells were treated with PKA agonists Forskolin and IBMX (3-isobutyl-1-methylxanthine), although a strong increase of pS133-CREB was observed, pS17-c-Src and pY419-c-

Src remained unchanged (**Supplementary Fig. 3l-n**). On the contrary, pS17-c-Src could be induced by PKA agonists in HCT116 colon cancer cell line and A375 melanoma cell line (**Supplementary Fig. 3o-p**). These results suggest that PKA-mediated S17 c-Src phosphorylation might not be a major event in breast cancer cells.

Following the instruction from the reviewer, we also examined the localization of D1-RLAA-c-Src using MDA-MB-231 cells stably expressing WT-c-Src and D1-RLAA-c-Src close to the endogenous level (**Fig. 5a**). As shown in **Supplementary Fig. 5j**, compared to WT-c-Src-expressing MDA-MB-231 cells, D1-RLAA-c-Src displayed a slight increase of plasma membrane localization, consistent with increased activity of D1-RLAA-c-Src.

3. It is unclear whether Cdh1 binding affects the stoichiometry of c-Src pY530, which is a key negative regulatory process - the author show blots for pY530, but these are not quantified. If Cdh1 binding to c-Src exposes pY530 this might lead to increased pY530 dephosphorylation.

Response: As kindly suggested, we have included shorter exposures for pY530-c-Src blots in the revised **Fig. 1a-c, 1d**, and **Supplementary Fig. 1a-c, 1e-f**. Additionally, quantified band intensities for both pY530-c-Src and pY419-c-Src were marked beneath the blots. As the reviewer could find out, depletion of Cdh1 did not significantly influence c-Src Y530 phosphorylation, indicating that Cdh1 may not be able to control c-Src Y530 phosphorylation at least in our experimental settings.

Y530 phosphorylation of c-Src promotes intracellular interaction between the *N*-terminal SH2 domain and the *C*-tail pY530 site, a mechanism that inactivates c-Src (Thomas SM & Brugge JS (1997). *Annu. Rev. Cell Dev. Biol.* 13, 513–609). Our results suggest that Cdh1 binds c-Src in a head-to-toe fashion to inhibit c-Src function (**Fig. 3-4**). Notably, Cdh1 reinforced the binding between *N*-terminal and *C*-terminal domains of c-Src regardless of Y530 phosphorylation status (**Fig. 4i**), suggesting that Cdh1/c-Src interaction is not controlled by c-Src Y530 phosphorylation. Furthermore, although a SH2-deficient R178A-c-Src-*N* (Bibbins KB et al. (1993). *Molecular and Cellular Biology.* 13, 7278–7287) failed to bind c-Src-*C* or Y530F-c-Src-*C*, the addition of Cdh1 promoted the interaction between both parts of the c-Src protein (**Fig. 4k and Supplementary Fig. 4k**). On the other hand, since Y530F-c-Src adopts a more open conformation, compared to WT-c-Src, it displayed a stronger binding with Cdh1 (**Supplementary Fig. 4i**). On the contrary, the *C*-tail EEI-mutant of c-Src, which bears a constant docking site for the SH2 domain, exhibited a reduced binding with Cdh1 presumably due to a more closed conformation (**Supplementary Fig. 4i**). In further support of this notion, our *in vitro* kinase assay results demonstrate that like WT-c-Src, Y530F-c-Src could also be inhibited by Cdh1 *in vitro* (**Fig. 4j and Supplementary Fig. 4h**). These results coherently support a model that Cdh1 binds to both *N*- and *C*-terminus of c-Src to inhibit Src function independent of c-Src Y530 phosphorylation status.

4. There is no analysis of how the Cdh1 N-terminal region interacts with the c-Src catalytic domain or how this interaction would inhibit c-Src substrate phosphorylation. Is this via the front side or backside?

Response: Our results reveal that Cdh1 *N*-terminal region could be phosphorylated by c-Src, suggesting *N*-terminus of Cdh1 as a c-Src substrate. To determine if Cdh1 *N*-terminal domain binds to c-Src kinase domain similar to other known c-Src substrates, we generated a c-Src mutant lack of the substrate binding segment (⁴²⁸PIKWTAPE⁴³⁵, Roskoski R (2015). *Pharmacol. Res.* 94, 9–25) (**e**). Co-IP and GST-pull down experiment results shown in **Supplementary Fig. 6e-f** support the notion that the *N*-terminal domain of Cdh1 interacts with c-Src through its substrate binding pocket. Furthermore, p85 Cortactin (CTTN), a cytoskeleton protein and c-Src substrate (Wu H & Parsons JT (1993). *J. Cell Biol.* 120,

1417–1426; Reynolds AB et al. (2014). *Oncogene*. 33, 4537–4547), competed with N-Cdh1 to bind c-Src *in vitro* (**Supplementary Fig. 6g**). Our results hence support the notion that as a c-Src substrate, Cdh1 N-terminus mainly interacts with c-Src via the front side of the kinase domain.

5. *Figure 1A/B: Although the authors confirmed the effects of Cdh1 depletion on c-Src pY419 using Cas9/sgRNA knockout, they should really show that re-expression of an shRNA-resistant form of Cdh1 reduces pY419 levels.*

Response: As kindly instructed by the reviewer, in the revised manuscript, we found that ectopically expressing Cdh1 in sgCdh1-MCF7 cells suppressed pY419-c-Src level (**Supplementary Fig. 1j**).

6. *Figure 1D/E: The authors generated a pool of Cdh1 knockout MCF7 cells, but it is clear that there are some cells that retain Cdh1. Can the authors generate a clone of Cdh1^{-/-} MCF7 cells?*

Response: We thank the reviewer for carefully examining our manuscript, the sgCdh1 cells used in our original figure panels were indeed generated as a pool from Cas9/sgRNA infection using lentiCRISPRv2 construct. Following the reviewer's instructions, we generated single sgCdh1 clones using sgCdh1-MCF7, T47D, and BT474 cells. We chose two clones from each cell line and the results could be found in **Fig. 1d and Supplementary Fig. 1e-f**. As shown in the figures, all the single sgCdh1 clones exhibited increased pY419-c-Src while pY530-c-Src remain unchanged, further supporting our results using anti-Cdh1 short hairpins and Cdh1^{-/-} MEFs.

Cdh1 needs to be re-expressed in the Cdh1^{-/-} MEFs (panel E) to show that the effects on PDGF-induced pY419 c-Src are reversed. Is the level of PDGFR pTyr increased in the Cdh1^{-/-} MEFs?

Response: As kindly suggested, in the **Supplementary Fig. 1l**, reintroducing Cdh1 in Cdh1^{-/-} MEFs suppressed PDGF-triggered c-Src activation as marked by the decrease of pY419-c-Src induction following PDGF treatment. It is noteworthy that Cdh1 genetic status did not affect PDGFR activation as evidenced by a relatively unchanged pattern of pY751-PDGFR-β signals (**Supplementary Fig. 1l**).

7. *Figure 2J-L: The authors conclude that the increased spontaneous mammary tumor formation observed in Cdh1^{+/-};Pten^{+/-} compound heterozygous mice was due to activation of c-Src, because levels of pY419 c-Src were elevated, but provide no direct evidence for this.*

Response: We fully agree with the reviewer that the increased breast tumor incident found in Cdh1^{+/-};Pten^{+/-} compound heterozygous mice might be due to multiple reasons, since Cdh1 itself targets a broad spectrum of ubiquitin substrates for proteolysis, many of which are well-characterized oncogenes, such as Plk1, Aurora kinases, Skp2, cyclin A, etc. To elucidate if the elevation of c-Src activity upon Cdh1 depletion at least in part contributes to Cdh1-deficiency induced tumorigenesis, we compared clonogenic survival and soft agar growth of MDA-MB-231 cells with Cdh1 knockdown and the combination of Cdh1 and c-Src knockdown (**Supplementary Fig. 1m**). As shown in **Supplementary Fig. 2f-i**, further depleting c-Src in shCdh1-MDA-MB-231 cells significantly suppressed cell proliferation and anchorage-independent growth. These results suggest Cdh1-mediated c-Src inhibition at least in part contributes to breast tumor development upon Cdh1 loss.

8. *Figure 3/S3: The c-Src 15RRSL18 D-box motif is very close to the N-terminal myristoyl group, which anchors c-Src to membranes, and this orientation may impose some topological constraints for Cdh1*

binding. Conversely Cdh1 binding might cause c-Src dissociation from the membrane, since the basic residues in this reading are thought to serve as anchors via interaction with phospholipid head groups. The GST-Src protein used for pull downs will lack this N-terminal lipid modification and therefore may not recapitulate what happens with native c-Src.

Response: We thank the reviewer for raising this important issue about the possible impact of c-Src myristoylation on Cdh1/c-Src interaction and vice versa. First of all, all the HA-c-Src constructs used in the manuscript, including transient expression constructs used in Fig. 3-4 and lentiviral constructs used in Fig. 5 are C-terminal HA-tagged. We also fully agree with the reviewer that the N-terminal GST-tagged c-Src proteins were expressed and purified from *E.coli*, lacking lipid modifications, thus are not suitable for assessing the binding between Cdh1 and myristoylated c-Src. Following the reviewer's kind suggestion, we generated a c-Src G2A mutant which has been shown as myristoylation-deficient (Patwardhan P & Resh MD (2010). *Molecular and Cellular Biology*. 30, 4094–4107). As shown in **Supplementary Fig. 3i**, we found that Cdh1 bound to both WT-c-Src and G2A-c-Src at the same level. Since c-Src has a strong myristoylation signal at its N-terminus (Cross FR et al. (1984). *Molecular and Cellular Biology*. 4, 1834–1842), this result indicates that c-Src myristoylation might not significantly affect Cdh1/c-Src interaction. On the other hand, immunofluorescence results demonstrate that deletion of Cdh1 did not significantly alter c-Src plasma membrane localization in breast cancer cells, a slightly increase of plasma membrane-associated c-Src could be found in the *Cdh1*-deleted cells (**Supplementary Fig. 3j**), which is consistently with the IF results from WT-c-Src and D1-RLAA-c-Src-expressing MDA-MB-231 cells (**Supplementary Fig. 5j**). Together our results suggest that although the D-box1 motif found in c-Src N-terminus is close to the c-Src myristoylation site, the binding between Cdh1 and c-Src is not significantly influenced by this modification. On the other hand, although Cdh1 negatively regulates c-Src activity, its role in c-Src plasma membrane localization tended to be marginal.

9. *Figure 4A: The authors have used an in vitro autophosphorylation assay to demonstrate an inhibitory effect of GST-Cdh1 on c-Src kinase activity, which is not really satisfactory. Since we are not shown the level of pY419 in the starting HA-Src sample, it is unclear how much this signal was increased during the assay.*

Response: We thank the reviewer for pointing out this issue, as kindly instructed, we repeated the *in vitro* kinase shown in the original Fig. 4a with a time 0 point prior to 60 min kinase reaction. As shown in the revised **Fig. 4a**, incubation of immunopurified c-Src kinase led to increased Y419 phosphorylation, which could be inhibited by co-incubation with purified, cleaved Cdh1 protein from the recombinant GST-Cdh1 fusion.

Moreover GST fusions proteins are dimers, making it impossible to determine the true efficiency of inhibition due to avidity effects,

Response: We agree with the reviewer that the strong dimer formation of GST-fusions might have unexpected outcomes. As kindly instructed, we repeated *in vitro* kinase assays using PreScission protease cleaved Cdh1 proteins as shown in **Fig. 4a-b**. We found the cleaved Cdh1 protein could also effectively inhibit both c-Src autophosphorylation and p85 Cortactin (CTTN) phosphorylation by c-Src.

and, in any case, the relative levels of HA-Src and GST-Cdh1 proteins used in the assay are not provided.

Response: As suggested, we loaded 10 times amount of the affinity-purified HA-Src immunoprecipitates and the recombinant Cdh1 protein used in our *in vitro* kinase assays onto the same SDS-PAGE gel. As demonstrated in **Supplementary Fig. 4b**, the quantified band intensity ratio of HA-Src versus Cdh1 was 1:2.2. Since we used 10% of the HA-Src for each *in vitro* kinase reaction, the actual ratio in a reaction should be 1:22. This information was included in the figure legend of **Supplementary Fig. 4a**.

An authentic c-Src substrate kinase assay needs to be carried out using monomeric Cdh1 - in Figure S4A the authors used recombinant CDK1 as a substrate, but CDK1 is not generally regarded as a bona fide c-Src substrate.

Response: Following the reviewer's insightful comment, in the revised manuscript, we repeated *in vitro* kinase assays using PreScission protease cleaved Cdh1 proteins as shown in **Fig. 4a-b**. We found the cleaved Cdh1 protein could also effectively inhibit both c-Src autophosphorylation and p85 Cortactin (CTTN) phosphorylation by c-Src.

We agree with the reviewer that CDK1 (p34^{cdc2}) is not among the best characterized c-Src substrates, although many commercially available c-Src substrate peptide products are synthesized based on the amino acids 6-20 of the CDK1 protein (KVEKIGEGTYGVVYK), such as the one from Millipore (cat#12-140, http://www.emdmillipore.com/US/en/product/Src-Substrate-Peptide,MM_NF-12-140). As kindly suggested by the reviewer, in the revised manuscript, we also included the C-terminal domain of p85 Cortactin (CTTN), a known c-Src substrate (Huang C *et al.* (1998). *J. Biol. Chem.* 273, 25770–25776; Reynolds AB *et al.* (2014). *Oncogene.* 33, 4537–4547), in our *in vitro* kinase assays. We validated c-Src-mediated CTTN phosphorylation (**Supplementary Fig. 4e**) and further demonstrated that monomeric Cdh1 suppressed c-Src-mediated CTTN phosphorylation *in vitro* (**Fig. 4b**). In addition, we found that Cdh1 N-terminus compete with CTTN-C in binding to c-Src, suggesting N-terminal Cdh1 as a c-Src substrate (**Supplementary Fig. 6e-f**).

Also, it is unclear why there are so many smaller GST antibody-positive bands in lanes 3 and 4.

Response: Again we thank the reviewer for careful examination of our manuscript. The small molecular weight bands were GST-Cdh1 truncations that were also co-purified using Glutathione Sepharose 4B beads during the affinity purification process. In the revised manuscript, we replaced the IB:GST blots with coomassie blue-stained gel blots to present the amount of GST proteins used in our experiments (**Supplementary Fig. 4a and 4c**). Furthermore, in the revised **Fig. 4b, 4j**, and **Supplementary Fig. 4f**, we also use coomassie blue staining to show the relative protein amount as well as the purity of both PreScission-cleaved Cdh1 and p85 Cortactin (CTTN) proteins.

Finally, did the authors check if the N-terminal domain of Cdh1 alone inhibits c-Src in vitro.

Response: As instructed, in the revised **Supplementary Fig. 4f**, we found that compared to WT-Cdh1, the N-terminal domain of Cdh1 displayed reduced capability in suppressing both c-Src autophosphorylation and c-Src-mediated CTTN phosphorylation. This result indicates that an efficient inhibition by Cdh1 requires both N- and C-terminal domains to restrain c-Src function, Cdh1 N-terminus might only serve a competitive substrate to compete with c-Src or CTTN for phosphorylation, thus only partially inhibited c-Src kinase activity (**Supplementary Fig. 4f**).

10. *Figure 4D: While it is true that co-expression of HA-Cdh1 increased the level of HA-Src-C that was brought down by Flag-Src-N (presumably this fragment lacks the N-terminal myristoyl group), all this demonstrates is that Cdh1 can bind both halves of Src and not that the association between the two halves of Src was increased by Cdh1 as the authors claim. The increased binding of the Y530F mutant Src-C might be due to the open configuration of the Y530F mutant, but a significant fraction of the N/C domain interaction energy comes from the SH2 domain-pY430 interaction, which would be missing in the case of Y530F mutant. Moreover, they did not use an SH2 domain (or an SH3 domain) mutant to determine whether loss of pY530/SH2 interaction is important.*

Response: We thank the reviewer for this insightful comment. Our results showed that Cdh1 promoted the binding between N-terminal and C-terminal domains of c-Src regardless of Y530 phosphorylation status (**Fig. 4i**), suggesting that Cdh1/c-Src interaction is not controlled by c-Src Y530 phosphorylation. As kindly suggested, we found that although a SH2-deficient R178A-c-Src-N (*Bibbins KB et al. (1993). Molecular and Cellular Biology. 13, 7278–7287*) failed to bind c-Src-C or Y530F-c-Src-C, the addition of Cdh1 promoted the interaction between both parts of the c-Src protein (**Fig. 4k and Supplementary Fig. 4k**). These results hence support a model that Cdh1 binds to both N- and C-terminus of c-Src to inhibit Src function independent of c-Src Y530 phosphorylation status.

11. *Figure 4J: The same criticisms apply to this experiment as to Figure 4A.*

Response: As kindly suggested, we repeated the *in vitro* kinase assay of original Fig. 4J using PreScission protease cleaved Cdh1 proteins as shown in revised **Supplementary Fig. 4h**. We found the cleaved Cdh1 protein could also effectively inhibit both WT-c-Src and Y530F-c-Src autophosphorylation. In addition, cleaved Cdh1 was found to suppress both WT-c-Src and Y530F-c-Src in phosphorylating C-terminus CTTN *in vitro* (**Fig. 4j**). In both revised figure panels, we included the phosphorylation signals before incubation (time 0 point) as kindly instructed.

12. *Figure 5: The authors attribute the increased tumorigenicity of the c-Src D1-RLAA expressing MDA-MB-231 cells to the fact that the mutant c-Src is no longer subject to inhibition by Cdh1, but they did not test the consequences of expressing c-Src D1-RLAA MDA-MB-231 in cells depleted of Cdh1, which is a key control.*

Response: We thank the reviewer for this perceptive remark. In the revised **Fig. 5i-j and Supplementary Fig. 5g-i**, we found that depletion of *Cdh1* in MDA-MB-231 cells stably expressing D1-RLAA-c-Src failed to further promote their clonogenic survival or anchorage independent growth in the soft agar. These results suggest that the increased tumorigenicity of Cdh1-binding deficient D1-RLAA c-Src mutant is largely due to its escape from the negative regulation by Cdh1.

13. *Figure 6: Based on the known structures of Cdh1, is Y148 actually accessible for phosphorylation. It should be noted that Y148 does not lie in a Src kinase consensus sequence, i.e. it lacks any preferred acidic residues on either side. Moreover, while c-Src expression was shown to increase Cdh1 pTyr levels upon co-expression in cells, the authors did not demonstrate that this increase occurred at Y148, and more importantly they did not determine what fraction of the Cdh1 population was phosphorylated at Y148, or whether or not pY148 was associated with APC/C. In addition, no measurements of APC/C E3 ligase activity with Y148F Cdh1 versus pY148 Cdh1 bound were carried out to establish that Y148 phosphorylation reduces its E3 ligase activity.*

Response: We thank the reviewer for pointing out that further experimental evidence to support a c-Src-dependent phosphorylation of Cdh1 at Y148 as well as an inhibitory role of pY148 in regulating APC^{Cdh1} ubiquitin E3 ligase activity. Inspired by the constructive comments from the reviewer, we have obtained the following results to demonstrate that c-Src-mediated Y148 phosphorylation of Cdh1 suppresses the binding between Cdh1 N-terminus and the APC core complex, and as a result, leads to reduced APC^{Cdh1} activity.

- 1) As kindly instructed, in the revised **Supplementary Fig. 6h**, we generated a structural illustration of Cdh1 docking in the APC complex with a D-box peptide from Hs11 bound to its WD40-repeats domain (PDB 5L9T: *Brown NG et al. (2016). Cell. 165, 1440–1453*). The S146-L150 region was highlighted in red. It is worth noting that this region of Cdh1, like other parts of the Cdh1 N-terminus, adopts a flexible configuration, and interacts with other APC subunits. In fact, a majority of the Cdh1 N-terminus appeared to be disordered in this Cryo-EM structure (PDB 5L9T), as well as from other Cryo-EM structures of the Anaphase Promoting Complex (PDB 4UI9, PDB 5A31: *Chang L et al. (2015). Nature. 522, 450–454*). This structural information indicates that the region flanking the Y148 site is likely exposed and quite flexible for protein-protein interaction as well as post-translational modifications.
- 2) In further support of this notion, previous reports including ours identified multiple serine and threonine phosphorylation sites for Cdh1, most of which are at a close proximity to Y148 (*Keck JM et al. (2007). Journal of Cell Biology. 178, 371–385*; *Fukushima H et al. (2013). Cell Rep. 4, 803–816*; *Lau AW et al. (2013). Cell Res. 23, 947–961*; *Wan L et al. (2017). Cancer Discovery. 7, 424–441*). These sites include pS151 and pS163 by CDK2/4 and ERK, pS146 by Plk1. Moreover, K169 has been shown acetylated in tumor cells to suppress APC^{Cdh1} function (*Kim HS et al. (2011). Cancer Cell. 20, 487–499*). Further, as the reviewer kindly pointed out above, Cdh1 contains a strong NLS signal at K156-K177 that mediates its interaction with importins (*Zhou Y et al. (2003). Journal of Biological Chemistry. 278, 12530–12536*). All these evidence indicate that Y148 is in a relatively flexible region that could be phosphorylated by c-Src.
- 3) We agree with the reviewer that the sequence flanking Y148 (¹⁴⁵VSPYSLSP¹⁵²) does not align very well with known c-Src substrate consensus E-E-I-Y-G-E-F (*Songyang Z & Cantley LC (1995). Trends Biochem. Sci. 20, 470–475*). However, experimental evidence from a number of well-characterized c-Src substrates indicates that the c-Src substrate consensus is not very stringent. For example, Y705-STAT3 has been shown as a c-Src target (AAPYLKT) (*Yu CL et al. (1995). Science. 269, 81–83*; *Bromberg JF et al. (1998). Molecular and Cellular Biology. 18, 2553–2558*), the surrounding sequence of which also lacks acidic residues.
- 4) Experimentally, in **Fig. 6g**, using a panel of YF mutants, we have clearly demonstrated that only the mutation of Y148 abrogated c-Src-mediated Cdh1 phosphorylation.
- 5) Furthermore, using the newly generated specific p-Y148-Cdh1 antibody, we confirmed that mutation of Y148 completely abolished c-Src-mediated phosphorylation both *in vitro* (**Fig. 6h**) and in cells (**Supplementary Fig. 6c**).
- 6) Our responses to the *General Comment section* described our newly obtained results that the majority of Cdh1 is localized to the cytoplasm where it interacts with c-Src in breast cancer cells (**Fig. 3e-f**).
- 7) Moreover, we found that Cdh1 binds to c-Src in both cytoplasmic and nuclear fractions with a stronger binding found in the cytoplasm, supporting the notion that APC-free Cdh1 interacts with c-Src in the cytoplasm to suppress c-Src kinase activity (**Fig. 7h**).
- 8) As kindly instructed by the reviewer, in the revised **Fig. 6i**, we found that PDGF treatment, which activates c-Src, led to a marked increase of Cdh1 Y148 phosphorylation.

- 9) On the other hand, deletion of c-Src using Cas9/sgSrc or inhibiting c-Src using dasatinib or saracatinib led to a reduction of pY148 and subsequent increased binding with APC subunits (**Fig. 6j-k and Supplementary Fig. 6i-j**).
- 10) Furthermore, using N-terminal domain of Cdc20 as APC^{Cdh1} substrate, we found that compared to WT-Cdh1, Y148E- and 4D-Cdh1 failed to promote Cdc20-N ubiquitination (**Supplementary Fig. 6l**). In contrast, 4A- and Y148F-Cdh1 exhibited increased activity in promoting Cdc20-N ubiquitination (**Supplementary Fig. 6l**).
- 11) In concert with the notion that N-terminal phosphorylation of Cdh1 disrupts Cdh1/APC interaction, we observed a reduction of binding between Cdh1 and APC6 in MDA-MB-231 cells stably expressing 4D- or Y148E-Cdh1 mutant (**Fig. 7e**).

Together our newly obtained results as well as evidence from previous reports demonstrate that c-Src phosphorylates Cdh1 at Y148, an event disrupts Cdh1's interaction with the APC core complex and thus inhibits APC^{Cdh1} E3 ligase activity.

14. Figure 6: Dasatinib is a dirty kinase inhibitor, inhibiting Tyr kinases other than c-Src, such as Abl /Arg, other SFKs, as well as several Ser/Thr kinases. In consequence, one cannot interpret the results of experiments using dasatinib as being due to c-Src inhibition. It would be better to use saracatinib/AZD0530, which is more selective but by no means c-Src selective.

Response: We fully agree with the reviewer that dasatinib is not a very specific c-Src inhibitor. As kindly instructed, we included the following results in the revised manuscript to demonstrate that like dasatinib, saracatinib also inhibits Cdh1 Y148 phosphorylation and restores APC^{Cdh1} function.

- 1) In **Supplementary Fig. 8t**, we demonstrated that analogous to dasatinib treatment, saracatinib suppressed Cdh1 Y148 phosphorylation, which led to increased Cdh1/APC interaction.
- 2) We also compared the role of Src inhibitors dasatinib and saracatinib, MEK inhibitors PD0325901 and trametinib in restoring APC^{Cdh1} activity in both MDA-MB-231 and SUM159PT cells (**Supplementary Fig. 8p-q**).
- 3) Additionally, we found that similar to dasatinib and PD0325901, combinational treatment of MDA-MB-231 cells with saracatinib and trametinib significantly suppressed the survival of MDA-MB-231 cells whereas the efficacy of single-agent treatment tended to be marginal (**Supplementary Fig. 8r-s**).

15. Figure 7: The authors provide no evidence that the expression of the 4D phosphomimetic mutant Cdh1 decreased Y148 phosphorylation. It is unclear what fraction of Cdh1 is phosphorylated at Y148, and whether this represents a significant fraction of Cdh1, or whether the pY148 Cdh1 molecules are selectively released from APC/C. The authors need to generate pY148 specific antibodies to characterize Y148 phosphorylation.

Response: Our responses to the *General Comment section* described our newly obtained results that the majority of Cdh1 is localized to the cytoplasm where it interacts with c-Src in breast cancer cells (**Fig. 3e-f**). Moreover, we found that Cdh1 binds to c-Src in both cytoplasmic and nuclear fractions with a stronger binding found in the cytoplasm, supporting the notion that APC-free Cdh1 interacts with c-Src in the cytoplasm to suppress c-Src kinase activity (**Fig. 7h**). Interestingly, in both 4A- and 4D-Cdh1-expressing MDA-MB-231 cells, we found Cdh1 Y148 phosphorylation was significantly reduced (**Fig. 7e**). Since 4A-Cdh1 was mainly found in the nucleus (**Fig. 7a-b**), the decreased p-Y148-Cdh1 found in 4A-Cdh1-expressing cells was likely due to the reduced 4A-Cdh1/Src binding in cells (**Fig. 7e**). On the other hand, 4D-Cdh1 exhibited a reduced binding with c-Src (**Supplementary Fig. 9f**), and was found incapable to suppress c-Src autophosphorylation *in vitro* (**Supplementary Fig. 9g**).

As kindly suggested by the reviewer, we generated a specific p-Y148-Cdh1 antibody (**Fig. 6h and Supplementary Fig. 6c**). We confirmed that mutation of Y148 completely abolished c-Src-mediated phosphorylation both *in vitro* (**Fig. 6h**) and in cells (**Supplementary Fig. 6c**). Moreover, p-Y148-Cdh1 was elevated upon PDGF treatment in breast cancer cells (**Fig. 6i**), while it was suppressed when treated with Src inhibitor or in Src-deleted breast cancer cells (**Fig. 6j-k and Supplementary Fig. 6i-j**).

16. *Figures 6 and 7: In general, many additional experiments on ERK and c-Src-mediated phosphorylation of Cdh1 are needed to provide stronger evidence for the authors hypothesis that c-Src phosphorylation of Y148 in Cdh1 is an important regulatory event that inhibits Cdh1/APC/C activity as a negative feedback mechanism.*

Response: We fully agree with the reviewer that it is important to provide further experimental evidence to support our major conclusion in the revised **Fig. 6-8** (original Fig. 6 and 7) that c-Src-mediated Y148 phosphorylation of Cdh1 suppresses APC^{Cdh1} function in triple negative breast cancer cells. Following the reviewer's kind suggestion, we have obtained the following results to further strengthen our points:

- 1) We generated MDA-MB-231 cell lines stably expressing WT-, 4D-, 4A-, Y148E-, and Y148F-Cdh1 mutants, and found that 4A-Cdh1 and Y148F-Cdh1-expressing cells exhibited a higher APC^{Cdh1} activity as evidenced by reduced expression Plk1 and Cdc6 (**Supplementary Fig. 8l**).
- 2) Compared to WT-Cdh1-expressing shCdh1-MDA-MB-231 cells, 4D-Cdh1 and Y148E-Cdh1-expressing cells exhibited faster growth in clonogenic survival assays (**Supplementary Fig. 8o**). In contrast, 4A-Cdh1 and Y148F-Cdh1-expressing cells showed poor survival and cell proliferation (**Supplementary Fig. 8o**).
- 3) Further depletion of c-Src failed to suppress cell proliferation in 4D-Cdh1 and Y148E-Cdh1-expressing MDA-MB-231 cells (**Supplementary Fig. 8m-n**).
- 4) Similar to 4D-Cdh1-expressing MDA-MB-231 cells, we found that Y148E-Cdh1-expressing MDA-MB-231 cells are resistant to dasatinib and PD0325901 treatment (**Fig. 8f-h**), and exhibited a high expression level of known APC^{Cdh1} ubiquitin substrate Plk1 and Cdc6 (**Fig. 8f**).
- 5) Furthermore, analogous to our *in vitro* tumorigenesis assays (**Fig. 8i-j and Supplementary Fig. 8r-s**), combinational treatment of nude mice bearing SUM159PT xenograft tumors with dasatinib (5 mg/kg) and trametinib (1 mg/kg) significantly suppressed tumor growth *in vivo* (**Fig. 8k and Supplementary Fig. 8u**). Further examination of tumor samples revealed a marked decrease of Plk1, Cdc6 and Skp2, suggesting an elevated APC^{Cdh1} E3 ligase activity (**Supplementary Fig. 8v**).

Reviewer #2, Expertise: cancer biology (Remarks to the Author)::

The manuscript by Han et al. reports a new interaction between the APC/C coactivator Cdh1 and the kinase Src in breast cancer cells. Silencing of Cdh1 results in Src hyperphosphorylation (Y419) and over-activation in an APC/C-independent manner, resulting in increased proliferation and tumorigenic properties of breast cancer cells. Biochemical studies show that Cdh1 and Scr can bind and inhibit each other using different mechanisms. On one hand, Cdh1 locks Src in a closed conformation inhibiting its ability to be activated. On the other, Src is able to phosphorylate Cdh1, thus preventing its binding to APC/C. The authors finally proposed that inhibiting Src kinase activity may have therapeutic activity in triple negative breast cancer cells, at least partially by re-activating the tumor suppressor properties of Cdh1.

In general, the information provided in the manuscript is of high technical quality and the conclusions are solid, novel and supported by the data. The current version of the manuscript contains a significant amount of information, well presented and discussed, and it could be published after revision of a couple of questions that are not completely clear or have not been analyzed in detail.

Response: We sincerely thank the reviewer for recognizing the novelty, the data quality and the significant impact of this study. We also thank the reviewer for raising the constructive comments that have been very helpful in guiding us along our revision. Below please find our point-by-point responses to the reviewer's insightful critiques.

Major points

1. Data in Figure 2 shows that ablation of Cdh1 results in higher oncogenic properties (proliferation, migration, etc.) in the presence of higher phosphorylation of Src. Similar correlation is found after crossing Cdh1 and Pten het mice. From these studies the authors conclude that "these findings demonstrate that Cdh1 functions as a tumor suppressor in vivo in part by suppressing the activation of the Src oncogenic pathway" (lines 159-160). This conclusion requires very easy assays in which (at least some of) the oncogenic properties of Cdh1 are prevented upon silencing (or inhibition or expression of kinase-dead mutants, etc) or Src (at least in vitro).

Response: As kindly suggested, to elucidate if the elevation of c-Src activity upon Cdh1 depletion at least in part contributes to Cdh1-deficiency induced tumorigenesis, we compared clonogenic survival and soft agar growth of MDA-MB-231 cells with Cdh1 knockdown and the combination of Cdh1 and c-Src knockdown (**Supplementary Fig. 1m**). As shown in **Supplementary Fig. 2f-i**, further depleting c-Src in shCdh1-MDA-MB-231 cells significantly suppressed cell proliferation and anchorage-independent growth. These results suggest Cdh1-mediated c-Src inhibition at least in part contributes to the increased breast cancer cell proliferation upon Cdh1 loss.

2. In Figure 1k, the kinetics of Plk1 is similar to Cdh1, but it should be the opposite as Plk1 is a Cdh1 target for proteasome-dependent degradation. This is unusual as they should follow a pattern similar to P-Y419? As this experiment describes the levels of these proteins during checkpoint recovery (see minor points below), a description of the exact stage of the cell cycle (either by FACS or expression of E, A, B cyclins) would help to understand this discrepancy.

Garcí-Higuera I et al. (2008). Nature Cell Biology. 10, 802–811 **Fig. 4b:** Immunodetection of the indicated cell-cycle proteins in total lysates from *Fzr1*^{+/+} and *Fzr1*^{-/-} MEFs at the indicated time-points after serum stimulation. An additional sample from asynchronous HeLa cells was used to identify the human proteins. β -actin was used as a loading control.

Wei W et al. (2004). Nature. 428, 194–198: **Fig. 3d:** Immunoblot analysis of HeLa cells transfected with the indicated siRNA, synchronized by growth in nocodazole, and then released for the indicated periods of time.

Response: We thank the reviewer for careful examination of our manuscript and for raising the concern regarding the Plk1 levels across the cell cycle in original Fig. 1k and Supplemental Fig. 1l. We agree with the reviewer that it is confusing that the fluctuation of Plk1 levels across the cell cycle mirrored the pattern of Cdh1. This similarity could also be found in previous reports. As shown in the figure panels adopted from *Garcí-Higuera et al. 2008* and *Wei et al. 2004*, the fluctuating patterns of APC^{Cdh1} substrates including Aurora A, Cdc6, Geminin, Cyclin A2, Plk1 and Skp2 reflect that of Cdh1. The activation of APC^{Cdh1} is tightly regulated during the cell cycle progression. Cdh1 is sequestered from the APC core complex until late M phase (*Kernan J et al. (2018). Biochim Biophys Acta Mol Cell Res. 1865, 1924–1933*). Although the Cdh1 protein level continues to accumulate from S to G2 phases, due to its N-terminal phosphorylation by S and G2 cyclin/CDKs as well as the inhibitory binding with Emi1, Cdh1 is still largely inactive. During anaphase, dephosphorylation of Cdh1 by the Cdc14 phosphatases allows the assembly of active APC^{Cdh1}, which replaces APC^{Cdc20} to ubiquitinate mitotic cyclins as well as Cdc20, driving cells out of mitosis into G0/G1. APC^{Cdh1} activity persists throughout G1 until Cdh1 is inactivated at the G1 to S transition through degradation, phosphorylation and binding with Emi1 (*Skaar JR & Pagano M (2008). Nat Cell Biol. 10, 755–757*).

As demonstrated in the figure panels from *Garcí-Higuera et al. 2008* and *Wei et al. 2004*, as well as from **Fig. 1k** and **Supplemental Fig. 1n, 1p**, although Cdh1 protein level peaks in G2/M phases, its activation only restricted to early through mid-late G1 phase. In such a scenario, Plk1 and other APC^{Cdh1} substrates also peak along with Cdh1, while getting destabilized when APC^{Cdh1} is activated (*Qiao X et al. (2010). Cell Cycle. 9, 3904–3912*). Activated APC^{Cdh1} also drives its self-degradation in early to mid-G1 and further destabilized by SCF ^{β -TRCP}- and SCF^{Cyclin F}-mediated proteolysis in late G1 and S phases (*Fukushima H et al. (2013). Cell Rep. 4, 803–816; Choudhury R et al. (2016). Cell Rep. 16, 3359–3372*).

As kindly instructed by the reviewer, we performed FACS analysis of the double thymidine block and release synchronization experiment using MDA-MB-231 cells shown in **Supplemental Fig. 1o**. The M/G1 transition occurs 10-12h after release from double thymidine block, when both Cdh1 and Plk1 levels started to decrease (**Fig. 1k**), supporting the notion that APC^{Cdh1} becomes active in early G1 phase. On the other hand, since Cdh1-mediated c-Src inhibition is independent of APC, the Cdh1 protein

abundance, rather than APC^{Cdh1} activity, dictates c-Src activity across the cell cycle. Therefore, in contrast to known APC^{Cdh1} ubiquitin substrates, p-Y419-c-Src exhibited an inversed pattern to Cdh1 levels (**Fig. 1k** and **Supplemental Fig. 1n, 1p**). Such a difference further supports an APC-independent regulation of c-Src function by the Cdh1 protein.

3. lines 287-290. Src deficiency results in decrease in the levels of various APCC-Cdh1 substrates. However, this may be the effect of cell cycle arrest upon Src inhibition. Cell cycle profiles or protein levels of cell cycle regulators that are not APC/C-targets should be analyzed here. This also applies to the effect of Src inhibitors (lines 296-297, 328-330).

Response: We fully agree with the reviewer that since c-Src also plays a role in controlling cell cycle progression via activating MAPK and PI3K pathways (*Thomas SM & Brugge JS (1997). Annu. Rev. Cell Dev. Biol. 13, 513–609*), it will be important to analyze cell cycle profiles and cell cycle factors upon Src-depletion or inhibition. To this end, we found that in MDA-MB-231 cells, *c-Src*-deletion or inhibition by dasatinib or saracatinib led to a moderate increase of G1 phase while a decrease of S phase cells (**Supplemental Fig. 6m-n**). These results support the observation that depletion of *c-Src* in MDA-MB-231 cells suppressed clonogenic survival and anchorage-independent growth in soft agar (**Supplemental Fig. 2f-i**). Furthermore, we found that compared to APC^{Cdh1} substrates, which exhibited a dramatic reduction upon *c-Src* deletion or inhibition, other cell cycle regulators including cyclin E and cyclin D1 remained largely unchanged in *c-Src*-deficient cells (**Fig. 6j-k** and **Supplemental Fig. 6i**).

Furthermore, using N-terminal domain of Cdc20 as APC^{Cdh1} substrate, we found that compared to WT-Cdh1, Y148E- and 4D-Cdh1 failed to promote Cdc20-N ubiquitination (**Supplementary Fig. 6l**). In contrast, 4A- and Y148F-Cdh1 exhibited increased activity in promoting Cdc20-N ubiquitination (**Supplementary Fig. 6l**). Together, our newly obtained results suggest that *c-Src* might regulate cell cycle progression through both Cdh1-dependent and Cdh1-independent routes.

Minor points

Line 126. Cdh1 peaks in G2/M. The expression of Cdh1 shown in this panel does not correspond to normal levels during a cell cycle but levels after checkpoint recovery (samples recover from thymidine block). The authors should modify the text accordingly.

Response: We agree with the reviewer that the oscillation of Cdh1 levels shown in **Fig. 1k** and **Supplemental Fig. 1n** were obtained from the recovered cells at different time points after release from double thymidine arrest. As kindly suggested, in the revised manuscript, we rephrased description to avoid confusion. In addition to double thymidine arrest, we also used nocodazole arrest to synchronize sgGFP-MDA-MB-231 and sg*Cdh1*-MDA-MB-231 cells and released them back into the cell cycle. As shown in the revised **Supplemental Fig. 1p**, we observed an inversed correlation between Cdh1 protein abundance and pY419-Src from M to G1 cell cycle phases.

Lines 334-336. Epithelial tumors were described in Cdh1 +/- mice, not in Sox2-Cre conditional mice.

Response: We thank the reviewer for pointing out this mistake, in the revised manuscript, we have corrected this error.

Reviewer #3, Expertise: Breast cancer, signalling (Remarks to the Author):

Cdh1 functions as a coactivator of the Anaphase Promoting Complex (APC/C) during late mitosis and early G1 phase negatively regulating the stability of several oncogenic substrates as Plk1, Cdc6, Skp2 or Cyclin A. However, Cdh1 protein has also been reported to have APC/C- independent functions. In this work, Han et al. report Cdh1 as a Src-interacting protein that negatively regulates its oncogenic activity in breast cancer cells independently of APC/C. Cdh1 reinforces Src inactive closed conformation perturbing its kinase activity and its depletion promotes tumor growth in diverse breast cancer cell lines and Cdh1 +/- Pten +/- mouse models. Moreover, the authors also found that Src kinase phosphorylates Cdh1 and disrupts its interaction with the APC/C complex, thus impairing the tumor suppressor role of APC/Cdh1. Finally, pharmacological inhibition of Src and MEK, which also phosphorylates the N-terminus part of Cdh1, synergistically potentiates the suppression of cell viability in TNBC cells suggesting this drug combination as a more efficient treatment of breast cancer than monotherapy.

In this manuscript, Han and colleagues describe a novel interplay among Cdh1, Src and APC/C and its contribution in breast tumor growth. The work is, in general, experimentally well supported. However, some questions must be addressed prior to publication.

Response: We thank the reviewer for recognizing the novelty of our manuscript, we also appreciate the reviewer's insights, especially in advising us to assess the synergy of trametinib and dasatinib *in vivo*. As the reviewer could find out in our point-by-point responses below, the results we obtained enlightened by the constructive comments from the reviewer have strengthened our manuscript and provided the translational value of our discoveries.

1) In the Figure 1 D. Authors show a Western-Blot performed in Cdh1 CRISPR-Cas9-mediated KO cells. However, a slight band is still present when Cdh1 is analyzed. Could be explained because the authors are not working with pure KO pools of cells?

Response: We thank the reviewer for carefully examining our manuscript, the sgCdh1 cells used in our original figure panels were indeed generated as a pool from Cas9/sgRNA infection using lentiCRISPRv2 construct. Following the reviewer's instructions, we generated single sgCdh1 clones using sgCdh1-MCF7, T47D, and BT474 cells. We chose two clones from each cell line and the results could be found in **Fig. 1d and Supplementary Fig. 1e-f**. As shown in the figures, all the single sgCdh1 clones exhibited increased pY419-c-Src while pY530-c-Src remain unchanged, further supporting our results using anti-Cdh1 short hairpins and Cdh1^{-/-} MEFs.

2) In the figure 1k it is not clear why Plk1, which is targeted by APC Cdh1, is upregulated when Cdh1 is high along cell cycle.

Response: We thank the reviewer for careful examination of our manuscript and for raising the concern regarding the Plk1 levels across the cell cycle in original Fig. 1k and Supplemental Fig. 1l. We agree with the reviewer that it is confusing that the fluctuation of Plk1 levels across the cell cycle mirrored the pattern of Cdh1. This similarity could also be found in previous reports. As shown in the figure panels adopted from *Garcí-Higuera et al. 2008* and *Wei et al. 2004*, the fluctuating patterns of APC^{Cdh1} substrates including Aurora A, Cdc6, Geminin, Cyclin A2, Plk1 and Skp2 reflect that of Cdh1. The activation of APC^{Cdh1} is tightly regulated during the cell cycle progression. Cdh1 is sequestered from the APC core complex until late M phase (*Kernan J et al. (2018). Biochim Biophys Acta Mol Cell Res. 1865,*

Garcí-Higuera I et al. (2008). *Nature Cell Biology*. 10, 802–811 **Fig. 4b**: Immunodetection of the indicated cell-cycle proteins in total lysates from *Fzr1*^{+/+} and *Fzr1*^{-/-} MEFs at the indicated time-points after serum stimulation. An additional sample from asynchronous HeLa cells was used to identify the human proteins. β-actin was used as a loading control.

Wei W et al. (2004). *Nature*. 428, 194–198: **Fig. 3d**: Immunoblot analysis of HeLa cells transfected with the indicated siRNA, synchronized by growth in nocodazole, and then released for the indicated periods of time.

1924–1933). Although the Cdh1 protein level continues to accumulate from S to G2 phases, due to its N-terminal phosphorylation by S and G2 cyclin/CDKs as well as the inhibitory binding with Emi1, Cdh1 is still largely inactive. During anaphase, dephosphorylation of Cdh1 by the Cdc14 phosphatases allows the assembly of active APC^{Cdh1}, which replaces APC^{Cdc20} to ubiquitinate mitotic cyclins as well as Cdc20, driving cells out of mitosis into G0/G1. APC^{Cdh1} activity persists throughout G1 until Cdh1 is inactivated at the G1 to S transition through degradation, phosphorylation and binding with Emi1 (Skaar JR & Pagano M (2008). *Nat Cell Biol*. 10, 755–757).

As demonstrated in the figure panels from *Garcí-Higuera et al. 2008* and *Wei et al. 2004*, as well as from **Fig. 1k** and **Supplemental Fig. 1n, 1p**, although Cdh1 protein level peaks in G2/M phases, its activation only restricted to early through mid-late G1 phase. In such a scenario, Plk1 and other APC^{Cdh1} substrates also peak along with Cdh1, while getting destabilized when APC^{Cdh1} is activated (Qiao X et al. (2010). *Cell Cycle*. 9, 3904–3912). Activated APC^{Cdh1} also drives its self-degradation in early to mid-G1 and further destabilized by SCF^{β-TRCP}- and SCF^{Cyclin F}-mediated proteolysis in late G1 and S phases (Fukushima H et al. (2013). *Cell Rep*. 4, 803–816; Choudhury R et al. (2016). *Cell Rep*. 16, 3359–3372).

As kindly instructed by the reviewer, we performed FACS analysis of the double thymidine synchronized and released MDA-MB-231 cells shown in **Supplemental Fig. 1o**. The M/G1 transition occurs 10-12h after release from double thymidine block, when both Cdh1 and Plk1 levels started to decrease (**Fig. 1k**), supporting the notion that APC^{Cdh1} becomes active in early G1 phase. On the other hand, since Cdh1-mediated c-Src inhibition is independent of APC, the Cdh1 protein abundance, rather than APC^{Cdh1} activity, dictates c-Src activity across the cell cycle. Therefore, in contrast to known APC^{Cdh1} ubiquitin substrates, p-Y419-c-Src exhibited an inversed pattern to Cdh1 levels (**Fig. 1k** and **Supplemental Fig. 1n, 1p**). Such a difference further supports an APC-independent regulation of c-Src function by the Cdh1 protein.

3) In this manuscript authors demonstrate a complex interplay among Cdh1, Src and APC/C. Cdh1 downregulation activates downstream oncogenic targets of Src (e.g. p-YAP). In fact, according to the pro-metastatic role of Src in cancer, Cdh1 depleted cells tend to increase its migration abilities (Figure 2g). On the other hand, APC Cdh1 substrates are also negatively regulated by Cdh1 (e.g. Plk1, Skp2,

Cdc6). However, it is unclear to what extent *Cdh1* action is mediated by *Src* or alternatively by the APC *Cdh1* complex in breast cancer. Results from Figure 7 suggest that *Cdh1* action is ultimately mediated through APC/C complex since *Src/MEK* inhibitors have not a deep effect when a phosphomimetic 4D-*Cdh1* mutant is used. To further complete the study, additional experiments can be performed:

- Since *Cdh1* deletion delays mitotic exit and increases genomic instability and chromosomal aberrations (García-Higuera et al. 2008) due to defective APC functionality, analysis of genomic defects or study of abnormal mitosis in *Cdh1* breast cancer cells is relevant.

Response: We thank the reviewer for this brilliant suggestion of examining genomic instability in breast cancer cells upon depleting *Cdh1*. As shown in the revised **Supplemental Fig. 2l-m**, deletion of *Cdh1* in MDA-MB-231 cells led to an increased number of aneuploid cells, a marker for genomic instability. As the reviewer kindly pointed out, delayed mitotic exit and increased chromosome abnormalities were observed in *Cdh1*^{-/-} cells (García-Higuera I et al. (2008). *Nature Cell Biology*. 10, 802–811; Li M et al. (2008). *Nature Cell Biology*. 10, 1083–1089), which might be due to inefficient degradation of mitotic cyclins and kinases, as well as DNA replication factors including *Cdc6* and Geminin. Different from *Cdh1*^{-/-} MEFs, which proliferate slower compared to *Cdh1*^{+/+} MEFs (García-Higuera I et al. (2008). *Nature Cell Biology*. 10, 802–811), breast cancer cells with *Cdh1* depletion exhibited faster growth (**Fig. 2**). We noticed that in both shRNA and Cas9/sgRNA-mediated *Cdh1* depleting experiments, breast cancer cells encountered a cell cycle arrest at the early stage upon *Cdh1*-depletion, but the cells soon recovered from arrest and started to proliferate at a higher rate compared to the control group. We speculate that *Cdh1* loss-triggered genomic instability activates the DNA damage response pathway, which induces the expression of CDK inhibitors and pro-apoptotic proteins (Qiao X et al. (2010). *Cell Cycle*. 9, 3904–3912), leading to cell cycle arrest. However, tumor cells typically harbor an incompetent DNA damage response pathway due to genetic aberrancies such as *p53* mutation or *BRCA1* silencing, therefore breast cancer cells might quickly recover from *Cdh1*-loss induced genotoxic stress and eventually took advantage of it. We are of the opinion that this will be an exciting direction for future in-depth investigation of *Cdh1*'s role in genomic instability and tumorigenesis.

- Downregulation of *Src* in sh*Cdh1* cells (Figure 2) would help to discern whether the mechanism of action of *Cdh1* is mainly mediated by *Src* downstream pathways or the APC *Cdh1* complex.

Response: We fully agree with the reviewer that the increased breast tumor incidence found in *Cdh1*^{+/-};*Pten*^{+/-} compound heterozygous mice might be due to multiple reasons, since *Cdh1* itself targets a broad spectrum of ubiquitin substrates for proteolysis, many of which are well-characterized oncogenes, such as *Plk1*, Aurora kinases, *Skp2*, cyclin A, etc. To elucidate if the elevation of c-*Src* activity upon *Cdh1* depletion at least in part contributes to *Cdh1*-deficiency induced tumorigenesis, we compared clonogenic survival and soft agar growth of MDA-MB-231 cells with *Cdh1* knockdown and the combination of *Cdh1* and c-*Src* knockdown (**Supplementary Fig. 1m**). As shown in **Supplementary Fig. 2f-i**, further depleting c-*Src* in sh*Cdh1*-MDA-MB-231 cells significantly suppressed cell proliferation and anchorage-independent growth. These results suggest *Cdh1*-mediated c-*Src* inhibition at least in part contributes to breast tumor development upon *Cdh1* loss.

- Alternatively, overexpression/downregulation of *Cdh1* in D1-RLAA *Src* mutant (abolishing the interaction between *Cdh1* and *Src*) also is a sound strategy to test the relevance of *Cdh1/Src* axis in breast tumorigenesis.

Response: We thank the reviewer for this perceptive remark. In the revised **Fig. 5i-j and Supplementary Fig. 5g-i**, we found that depletion of *Cdh1* in MDA-MB-231 cells stably expressing D1-RLAA-c-Src failed to further promote their clonogenic survival or anchorage independent growth in the soft agar. These results suggest that the increased tumorigenicity of Cdh1-binding deficient D1-RLAA c-Src mutant is largely due to its escape from the negative regulation by Cdh1.

4) *Althouht it seems that combinatorial treatment might be potentially promising in BCa treatment, in vitro studies presented in the figure 7 could be significantly improved by treating mice harboring some of the breast cancer cell lines used in the work with monotherapy or combined therapy (Src/MEK inhibitors).*

Response: Following the reviewer's insightful suggestion, we found that analogous to our *in vitro* tumorigenesis assays (**Fig. 8i-j and Supplementary Fig. 8r-s**), combinational treatment of nude mice bearing SUM159PT xenograft tumors with dasatinib (5 mg/kg) and trametinib (1 mg/kg) significantly suppressed tumor growth *in vivo* (**Fig. 8k and Supplementary Fig. 8u**). Further examination of tumor samples revealed a marked decrease of Plk1, Cdc6 and Skp2, suggesting an elevated APC^{Cdh1} E3 ligase activity (**Supplementary Fig. 8v**).

Minor points:

1) *Size (KDa) of the bands should be shown in the WB figures to facilitate the understanding.*

Response: As kindly instructed, we have included the molecular weight markers to all the WB blots and coomassie blue-stained gel images in the revised manuscript.

2) *In the line 138 mention: "having demonstrated that Cdh1 suppresses Src function in melanoma cells" should be replaced by "breast cancer cells" .*

Response: We apologize for this mistake in the manuscript, we have made corrections as suggested in the revised manuscript.

3) *In the lines 298/299 is mentioned that MEK has been shown "to phosphorylate the N-terminus of Cdh1 and to activate APC Cdh1 in melanoma cells" . However, according to the bibliography (Wan et al. 2017) MEK/ERK inhibits APC Cdh1 activity.*

Response: Again, we apologize for this error in the original manuscript. In the revised manuscript, we have clarified the statement as follows:

"in a similar fashion as using the MEK inhibitor PD0325901 (MEKi), which has been shown to inhibit Cdh1 N-terminal phosphorylation and to restore APC^{Cdh1} function in melanoma cells".

Reviewers' comments:

Reviewer #1:

The authors have done a significant amount of additional work to address my concerns and those of Reviewers 2 and 3. For the most part I am now satisfied, and am reasonably convinced that cytoplasmic Cdh1 can play a role in the negative regulation of c-Src tyrosine kinase activity in breast cancer cells. A few remaining comments/concerns:

1. If Cdh1 is a c-Src substrate and the Y148 target region is bound to the active site as a substrate, then one would expect Cdh1 to be released once it is phosphorylated. Did the authors check with their new anti-pY148 antibodies whether the pY148 population of Cdh1 is associated with c-Src or rather is mostly unassociated and soluble.
2. While their new anti-pY148 antibodies provide direct evidence that Y148 is phosphorylated in breast cancer cells, they did not check the stoichiometry of Y148 phosphorylation in Cdh1 in these cells.
3. The authors should realize that, in general, glutamate cannot be used as a phosphomimetic for pTyr, because the charge and geometry of the negative charge on the Glu COOH group are totally different from those of the phosphate on pTyr, and for this reason it is hard to interpret any phenotypes they observed for the pY148E Cdh1 mutant.

Reviewer #2:

The revised manuscript has improved greatly and I congratulate the authors for the effort in addressing all reviewers' points. My only concern remains at the conclusion level as it is very difficult to conclude that the role of Cdh1 in breast cancer depends on the Src-Cdh1 axis. I do agree that this connection is very important and the rest of the manuscript is really great. Simply I would recommend to note down the conclusions about the relevance in vivo, which cannot be established from the experiments in the manuscript. For instance the word "governs" in the title may be too strong. Other than that, I think the manuscript is of very high quality and very interesting technical and scientifically and deserves publication.

Reviewer #3:

Overall, the authors have address most of my previous concerns. In particular, they have provided reasonable explanations and data to understand and clarify the initially reported variability in the levels of Plk1 through cell cycle phases with high Cdh1. They have also clarify my concerns regarding the remaining Cdh1 expression in CRISPR/Cas9 mediated Cdh1 knockout cells. But more importantly, they have engaged on a series of new experiments to clarify the consequences of abnormal mitosis in Cdh1 breast cancer cells, its relevance and the potential consequences of the newly created genotoxic stress. In addition, they have extended the implication of their results to preclinical experimental mouse models. Overall, the manuscript has largely improved from the review process and provides a well-supported relevant novel insight into the interplay between Cdh1, Src and APC/C.

Point-by-Point Responses to the Reviewers' Critiques (NCOMMS-18-18370A)

Reviewer #1:

The authors have done a significant amount of additional work to address my concerns and those of Reviewers 2 and 3. For the most part I am now satisfied, and am reasonably convinced that cytoplasmic Cdh1 can play a role in the negative regulation of c-Src tyrosine kinase activity in breast cancer cells. A few remaining comments/concerns:

Response: We appreciate the positive comments from the reviewer to our revised manuscript, and thank the reviewer again for thorough analysis of our data and raising constructive critiques. Below please find our point-by-point responses to the remaining concerns:

1. If Cdh1 is a c-Src substrate and the Y148 target region is bound to the active site as a substrate, then one would expect Cdh1 to be released once it is phosphorylated. Did the authors check with their new anti-pY148 antibodies whether the pY148 population of Cdh1 is associated with c-Src or rather is mostly unassociated and soluble.

Response: We thank the reviewer for raising this important question. We have obtained new experimental evidence to address this comment, and the results could be found in **Fig. R1**. As shown in **Fig. 6i**, we found that PDGF treatment, which activates c-Src, led to a marked increase of Cdh1 Y148 phosphorylation. In contrast to the increased p-Y148-Cdh1, the binding between p-Y148-Cdh1 and HA-Src was found reduced in MDA-MB-231 cells after PDGF treatment (**Fig. R1a**), suggesting a reduced binding between Cdh1 and c-Src after Cdh1 Y148 phosphorylation. On the other hand, the amount of total Cdh1 associated with c-Src was not significantly affected after PDGF treatment, which might be owing to the observation that a small portion of Cdh1 was phosphorylated at Y148 in cells (see the response to comment #2 and revised **Supplementary Fig. 6d**).

We also noticed a strong interaction between p-Y148-Cdh1 and c-Src prior to PDGF treatment (**Fig. R1a**). We reasoned that such interaction might be due to the dual binding sites we found between Cdh1 and c-Src (**Fig. 4f**). It is plausible that after N-terminal Y148 phosphorylation, which leads to disassociation of Cdh1 N-terminus from c-Src kinase domain, Cdh1 remains interacting with c-Src N-terminal region through its WD40 domain. Under the condition shown in **Fig. R1a**, upon PDGF treatment, in addition to c-Src-mediated Y148 phosphorylation of Cdh1, activated ERK also promoted S/T phosphorylation of Cdh1. We have demonstrated that 4D-Cdh1 exhibited a reduced binding with c-Src (**Supplementary Fig. 9f**), hence we hypothesized that under basal conditions, although Y148 phosphorylation of Cdh1 leads to the disassociation of N-terminal Cdh1 from the c-Src kinase domain, given the existence of Cdh1-WD40/c-Src-N interaction, the binding of Cdh1/c-Src remains. When a strong stimulation like PDGF treatment occurs, a robust but transient multisite phosphorylation of Cdh1 by both c-Src and other serine/threonine kinases could result in dissociation of Cdh1 from c-Src.

To test this hypothesis, *in vitro* kinase assays of full length (FL) Cdh1 and its N-terminus were performed using immunopurified HA-Src that remained associated with the agarose-conjugated anti-HA antibody. As shown in **Fig. R1b**, c-Src catalyzed the phosphorylation of both FL-Cdh1 and N-Cdh1, while the Y148F mutation completely abolished such phosphorylation. Immediately after the reaction, 500 μ l of ice-cold IP washing buffer containing 1 mM EDTA and 1 μ M saracatinib was added to the reaction mix to stop the kinase reaction. The reaction suspension was then rotated at 4°C for 2 hours followed by spinning down and washing the HA-Src beads to determine the interaction between Cdh1 and HA-Src. We found that after HA-pull down (PD), compared to WT-FL-Cdh1, there was an increased interaction between HA-Src and Y148F-FL-Cdh1 (**Fig. R1b**). On the other hand, the N-

terminal Cdh1 completely lost its interaction with HA-Src while the N-Y148F mutant still bound to HA-Src. This result indicates that the WD40-Cdh1/N-c-Src interaction is important to maintain Cdh1/c-Src interaction after phosphorylated N-terminus of Cdh1 released from c-Src kinase domain. In support of this notion, D1-RLAA-c-Src, which failed to bind WD40-Cdh1 (**Fig. 3I**), was not able to remain contact with Cdh1 after kinase assay incubation (**Fig. R1c**).

Fig. R1. Reduced Cdh1-Src interaction upon Y148 phosphorylation of Cdh1. **a**) IB analysis of WCL and anti-HA IP derived from MDA-MB-231 cells stably expressing the lentiviral constructs as indicated. 4 ng/mL PDGF was used to stimulate Src activity. Cells were serum starved overnight before 30 min PDGF treatment. **b-c**) *In vitro* kinase assay (KA) was performed using immunopurified HA-Src and the purified recombinant Cdh1 proteins as indicated. The kinase assays were terminated by adding ice-cold IP washing buffer containing 1mM EDTA and 1 μ M saracatinib. The reaction mixtures were further rotated at 4°C for 2 hours since the HA-Src proteins were still associated with the anti-HA-agarose beads, the interaction between Cdh1 and Src after kinase assays were determined by examining the binding between Cdh1 and HA-Src using the HA pull down (HA PD) assays.

2. While their new anti-pY148 antibodies provide direct evidence that Y148 is phosphorylated in breast cancer cells, they did not check the stoichiometry of Y148 phosphorylation in Cdh1 in these cells.

Response: Following the reviewer's insightful suggestion, the percentage of Cdh1 Y148 phosphorylation in MDA-MB-231 cells was estimated using the p-Y148-Cdh1 antibody (revised **Supplementary Fig. 6d**). Anti-p-Y148-Cdh1 immunoprecipitations (IP) using cell lysates derived from shCdh1-MDA-MB-231 cells stably expressing HA-Cdh1-WT or the Y148F mutant were performed to generate the 100% p-Y148-Cdh1 and the 0% p-Y148-Cdh1 standards, respectively. These two standard IP samples were normalized to Cdh1 total protein amount using anti-HA IB followed by mixing at the indicated ratio as shown in lanes 1-6. An anti-Cdh1 IP was performed using shCdh1-MDA-MB-231 cells stably expressing HA-Cdh1-WT and normalized as described above (lane 7). Anti p-Y148-Cdh1 IB was carried out using both the standards and the sample IP, the IB:p-Y148-Cdh1 band intensities were determined using ImageJ and Cdh1 Y148 phosphorylation percentage in MDA-MB-231 cells was estimated by comparing to the standards. As shown in revised **Supplementary Figure 6d**, there are approximately 15% Cdh1 was phosphorylated at Y148 in MDA-MB-231 cells. We also acknowledge that this is not a stringent method to quantify the stoichiometry of p-Y148-Cdh1 in cells, quantitative mass spectrometry is required for an accurate assessment.

3. The authors should realize that, in general, glutamate cannot be used as a phosphomimetic for pTyr, because the charge and geometry of the negative charge on the Glu COOH group are totally different from those of the phosphate on pTyr, and for this reason it is hard to interpret any phenotypes they observed for the pY148E Cdh1 mutant.

Response: We fully agree with the reviewer that glutamate cannot faithfully mimic phosphotyrosine due to the distinct side chain structure as shown in **Fig. R2**.

Fig. R2. Structures of phosphotyrosine and glutamate

A number of studies demonstrated such difference. For instance, Y747 phosphorylation of β 3-integrin mediates its binding with Dok1, a process could not be recaptured by using the Y747E mutation (*Anthis NJ et al. (2009). J. Biol. Chem. 284, 36700–36710*). A structural study using E.coli tyrosine kinase Wzc and tyrosine phosphatase Wzb found that the interaction between tyrosine phosphorylated Wzc C-tail and Wzb could not be observed when a Y708E/Y710E/Y711E/Y713E/Y715E-Wzc C-tail peptide was used, again indicating that Y-E substitution failed to mimic phosphotyrosine (*Temel DB et al. (2013). J. Biol. Chem. 288, 15212–15228*). Furthermore, Y55 phosphorylation of MDMX creates a salt bridge to the nearby R18 residue in the MDMX N-terminus, such intra-molecular interaction competes with the binding between p53 and MDMX. However, compared with p-Y55-MDMX, Y55E-MDMX failed to mimic the interaction between p-Y55-MDMX and R18 (*Chan JV et al. (2017). Oncotarget. 8, 112825–112840*). Moreover, the activation of ERK2 requires phosphorylation at both T183 and Y185. A structural study revealed that neither the T183E nor the Y185E substitutions could fully mimic the T183 or the Y185 phosphorylation since these T-E and Y-E substitutions failed to fold into the active conformation observed in p-T183/p-Y185-ERK2 (*Zhang J et al. (1995). Structure. 3, 299–307*). All these reports suggest that in the scenario where the phosphotyrosine serves as a unique amino acid side chain to facilitate protein-protein interaction or to confer a conformational change, the glutamate substitution is unlikely to fully imitate.

However, in situations where the phosphorylation on the tyrosine residue primarily introduces a negative charge to disrupt existing protein-protein interactions, the glutamate mutation could be able to functionally mimic the impact of tyrosine phosphorylation. The latter has also been supported by a number of reports. For example, the interaction between WAVE1 and Sra1 plays a central role in the regulation of WRC (WAVE regulatory complex) activity. Y151 phosphorylation of WAVE1 disrupts the binding between WAVE1 and Sra1, which activates WRC. Similar to p-Y151, Y151E-WAVE1 exhibited reduced binding to Sra1 (*Chen ZC, et al. (2010), Nature. 468, 533-538*). Furthermore, FAK (focal adhesion kinase) interacts with CAS (Crk-associated substrate) through N-terminal SH3 domain of CAS. Analogous to p-Y12-SH3-CAS, Y12E-CAS disrupts the binding between CAS-SH3 and FAK (*Janoštiak R et al. (2011). Mol. Boil. Cell 22, 4256–4267*). In addition, similar to p-Y298-Cdc37, Y298E-Cdc37 lost interaction with its binding partner Raf-1 (*Xu W et al. (2012). Mol. Cell. 47, 434–443*). Moreover, analogous to phyB (phytochrome B) Y104 phosphorylation, Y104E-phyB completely abolishes its binding with PIF3 (*Nito K et al. (2013). Cell Reports. 3, 1970–1979*). Furthermore, Y211 phosphorylation of PCNA (proliferating cell nuclear antigen) inhibits DNA mismatch repair (MMR)

through inhibiting the interaction between PCNA and the mismatch-recognition proteins MutS α and MutS β . Similar to Y211-phosphorylated PCNA, PCNA-Y211E mutation suppresses MMR by disrupting its interaction with MutS α and MutS β (Ortega *et al.* (2015) *Proc. Natl. Acad. Sci. U. S. A.* 112, 5667-5672). In addition, p27 inhibits cell cycle progression by interacting with Cdk2/cyclin A. Y88 phosphorylation of p27 disrupts the binding between p27 and Cdk2/cyclin A. Analogous to p27 Y88 phosphorylation, the Y88E mutation of p27 interferes with the interaction between p27 and Cdk2/cyclin A (Tsytonok *et al.* (2019) *Nat. Commun.* 10, 1-13). Moreover, tyrosine kinase FGFR3 (fibroblast growth factor receptor 3) inhibits human papillomavirus (HPV) replication by catalyzing Y138 phosphorylation of the virus E2 protein, a process serves to disrupt the binding between E2 and BRD4 C-terminal domain. The Y138E mutation of E2 was able to mimic such phosphorylation thus inhibits HPV replication (DeSmet *et al.* (2019) *J. Virol.*).

In the case of Cdh1 N-terminal phosphorylation, it has been well documented that N-terminal serine/threonine phosphorylation or lysine acetylation of Cdh1 disrupts the interaction between Cdh1 and the APC core complex (Keck *et al.* (2007). *J. Cell. Biol.* 178, 371–385; Kim *et al.* (2011). *Cancer Cell.* 20, 487–499). Given the close proximity of Y148 to the identified p-S151 and Ac-K159 sites, we hypothesized that p-Y148 might function similarly to prevent the N-Cdh1-APC interaction. Indeed, when breast cancer cells were treated with Src inhibitor or deleted of Src, an increased Cdh1-APC6 interaction was observed, which is accompanied with decreased Y148 phosphorylation (**Fig. 6j-k**, **Supplementary Fig. 6j-k**). Consistent with this notion, both 4D and Y148E-Cdh1 exhibited reduced binding with APC6, while the non-phosphorylatable 4A and Y148F-Cdh1 displayed a stronger interaction (**Fig. 7e**). In addition, both 4D- and Y148E-Cdh1 were less active in promoting Cdc20-N ubiquitination in cells, further supporting a Y148E might be able to recapture the impact of Y148 phosphorylation in disrupting Cdh1-APC interaction. Moreover, consistent with the previous report that 4D-Cdh1 is mainly localized to the cytosol while 4A-Cdh1 resides in the nucleus (Zhou *et al.* (2003). *J. Biol. Chem.* 278, 12530–12533), our results in **Fig. 7** indicate that Y148 phosphorylation may function analogous to these serine/threonine phosphorylations. Hence, we hope the reviewer could concur with us that due to the technical limitation, at least in our experimental setting, the substitution of Y148 with glutamate could serve to mimic the impact of Y148 phosphorylation on Cdh1-APC interaction, which is one of our major conclusions. On the other hand, it is possible that pY148-Cdh1 might possess additional functions in addition to disrupting Cdh1-APC interaction, which will be definitely worth of future investigation by utilizing newly developed genetic tools.

Ideally, the modified phospho-amino acids phosphotyrosine, phosphoserine and phosphothreonine should be used to substitute Tyr, Ser, and Thr, respectively, to faithfully evaluate the mechanism and the biology of such phosphorylation events. Progress has been made by chemists and geneticists during the past decade in developing genetic code expansion techniques (Chin *et al.* (2017). *Nature.* 550, 53–60). One approach is using orthogonal tRNA synthetase and amber suppression to incorporate non-natural amino acids into proteins. Recent reports have achieved success in placing phosphoserine and phosphotyrosine into proteins (Rogerson *et al.* (2015). *Nat. Chem. Biol.* 11, 496–503; Luo *et al.* (2017). *Nat. Chem. Biol.* 13, 845–849). However, such approach also brings downsides. For example, although proteins with non-natural amino acid substituted could be used to interrogate protein functions in vitro, it is difficult to assess its function in cells or in vivo due to global amber suppression that may occur at other endogenous amber codons.

Reviewer #2:

The revised manuscript has improved greatly and I congratulate the authors for the effort in addressing all reviewers' points. My only concern remains at the conclusion level as it is very difficult to conclude that the role of Cdh1 in breast cancer depends on the Src-Cdh1 axis. I do agree that this connection is very important and the rest of the manuscript is really great. Simply I would recommend to note down the conclusions about the relevance in vivo, which cannot be established from the experiments in the manuscript. For instance the word "governs" in the title may be too strong. Other than that, I think the manuscript is of very high quality and very interesting technical and scientifically and deserves publication.

Response: We thank the reviewer again for recognizing the efforts we have made in the revised manuscript. We also agree with the reviewer that the tumor suppressor function of Cdh1 in breast cancer is attributed to many different signaling pathways that Cdh1 modulates. Apparently, Src is only one of them. As kindly suggested, in the revised manuscript, we have toned down the conclusion on the Cdh1/Src signaling axis. Furthermore, as suggested, we changed our title to “***Interplay between c-Src and the APC/C Co-activator Cdh1 Regulates Mammary Tumorigenesis***”.

Reviewer #3:

Overall, the authors have address most of my previous concerns. In particular, they have provided reasonable explanations and data to understand and clarify the initially reported variability in the levels of Plk1 through cell cycle phases with high Cdh1. They have also clarify my concerns regarding the remaining Cdh1 expression in CRISPR/Cas9 mediated Cdh1 knockout cells. But more importantly, they have engaged on a series of new experiments to clarify the consequences of abnormal mitosis in Cdh1 breast cancer cells, its relevance and the potential consequences of the newly created genotoxic stress. In addition, they have extended the implication of their results to preclinical experimental mouse models. Overall, the manuscript has largely improved from the review process and provides a well-supported relevant novel insight into the interplay between Cdh1, Src and APC/C.

Response: We sincerely appreciate the reviewer for the insightful comments that helped us strengthen the manuscript.

REVIEWERS' COMMENTS:

Reviewer #2 (Remarks to the Author):

The revised version of the manuscript is of very high quality and deserves publication.

Point-by-Point Responses to Reviewers' Critiques (NCOMMS-18-18370B)

REVIEWERS' COMMENTS

Reviewer #2 (Remarks to the Author):

The revised version of the manuscript is of very high quality and deserves publication.

Response: Once again, we sincerely appreciate the insightful comments from the reviewers that have been very helpful in guiding us along our revision.